

# On the Q operator and the spectrum of the XXZ model at root of unity

Yuan Miao[1*], Jules Lamers[2] and Vincent Pasquier[3]

**1** Institute for Physics and Institute for Theoretical Physics, University of Amsterdam, Postbus 94485, 1090 GL Amsterdam, The Netherlands
**2** School of Mathematics and Statistics, University of Melbourne, Vic 3010, Australia
**3** Institut de Physique Théorique, Université Paris Saclay, CEA, CNRS, F-91191 Gif-sur-Yvette, France

★ y.miao@uva.nl

## Abstract

The spin-$\frac{1}{2}$ Heisenberg XXZ chain is a paradigmatic quantum integrable model. Although it can be solved exactly via Bethe ansatz techniques, there are still open issues regarding the spectrum at root of unity values of the anisotropy. We construct Baxter's Q operator at arbitrary anisotropy from a two-parameter transfer matrix associated to a complex-spin auxiliary space. A decomposition of this transfer matrix provides a simple proof of the transfer matrix fusion and Wronskian relations. At root of unity a truncation allows us to construct the Q operator explicitly in terms of finite-dimensional matrices. From its decomposition we derive truncated fusion and Wronskian relations as well as an interpolation-type formula that has been conjectured previously. We elucidate the Fabricius–McCoy (FM) strings and exponential degeneracies in the spectrum of the six-vertex transfer matrix at root of unity. Using a semicyclic auxiliary representation we give a conjecture for creation and annihilation operators of FM strings for all roots of unity. We connect our findings with the 'string-charge duality' in the thermodynamic limit, leading to a conjecture for the imaginary part of the FM string centres with potential applications to out-of-equilibrium physics.

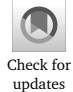

# 1  Introduction

To study the dynamics of a quantum many-body system, it is of vital importance to know the full spectrum, i.e. *all* eigenstates of the Hamiltonian. For instance, with the knowledge of the spectrum it is possible to calculate the density matrix, which is the central object to study the entanglement properties of many-body systems [1]. Furthermore, one can use the spectrum to study the quantum quench problem, a paradigmatic example of out-of-equilibrium dynamics following the logic of the Quench Action [2,3]. The spectrum of quantum Hamiltonian is also closely related to the partition function of the corresponding classical statistical physics model, which can be used to detect phase transitions [4].

However, it is very difficult to obtain the full spectrum, especially for strongly interacting systems. Some models can be mapped into a free fermion (parafermion) model through a non-local mapping [5–8], resulting in relatively simple spectra. However, generally such a construction does not exist for an interacting many-body system.

Fortunately, there are strongly interacting models amenable to exact methods. Using quantum integrability [9,10] it is possible to obtain the spectrum. For these models, the transfer matrices are the generating functions of the conserved (quasi-)local charges that contribute to the dynamics in the thermodynamic limit. In this setting the generalized Gibbs ensemble (GGE) [11] has proven a crucial tool to study quench problems in integrable models [12], with a close relation to the transfer matrix approach. For quantum integrable models Bethe-ansatz techniques provide an exact characterisation of the spectrum. The eigenvalues of the transfer matrix are obtained via a set of rapidity parameters whose physical values, called Bethe roots, obey the Bethe equations [13]. The latter follow in a straightforward way from a difference equation known as Baxter's TQ relation [9,14–16]. The problem of determining whether the solutions of the Bethe equations provide a complete set of eigenstates has drawn attention in

both mathematics and physics [17–24].[1] The results in this article can help to understand the completeness at root of unity by describing and illustrating different structures present in the spectrum.

It is in principle possible to obtain the full spectrum of the spin-$\frac{1}{2}$ XXZ chain at root of unity by solving Baxter's TQ relation. The main problem is that there are (a large number of) degenerate eigenstates of the transfer matrix, which cannot be easilly interpreted as solution to the (functional) TQ relation [25–29]. One of the difficulties specific to the root of unity case, is that given a solution of the TQ relation which corresponds to an eigenstate, it is possible to multiply the Q function by certain polynomial factors left undetermined by the TQ relation, which provide other solutions called descendants. When these new solutions turn out to correspond to true eigenstates, they can be interpreted as the presence of bound states called 'exact strings' (Fabricius–McCoy strings) that do not scatter with other excitations [16,20,30–36]. In Ref. [32], Fabricius and McCoy proposed to organise the descendants into irreducible representations of the loop algebra of $\mathfrak{sl}_2$, cf. [37–39]. They can be obtained by acting with creation operators on highest weight states. These representations have dimension $2^n$ and are characterised by a degree-$n$ polynomial called the Drinfeld polynomial which, unlike the regular eigenvalues of the Q operator, shows when the descendants are present.

In this article, we propose an operatorial approach to this problem. Namely, building on Refs. [36,40] we define a two-parameter transfer matrix that is used to construct the Q operator, i.e. the solution to the matrix TQ relation. Once the Q operator is obtained, all the physical solutions to Bethe equations can be extracted as the zeroes of the eigenvalues, including the solutions associated to exact strings. Baxter found a Q operator for the six-vertex model by directly solving the matrix TQ relation [9]. Our Q operator is closer to that proposed by Korff [33]. It can be easily implemented on a laptop for system size $N \leq 16$ when the denominator $\ell_2$ of the root of unity is less than 10. We illustrate with several explicit examples the appearance of Fabricius–McCoy strings as well as Bethe roots at infinity. In addition, we prove in an elementary way some conjectured results in Refs. [32,34,41]. Closely related to the Q operator, we obtain a second solution of the TQ relation corresponding to the polynomial $P$ of Pronko and Stroganov [25]. The eigenvalues of the P and Q operators both contain factors associated to Fabricius–McCoy strings. The product of these factors coincides with the Drinfeld polynomial of [32,39], see also Section 8.5. In this setting, the degeneracy of the loop-algebra multiplets is recovered from the different ways to decompose the Drinfeld polynomial into two factors belonging to Q and P, respectively, cf. Section 7.1. We present conjectures for the creation and annihilation operators for eigenstates associated to the degeneracies in Section 9.

Furthermore, we find that the existence of Fabricius–McCoy strings is closely related to the quasilocal Z charges [42–44] that lie at the origin of the fractal structure of the spin Drude weight in the spin-$\frac{1}{2}$ XXZ model at root of unity [45]. The 'string-charge duality' [46], a functional relation that links the root density of bound states to the generating functions of conserved charges based on the thermodynamic Bethe ansatz, can be illustrated by considering the spectrum with Fabricius–McCoy strings at root of unity. In addition, we give a conjecture about the string centres of Fabricius–McCoy strings based on the string hypothesis.

**Outline.** We start in Section 2 with some examples to illustrate the problems that one encounters at root of unity. The main body of the paper can be divided into two parts. Sections 3–6 deal with the formalism. After a short introduction to the framework of the quantum inverse scattering method in Section 3, we define the two-parameter transfer matrices and derive

---

[1] The Bethe ansatz has been proven to be complete for the XXZ spin chain with generic inhomogeneities and twist [18]. At a practical level these results are sufficient as one can calculate physical quantities for the inhomogeneous model and take the homogeneous limit at the end. In this work we focus on the homogeneous XXZ spin chain. We do not address completeness at a rigorous level here.

their factorisation properties in Section 4. These results allow us to construct the Q operator explicitly as well as transfer matrix fusion and Wronskian relations for arbitrary anisotropy parameter $\Delta$ in Section 5. In Section 6 we prove the truncated transfer matrix fusion and Wronskian relations at root of unity, leading to an interpolation formula that has been previously conjectured in Refs. [32, 34, 41].

Sections 7–10 deal with the applications to understanding the spectrum of the XXZ spin chain (and six-vertex model) at root of unity. Equipped with the exact form of the Q operator, we demonstrate the descendant tower structure with numerous examples in Sections 7–8. We present a conjecture for the Fabricius–McCoy string creation and annihilation operators in Section 9. Discussions on the thermodynamic limit of solutions with Fabricius–McCoy strings follow in Section 10.

We conclude in Section 11. Some technical details are provided in the Appendices.

## 2 Motivation

Before we start with the technical part, let us describe features that motivate our study of the spectrum of the XXZ spin chain at root of unity.

### 2.1 XXZ spin chain

Consider the quasiperiodic spin-$\frac{1}{2}$ Heisenberg XXZ spin chain of $N$ sites, with Hamiltonian

$$
\begin{aligned}
\mathbf{H} = \sum_{j=1}^{N-1} & \left( \frac{1}{2} \big( \sigma_j^+ \sigma_{j+1}^- + \sigma_j^- \sigma_{j+1}^+ \big) + \frac{\Delta}{4} \big( \sigma_j^z \sigma_{j+1}^z - 1 \big) \right) \\
& + \frac{e^{i\phi}}{2} \sigma_N^+ \sigma_1^- + \frac{e^{-i\phi}}{2} \sigma_N^- \sigma_1^+ + \frac{\Delta}{4} \big( \sigma_N^z \sigma_1^z - 1 \big) .
\end{aligned}
\tag{2.1}
$$

Here $\sigma_j^\pm := (\sigma_j^x \pm i \sigma_j^y)/2$ and $\sigma_j^\alpha = \mathbb{1}^{\otimes(j-1)} \otimes \sigma^\alpha \otimes \mathbb{1}^{\otimes(N-j)}$ denotes the $\alpha$th Pauli matrix acting at site $j$. The Hamiltonian is hermitian provided the twist $\phi$ and anisotropy parameter $\Delta$ are real. The latter can be parametrised as

$$
\Delta = \frac{q + q^{-1}}{2} = \cosh \eta , \qquad q = e^\eta .
\tag{2.2}
$$

We will be particularly interested in case where $q$ is a root of unity, i.e.

$$
\eta = i\pi \frac{\ell_1}{\ell_2} , \qquad q^{2\ell_2} = 1 ,
\tag{2.3}
$$

for $\ell_1, \ell_2 \in \mathbb{Z}_{>0}$ coprime. We will sometimes write

$$
\varepsilon := q^{\ell_2} \in \{\pm 1\} .
\tag{2.4}
$$

The root-of-unity case is important for the gapless (massless) regime $-1 < \Delta < 1$.

By partial isotropy (2.1) commutes with $\mathbf{S}^z = \sum_{j=1}^N \sigma_j^z / 2$, so we can fix the magnetisation $\mathbf{S}^z = \frac{1}{2} N - M$, $0 \le M \le N$. Away from the isotropic point, $\Delta \ne \pm 1$, the usual ($\mathfrak{sl}_2$) ladder operators $\sum_j \sigma_j^\pm$ are no longer relevant. Instead, the modification ($q$-deformation)

$$
\mathbf{S}^\pm = \sum_{j=1}^N q^{\sigma_1^z/2} \cdots q^{\sigma_{j-1}^z/2} \sigma_j^\pm q^{-\sigma_{j+1}^z/2} \cdots q^{-\sigma_N^z/2} , \qquad q^{\sigma^z/2} = \mathrm{diag}(q^{1/2}, q^{-1/2}),
\tag{2.5}
$$

will have a role to play, although they do not commute with $\mathbf{H}$ in general either.[2] These operators obey a variant of the relations of $U_q(\mathfrak{sl}_2)$, quantum $\mathfrak{sl}_2$:

$$[\mathbf{S}^z, \mathbf{S}^\pm] = \pm \mathbf{S}^\pm, \qquad [\mathbf{S}^+, \mathbf{S}^-] = [2\mathbf{S}^z]_q, \qquad [x]_q := \frac{q^x - q^{-x}}{q - q^{-1}}. \tag{2.6}$$

Here the $q$-deformed integer applied to $x = 2\mathbf{S}^z$ is understood via $q^x = \exp(\eta x)$, cf. (2.5). Note that the representation (2.5) breaks the symmetry of (2.1) under parity, i.e. spatial reflection. We will also encounter the parity-conjugate (opposite) representation of $U_q(\mathfrak{sl}_2)$, which we denote by $\bar{\mathbf{S}}^z = \mathbf{S}^z$ and $\bar{\mathbf{S}}^\pm = \mathbf{S}^\pm|_{q \mapsto q^{-1}}$. We call such representations on the spin-chain Hilbert space $(\mathbb{C}^2)^{\otimes N}$ 'global'. See Appendix A for more about $U_q(\mathfrak{sl}_2)$ and its representations.

The XXZ spin chain is exactly solvable by Bethe ansatz [50–54]. Here one seeks eigenvectors $|\{u_m\}_{m=1}^M\rangle$ parametrised in a clever way that involve $M$ unknown parameters $u_m \in \mathbb{C}$. We will outline the algebraic Bethe ansatz, i.e. the construction of $|\{u_m\}_{m=1}^M\rangle$ through the quantum inverse scattering method, in Section 3. The outcome is that $|\{u_m\}_{m=1}^M\rangle$ is an eigenvector of (2.1) provided the rapidities $u_m$ solve the Bethe equations

$$\left(\frac{\sinh(u_m + \eta/2)}{\sinh(u_m - \eta/2)}\right)^N \prod_{m'(\neq m)}^M \frac{\sinh(u_m - u_{m'} - \eta)}{\sinh(u_m - u_{m'} + \eta)} = e^{-i\phi}, \qquad 1 \le m \le M. \tag{2.7}$$

Its (twisted) momentum and energy are

$$p = \phi + \sum_{m=1}^M p_m, \qquad E = \sum_{m=1}^M E_m, \tag{2.8}$$

where the (quasi)momentum and (quasi)energy associated with the $m$th Bethe root are

$$\begin{aligned} p_m &= i \log \frac{\sinh(u_m - \eta/2)}{\sinh(u_m + \eta/2)}, \\ E_m &= \cos(p_m) - \Delta = \frac{\sinh^2 \eta}{2 \sinh(u_m + \eta/2) \sinh(u_m - \eta/2)}. \end{aligned} \tag{2.9}$$

Here the momentum is defined such that $e^{ip}$ is the eigenvalue of the (twisted, right) translation operator, see Appendix B.2. The Bethe equations (2.7) can be rewritten as

$$e^{iNp_m} \prod_{m'(\neq m)}^M S(p_m, p_{m'}) = e^{-i\phi}, \qquad 1 \le m \le M, \tag{2.10}$$

where the scattering phase for two Bethe roots is

$$S(p_m, p_n) = -\frac{e^{i(p_m + p_n)} - 2\Delta e^{ip_n} + 1}{e^{i(p_m + p_n)} - 2\Delta e^{ip_m} + 1} = \frac{\sinh(u_m - u_n - \eta)}{\sinh(u_m - u_n + \eta)}. \tag{2.11}$$

By slight abuse of notation we will denote the final expression by $S(u_m, u_n)$.

Let us stress that, more precisely, by 'Bethe roots' we will mean the roots of the $Q$ function, i.e. the eigenvalue of Baxter's $Q$-operator, see Section 5.5. Each set $\{u_m\}$ of Bethe roots in the strip $-\pi/2 < \text{Im}\, u \le \pi/2$ [3] specifies an eigenstate. The distribution of Bethe roots in

---

[2] One gets $U_q(\mathfrak{sl}_2)$ symmetry by changing the boundary terms in (2.1) to obtain 'open boundaries' [47, 48] or nonlocal twists arising from 'braid translations' [49]. Alternatively, at special values of the twist in (2.1) the $q$-deformed ladder operators act as a sort symmetry [47]. Namely, if $\mathbf{H}^k := \mathbf{H}|_{\phi = k i \eta}$ then $\mathbf{S}^\pm \mathbf{H}^{2\pm(N-2M)} = \mathbf{H}^{\pm(N-2M)} \mathbf{S}^\pm$ on vectors with $\mathbf{S}^z = \frac{1}{2}N - M$.

[3] Note that $u + i\pi \equiv u$ since $\sinh(u + i\pi) = -\sinh u$ and there is an even number of such factors in the Bethe equations (2.7) and eigenvalues (2.8)–(2.9).

the complex plane is crucial to understand the thermodynamic properties of integrable systems, leading to thermodynamic Bethe ansatz (TBA). As we will see in Section 10 the study of the spectrum at finite system size $N$ may already shed light on the origin of string-charge duality [46] which is important to study dynamics in the thermodynamic limit.

The main question that we set out to understand is *what is the structure of the spectrum of the XXZ spin chain at root of unity?* Despite of its exact characterisation this is not yet understood systematically. In the following two subsections we discuss phenomena that motivate our study. Although we focus on the spin-chain perspective, note that these phenomena are equally relevant for the root-of-unity case of the six-vertex model, whose transfer matrix gives rise to the XXZ Hamiltonian (2.1), see Section 3.

## 2.2 Bethe roots at infinity

When $\Delta = \pm 1$ the spectrum exhibits degeneracies reflecting the model's isotropy ($\mathfrak{sl}_2$-invariance). Although there are additional degeneracies in the spectrum of **H** due to parity invariance, the latter are lifted when taking into account the momentum — or any other parity-odd charge generated by the transfer matrix, or indeed the transfer matrix itself. Let us focus on the degeneracies in the joint spectrum and return to isotropy. The lowering operator $\sum_j \sigma_j^-$ of $\mathfrak{sl}_2$ can be thought of as adding a magnon with vanishing quasimomentum; indeed, it can be shown that a Bethe-ansatz vector has highest weight iff $p_m \neq 0 \mod 2\pi$ [10]. The isotropic limit of (2.9) (rescale $u_m \rightsquigarrow \eta u_m$ and let $\eta \to 0$) shows that $p_m = 0$ corresponds to $u_m = \pm\infty$. That is, for the XXX spin chain Bethe roots at infinity signal descendant states.

Now switch on the anisotropy parameter $\Delta \neq \pm 1$. Then $\sum_j \sigma_j^\pm$ are no longer symmetries, so one might expect there to be no more Bethe roots at infinity. However, numerical solutions of the Bethe equations (e.g. via the recipe from Appendix C.1) show that infinite Bethe roots are in fact present for the XXZ spin chain.

To see when Bethe roots at infinity occur we turn to the Bethe equations (2.7). Write $n_{\pm\infty}$ for the number of Bethe roots at $\pm\infty$, so that of course $n_{+\infty} + n_{-\infty} \leq M$. Note that $p_m \to \mp i\eta$ for $u_m \to \pm\infty$, while $S(u_m, u_n) \to e^{\mp 2\eta}$ as long as $u_n$ is finite or goes to $\mp\infty$. Let us assume that the roots at $+\infty$ do not scatter ($S = 1$) amongst each other, and likewise for those at $-\infty$. Then the Bethe equation for the infinite root $u_m = \pm\infty$ becomes [20, 30, 55]

$$\exp\!\big[\pm\big(N - 2(M - n_{\pm\infty})\big)\eta\big] = \exp(-i\phi). \tag{2.12}$$

Let us examine the possible values of $n_{\pm\infty} \in \mathbb{N}$ in each regime of the XXZ spin chain.

For $\eta \in \mathbb{R}$, i.e. in the gapped regime ($|\Delta| > 1$), and twist $\phi \in \mathbb{R}$ the only solutions are

$$\phi = 0, \qquad n_{+\infty} = n_{-\infty} = \frac{2M - N}{2}. \tag{2.13}$$

This implies that physical solutions with Bethe roots at infinity only exist when $M > N/2$ ('beyond the equator') and $N$ is even, and that the infinite Bethe roots come in pairs.

Next consider the gapless regime ($|\Delta| < 1$). When $\eta \in i(\mathbb{R} \setminus \pi\mathbb{Q})$, so *not* at root of unity, there are more possibilities to allow for Bethe roots at infinity, as long as the twist $\phi$ is tuned to an (even or odd, depending on the parity of $N$) integer multiple of $i\eta$:

$$n_{\pm\infty} = \frac{2M - N \mp i\phi/\eta}{2}. \tag{2.14}$$

In particular, the number of roots at $+\infty$ and $-\infty$ do not coincide if $\phi \neq 0$. Finally, at root of unity $\eta = i\pi\ell_1/\ell_2$ the condition becomes

$$n_{\pm\infty} = \frac{2M - N \mp i\phi/\eta + 2\pi i k_\pm/\eta}{2}, \qquad k_\pm \in \mathbb{Z}. \tag{2.15}$$

In this case there is an additional condition that we will discuss momentarily, see (2.17).

The meaning of $n_{\pm\infty} \neq 0$ for the XXZ model can be understood from the algebraic Bethe ansatz (see Section 3): Bethe roots at infinity correspond to applications of the lowering operators of the global $U_q(\mathfrak{sl}_2)$ algebra (see Appendix A.1). Namely, when $N$ is even and $\phi$ vanishes, each eigenstate beyond the equator (say at $M' > N/2$) is obtained from a Bethe eigenvector at $M = N - M' < N/2$ by $M' - N/2 = N/2 - M$ applications of the parity-invariant product $\mathbf{S}^- \bar{\mathbf{S}}^-$. Up to an overall factor the result is the spin-reverse of the Bethe vector we started with. That is, we find that

$$|\{v_{m'}\}_{m'=1}^{M'}\rangle \propto \left(\mathbf{S}^- \bar{\mathbf{S}}^-\right)^{N/2-M} |\{u_m\}_{m=1}^M\rangle \propto \prod_{k=1}^{N} \sigma_k^x \, |\{u_m\}_{m=1}^M\rangle. \qquad (2.16)$$

If $N$ is odd the XXZ spin chain is still invariant under a global spin flip, reversing $\uparrow \leftrightarrow \downarrow$ everywhere, but there are no infinite roots and we have not been able to find a simple relation between the Bethe roots $\{v_{m'}\}_{m'=1}^{M'}$ and $\{u_m\}_{m=1}^M$ on the two sides of the equator.

At root of unity we have to be a little more careful since $\mathbf{S}^-$ and $\bar{\mathbf{S}}^-$ are nilpotent. Here an additional requirement is needed to ensure that (2.16) is nonzero:

$$0 \leq n_{+\infty} \leq \ell_2 - 1, \qquad 0 \leq n_{-\infty} \leq \ell_2 - 1. \qquad (2.17)$$

In particular, in the periodic case ($\phi = 0$) there is at most one nonzero solution to (2.15) in the range (2.17), leaving only three possible scenarios:

$$n_{+\infty} = n_{-\infty}; \qquad n_{+\infty} > n_{-\infty} = 0; \qquad n_{-\infty} > n_{+\infty} = 0. \qquad (2.18)$$

The machinery that we shall develop in Sections 4–6 will help to understand the structure present in the spectrum due to such infinite Bethe roots. The conclusion (2.18) will be useful in Section 8.1 when we discuss applications to the spectrum of the XXZ model.

## 2.3 Fabricius–McCoy strings

At root of unity — including, but certainly not limited to, the free-fermion point $\Delta = 0$, $\eta = i\pi/2$ — the spectrum of the XXZ spin chain has many degeneracies [37]. Fabricius and McCoy realised [30, 31] that this is related to solutions to the Bethe equations (2.7) that contain exact $\ell_2$-strings. An earlier work of Baxter [16] has shown similar solutions in the 8-vertex model (XYZ model). A *Fabricius–McCoy (FM) string* consists of $\ell_2$ Bethe roots that are equally spaced (in the imaginary direction, chosen in the interval $(-i\pi/2, i\pi/2]$ as explained in Footnote 3 on p. 6) around a string centre $\alpha^{\mathrm{FM}} \in \mathbb{C}$:

$$u_m = \alpha^{\mathrm{FM}} + \frac{2m - 1 - \ell_2}{2\ell_2} i\pi, \qquad 1 \leq m \leq \ell_2. \qquad (2.19)$$

This describes a bound state, which as a whole does not scatter with other Bethe roots:

$$\prod_{m=1}^{\ell_2} S(u_m, u_{m'}) = 1, \qquad \ell_2 < m' \leq M. \qquad (2.20)$$

A FM string is not only 'transparent' in scattering, but has vanishing energy; indeed, it does not carry any local charge generated by transfer matrix $\mathbf{T}_{1/2}(u)$, except that it may carry momentum $\pi$ as we explain Section 8. It actually carries a specific type of quasilocal charges called the *Z charges* [42, 43, 45]; the physical implications of this will be discussed in Section 10.2.

When any FM string is present amongst the Bethe roots for a given eigenstate, it is not possible to determine the location of the string centre $\alpha^{\mathrm{FM}}$ by solving the (functional) TQ

relation using method in Appendix C.1. In Ref. [31] the authors derive an equation for the string centres; we find (see Section 8.5) that its solutions correspond to eigenvectors of **H** that are not eigenvectors for the Q operator (cf. the following example), and in particular the FM strings obtained following [31] do not correspond to zeroes of the Q function. In this article we present a way that enables us to determine $\alpha^{\text{FM}}$ in a way that remedies this. A conjecture for the imaginary part of the FM string centres is given in Section 10.2.

*Example.* To preview the kind of results we are able to get with our methods consider a homogeneous spin chain with $N = 6$, $\phi = 0$ and $\Delta = 1/2 = \cos(\pi/3)$ so $\ell_1 = 1$ and $\ell_2 = 3$. At $M = 3$ there are two degenerate states with the same eigenvalues of $\mathbf{T}_s(u)$, $2s \in \mathbb{Z}_{>0}$. These eigenvalues moreover coincide with those of $|\uparrow\uparrow\uparrow\uparrow\uparrow\uparrow\rangle$ and $|\downarrow\downarrow\downarrow\downarrow\downarrow\downarrow\rangle$ up to a sign. Using the techniques from Section 6 we find that the two degenerate states have FM strings whose centres we can determine exactly:

$$
\begin{aligned}
u_m &= \alpha_1^{\text{FM}} + (m-2)\frac{i\pi}{3}, \\
v_m &= \alpha_2^{\text{FM}} + (m-2)\frac{i\pi}{3},
\end{aligned}
\qquad
\alpha_{1,2}^{\text{FM}} = \pm\frac{\log(10+3\sqrt{11})}{6} + \frac{i\pi}{6}, \qquad 1 \le m \le 3, \qquad (2.21)
$$

where the Bethe roots are found analytically using the truncated two-parameter transfer matrix that we will introduce in Section 6.

The degeneracies signal some symmetry acquired by the XXZ spin chain at root of unity. Namely, in this case a there is a representation of the loop algebra of $\mathfrak{sl}_2$ [37–39]. In short: although the $\ell_2$th powers of $\mathbf{S}^{\pm}$ and $\bar{\mathbf{S}}^{\pm}$ vanish one can regularise these operators to obtain generators of the loop algebra. Its lowering operators produce eigenvectors that correspond to the FM string (2.21). In fact, the eigenspace is spanned by any two out of the four vectors

$$
\begin{aligned}
\lim_{\eta \to i\pi/3} \frac{(\mathbf{S}^-)^3}{[3]_q!} |\uparrow\uparrow\uparrow\uparrow\uparrow\uparrow\rangle, &\qquad |\{u_1, u_2, u_3\}\rangle, \\
\lim_{\eta \to i\pi/3} \frac{(\bar{\mathbf{S}}^-)^3}{[3]_q!} |\uparrow\uparrow\uparrow\uparrow\uparrow\uparrow\rangle, &\qquad |\{v_1, v_2, v_3\}\rangle.
\end{aligned}
\qquad (2.22)
$$

(Here $[3]_q! = [3]_q[2]_q$.) That is, the Bethe vectors on the right in (2.22) are nontrivial linear combinations of the two 'loop descendants' on the left. In particular, the latter are not eigenvectors of the two-parameter transfer matrix $\tilde{\mathbb{T}}(x, y)$ that we will introduce in Section 6.2, and hence neither of the Q operator.

## 3 Basics of the QISM

Let us briefly review the quantum inverse scattering method (QISM). We start with the Lax operator associated to a single spin-$\frac{1}{2}$ site. Consider an auxiliary space $V_a$, which is an irreducible representation of $U_q(\mathfrak{sl}_2)$. Let us work with generators $\mathbf{S}_a^{\pm}$ along with $\mathbf{K}_a = q^{\mathbf{S}_a^z}$. In terms of this formulation the commutation relations (2.6) read

$$
\mathbf{K}_a \mathbf{S}_a^{\pm} \mathbf{K}_a^{-1} = q^{\pm 1} \mathbf{S}_a^{\pm}, \quad [\mathbf{S}_a^+, \mathbf{S}_a^-] = \frac{\mathbf{K}_a^2 - \mathbf{K}_a^{-2}}{q - q^{-1}}. \qquad (3.1)
$$



Figure 1: Graphical representation of the transfer matrix with $N$ sites. The (cyclic) thick red line is the auxiliary space and the black lines represent the spin-$\frac{1}{2}$ spaces. The twist operator is labelled by $\phi$.

Although the sites of the spin chain will always have spin $\frac{1}{2}$, we will consider various representations $V_a$, summarised in Appendix A.2. The Lax operator on $V_a \otimes \mathbb{C}^2$ is

$$
\begin{aligned}
\mathbf{L}_{aj}(u) &= \sinh(u)\frac{\mathbf{K}_a + \mathbf{K}_a^{-1}}{2} + \cosh(u)\frac{\mathbf{K}_a - \mathbf{K}_a^{-1}}{2}\sigma_j^z + \sinh(\eta)\big(\mathbf{S}_a^+ \sigma_j^- + \mathbf{S}_a^- \sigma_j^+\big) \\
&= \frac{1}{2}\begin{pmatrix} e^u \mathbf{K}_a - e^{-u}\mathbf{K}_a^{-1} & 2\sinh(\eta)\mathbf{S}_a^- \\ 2\sinh(\eta)\mathbf{S}_a^+ & e^u \mathbf{K}_a^{-1} - e^{-u}\mathbf{K}_a \end{pmatrix}_j \\
&= \begin{pmatrix} \sinh\big(u + \eta\,\mathbf{S}_a^z\big) & \sinh(\eta)\mathbf{S}_a^- \\ \sinh(\eta)\mathbf{S}_a^+ & \sinh\big(u - \eta\,\mathbf{S}_a^z\big) \end{pmatrix}_j,
\end{aligned}
\tag{3.2}
$$

where the matrix acts on the spin-$\frac{1}{2}$ representation at site $j$ and the entries are operators on the auxiliary space. Importantly, the Lax operator obeys the RLL relations

$$
\mathbf{R}_{ab}(u-v)\,\mathbf{L}_{aj}(u)\,\mathbf{L}_{bj}(v) = \mathbf{L}_{bj}(v)\,\mathbf{L}_{aj}(u)\,\mathbf{R}_{ab}(u-v)
\tag{3.3}
$$

on $V_a \otimes V_b \otimes \mathbb{C}^2$, where $V_b \cong V_a$ is a second copy of the auxiliary space. We will parametrise the entries of the R-matrix in such a way that the algebraic Bethe ansatz immediately gives the usual Bethe equations, which requires the shift by $\eta/2$ in (3.3). In case the auxiliary space is the spin-$\frac{1}{2}$ representation the R-matrix which can be expressed as

$$
\mathbf{R}_{ab}(u) = \begin{pmatrix} \sinh(u+\eta) & 0 & 0 & 0 \\ 0 & \sinh u & \sinh \eta & 0 \\ 0 & \sinh \eta & \sinh u & 0 \\ 0 & 0 & 0 & \sinh(u+\eta) \end{pmatrix}_{ab},
\tag{3.4}
$$

and the Lax operator (3.2) is given by the same matrix with $u \rightsquigarrow u - \eta/2$.[4]

Now consider $N$ spin-$\frac{1}{2}$ sites, each with its own (local) Lax operator $\mathbf{L}_{aj}(u)$. The monodromy matrix (global Lax operator) on $V_a \otimes (\mathbb{C}^2)^{\otimes N}$ is[5]

$$
\mathbf{M}_a(u, \phi) = \mathbf{L}_{aN}(u)\cdots\mathbf{L}_{a2}(u)\mathbf{L}_{a1}(u)\mathbf{E}_a(\phi),
\tag{3.5}
$$

where we included $\phi$ through the twist operator $\mathbf{E}_a(\phi)$. We will only consider diagonal twists, so that the R-matrix commutes with $\mathbf{E}_a(\phi)\mathbf{E}_b(\phi)$. When $V_a$ is the spin-$\frac{1}{2}$ irrep we have $\mathbf{E}_a(\phi) = \mathrm{diag}(e^{\mathrm{i}\phi}, 1)$; see Appendix B.1 for the other representations that we will use. It is easy to show (using the 'train argument') that the monodromy matrix obeys the RLL relations (3.3) too. There are several things we get out of this setup.

First of all we can construct the Hamiltonian (2.1) and its symmetries. The transfer matrix $\mathbf{T}$ is the trace of monodromy matrix $\mathbf{M}_a$ over the auxiliary space

$$
\mathbf{T}(u, \phi) = \mathrm{tr}_a\,\mathbf{M}_a(u, \phi),
\tag{3.6}
$$

---

[4] When $V_a$ is two-dimensional the Lax matrix (3.2) is symmetric in the sense that $\mathbf{P}\mathbf{L}\mathbf{P} = \mathbf{L}$, with $\mathbf{P}$ the permutation on $\mathbb{C}^2 \otimes \mathbb{C}^2$. This allows us to reverse the roles of $a$ and $j$ in (3.2). One obtains the matrix in (3.4) with $u$ replaced by $u - \eta/2$ using $\mathbf{S}_a^\pm = \sigma_a^\pm$ and $\mathbf{K}_a = \exp(\eta\,\sigma_a^z/2) = \mathrm{diag}(e^{\eta/2}, e^{-\eta/2})$.

[5] Inhomogeneity parameters can be effortless included as usual. Since we are interested in the spectrum of the homogeneous spin chain we omit the inhomogeneities from the start to keep the notation light.

provided this trace can be taken — which is not obvious when $V_a$ is infinite dimensional; we will get back to this. The RLL relations[6] imply that this is a one-parameter family of commuting operators,

$$[\mathbf{T}(u,\phi),\mathbf{T}(v,\phi)] = 0, \qquad \text{for all} \quad u,v \in \mathbb{C}. \tag{3.7}$$

As a consequence any expansion in $u$ generates a hierarchy of commuting operators that do not depend on $u$. Particularly important charges are obtained by taking logarithmic derivatives with respect to the spectral parameter $u$:

$$\mathbf{I}^{(j)} = -\mathrm{i} \frac{\mathrm{d}^{j-1}}{\mathrm{d}u^{j-1}} \log \mathbf{T}(u,\phi)\Big|_{u=\eta/2}. \tag{3.8}$$

In particular, when the auxiliary space has spin $\frac{1}{2}$ — the transfer matrix of the six-vertex model — this includes the twisted translation operator ($j=1$) and the XXZ Hamiltonian (2.1) ($j=2$), see Appendix B.2. More generally, when $V_a \cong \mathbb{C}^{2s+1}$ is the spin-$s$ irrep of $U_q(\mathfrak{sl}_2)$ with $s \geq 1$ the resulting conserved charges are quasilocal [42, 43, 45].

Next, the spectrum of the transfer matrix with $V_a \cong \mathbb{C}^2$, and in particular the XXZ spin chain, can be characterised via the algebraic Bethe ansatz. Keeping $V_a \cong \mathbb{C}^2$ the monodromy matrix can be written as a matrix in auxiliary space,

$$s = 1/2: \qquad \mathbf{M}_a(u,\phi) = \begin{pmatrix} \mathbf{A}(u) & \mathbf{B}(u) \\ \mathbf{C}(u) & \mathbf{D}(u) \end{pmatrix}_a \begin{pmatrix} e^{\mathrm{i}\phi} & 0 \\ 0 & 1 \end{pmatrix}_a, \tag{3.9}$$

with entries that act on the spin-chain Hilbert space $(\mathbb{C}^2)^{\otimes N}$. Let us stress that this point of view is opposite to that in (3.2), where we were thinking of the Lax operator as a $2 \times 2$ matrix on the physical space associated to site $j$, with entries that were operators on the auxiliary space. When $V_a = \mathbb{C}^2$ and $N = 1$ the two perspectives happen to coincide, see Footnote 4 on p. 10.

The operators on the diagonal in (3.9) give rise to the (twisted) transfer matrix,

$$s = 1/2: \qquad \mathbf{T}(u,\phi) = e^{\mathrm{i}\phi}\mathbf{A}(u) + \mathbf{D}(u). \tag{3.10}$$

The operators in the off-diagonal entries of (3.9) are used for the algebraic Bethe ansatz. Namely, let $|\Omega\rangle = |\uparrow\uparrow\cdots\uparrow\rangle$ be the pseudovacuum state, killed by $\mathbf{C}(u)$. A routine computation using the RLL relations shows that

$$|\{v_m\}_{m=1}^M\rangle = \prod_{m=1}^M \mathbf{B}(v_m)|\Omega\rangle \tag{3.11}$$

is an eigenvector of (3.10) with eigenvalue

$$\begin{aligned}
T(u,\{v_m\}_{m=1}^M,\phi) = {}& e^{\mathrm{i}\phi} \, \sinh^N(u+\eta/2) \prod_{m=1}^M \frac{\sinh(v_m-u+\eta)}{\sinh(v_m-u)} \\
& + \sinh^N(u-\eta/2) \prod_{m=1}^M \frac{\sinh(u-v_m+\eta)}{\sinh(u-v_m)}
\end{aligned} \tag{3.12}$$

provided the parameters $\{v_m\}$ satisfy the Bethe equations (2.7). The energy and momentum in (2.8)–(2.9) follow from (3.8).

**Remark.** When the spectrum of transfer matrices of a system at root of unity contains eigenstates associated with FM strings, the construction (3.11) is *not* enough to obtain all the eigenstates [20, 30, 31]. However, it is still possible to label the eigenstates as $|\{u_j\}_{j=1}^M\rangle$ where $\{u_j\}_{j=1}^M$

---

[6] A more general construction of the R matrix is given in Section 4.3, cf. Eq. (4.23).

are interpreted as the zeroes of the eigenvalue of the Q operator constructed in Section 6.2, see also Sections 5.5 and 6.6. We will use this definition to label all the eigenstates of the Q operator. In the absence of FM strings they can be explicitly constructed via the algebraic Bethe ansatz (3.11); for the case with FM strings see our conjectures in Section 9.

## 3.1 Transfer matrices

By varying the dimension of the auxiliary space one obtains different transfer matrices. In general, for a choice of auxiliary space $V_a$, local Lax operator $\mathbf{L}_{aj}(u)$ on $V_a \otimes \mathbb{C}^2$ and twist operator $\mathbf{E}_a(\phi)$ on $V_a$ one gets a monodromy matrix and corresponding transfer matrix as in (3.5)–(3.6). In (3.9)–(3.10) we encountered the example where $V_a = \mathbb{C}^2$ is a spin-1/2 space, yielding the ('fundamental') $s = 1/2$ transfer matrix of the six-vertex model.

In order to remember which auxiliary space is used in the definition of a particular transfer matrix we will from now on decorate each $\mathbf{T}$ (with e.g. a subscript, superscript, tilde) to keep track of the choice of $V_a$ that was traced over; here we deviate from the usual meaning of subscripts that we used so far. That is, if $V_a$ is some $U_q(\mathfrak{sl}_2)$ irrep $R$ then we will write

$$\mathbf{T}_R(u, \phi) = \mathrm{tr}_a\big[ \mathbf{L}_{aN}(u) \cdots \mathbf{L}_{a2}(u) \mathbf{L}_{a1}(u) \mathbf{E}_a(\phi) \big], \qquad V_a = R. \tag{3.13}$$

For example, from now on $\mathbf{T}_{1/2}$ denotes the fundamental transfer matrix (3.9)–(3.10). We will need the following choices of $R$, whose details can be found in Appendix A.2:

- The unitary spin-$s$ representation with $s \in \frac{1}{2}\mathbb{Z}_{\geq 0}$. Here $V_s = \mathbb{C}^{2s+1}$. We denote the monodromy and transfer matrices by $\mathbf{M}_s$ and $\mathbf{T}_s$. This generalises the case $s = 1/2$ considered so far, and leads to the quasilocal charges mentioned above.

- The highest-weight spin-$s$ representation with 'complex spin' $s \in \mathbb{C}$. We denote its transfer matrix by $\mathbf{T}_s^{\mathrm{hw}}$.

  - For generic $q$ (not at root of unity) this representation is infinite dimensional. One has to take care that the trace in (3.6) makes sense in this case.

  - If $s \in \frac{1}{2}\mathbb{Z}_{\geq 0}$ it can be truncated to a $(2s+1)$-dimensional (sub)representation. The result coincides with the unitary spin-$s$ representation up to a gauge transformation (conjugation). In particular, having taken the trace over the auxiliary space, the truncated transfer matrix is equal to $\mathbf{T}_s$. See Section 5.1.

  - At root of unity both the lowering and raising operators of the complex-spin highest-weight representation become nilpotent, allowing for another truncation to a $\ell_2$-dimensional subspace for any $s \in \mathbb{C}$. We will denote this truncated transfer matrix by $\tilde{\mathbf{T}}_s$. It will be introduced in Sections 6.1–6.2 .

Let us already stress an important difference between the two truncated transfer matrices. Since $\ell_2$ varies wildly as $\eta = i\pi\frac{\ell_1}{\ell_2}$ runs through $i\pi\,\mathbb{Q}$ the spectrum of the truncated transfer matrices $\tilde{\mathbf{T}}_s$, with $\ell_2$-dimensional auxiliary space, will *not* be continuous in $\eta$. This is illustrated in Appendix E.2. Instead, the transfer matrices $\mathbf{T}_s$ for $s \in \frac{1}{2}\mathbb{Z}$ have spectra that vary smoothly with $\eta$.

A generalisation (obtained by fusion in the auxiliary spaces) of the RLL relations (3.3) guarantees that each of the above transfer matrices commute amongst themselves, like in (3.7), as well as with each other (for different $s, s'$). As a result, they share the eigenvectors produced by the algebraic Bethe ansatz (3.11).

The *transfer-matrix fusion relations* are a system of equations that show how the higher-spin transfer matrices $\mathbf{T}_s$ can be constructed from $\mathbf{T}_{1/2}$. This is usually proven by fusion, taking the tensor product of several spin-$\frac{1}{2}$ representations (putting several monodromy matrices on

top of each other) and projecting onto the spin-$s$ submodule, see e.g. [56, 57]. We will get back to the fusion relations in Section 5.3 and 6.4.

## 3.2 Q operators

There have been numerous endeavours to understand the relations between the different transfer matrices and how they can be used to characterise all eigenstates of the XXZ spin chain. The most famous of these was found by Baxter in the 70s [14–16], where he constructed the Q operator that satisfies the *matrix TQ relation* (with respect to $\mathbf{T}_{1/2}$) for the eight-vertex model, which is closely related to the XYZ spin chain. A similar construction can be performed for the six-vertex model and the XXZ spin chain [9]. The eigenvalues of the Q operator are called *Q functions*, and their zeroes are precisely Bethe roots, i.e. solutions to the Bethe equations (2.7), yielding eigenvectors via the Bethe ansatz as in (3.11). Baxter constructed the Q operator by solving the matrix TQ relation directly. For the purpose of numerically obtaining Bethe roots it is much easier to solve the *functional* TQ relation, i.e. the relation between the eigenvalues of $\mathbf{T}_{1/2}$ and the Q operator. We summarise how this works in Appendix C.1. At root of unity, however, the functional TQ relation is not enough to determine the full spectrum.

In the following sections we will construct the Q operator explicitly and prove the matrix TQ relation and the transfer matrix fusion relation. We use a new approach that is based on the factorisation and decomposition of transfer matrix $\mathbf{T}_s^{\text{hw}}$ associated to an auxilary space that is an infinite-dimensional complex-spin representation [40]. This construction works for any anisotropy parameter $\Delta \in \mathbb{R}$. In the case of root of unity we prove truncated fusion relations for the transfer matrices $\tilde{\mathbf{T}}_s$ using the same method. These truncated fusion relations will in turn allow us to prove a conjecture of [32, 34, 41]. This enables us to construct the Q operator explicit at root of unity, and elucidate the exponential degeneracies, which are closely related to the FM string from Section 2.3 but cannot be resolved via the functional TQ relation.

# 4 Factorisation of $\mathbf{T}_s^{\text{hw}}$

We start with an important technical result: the spin $s \in \mathbb{C}$ highest-weight transfer matrix gives rise to a two-parameter transfer matrix that admits a useful factorisation. In this and the following sections we consider arbitrary $q$; we will specialise to root of unity later.

## 4.1 Factorisation of Lax operator

The factorisation of the spin-chain transfer matrix has a long history starting with the chiral Potts model [58, 59]. In particular it was used for the XXX model [60], for the XXZ chain [61] and for the XXZ chain with nonconservative boundary conditions [40]. We follow here the approach of [40].

Before we start to prove the factorisation of the transfer matrix $\mathbf{T}_s^{\text{hw}}$ we need to study the factorisation of the Lax operator when the auxiliary space $V_a$ is an infinite-dimensional complex spin-$s$ representation. Although for $\mathbf{T}_s^{\text{hw}}$ we are interested in the 'half-infinite' highest-weight (Verma) module, see (A.9) in Appendix A.2, our starting point is a bigger space. Although the monodromy matrix can be defined using this internal space, its trace cannot be taken as discussed in Appendix B.1 and needs to be truncated. Let $V_a$ be the infinite-dimensional Hilbert space with orthonormal basis $|n\rangle_a$, $n \in \mathbb{Z}$. It decomposes as a direct sum $V_a = V_a^+ \oplus V_a^-$, where $V_a^+$ is spanned by $|n\rangle_a$ with $n \geq 0$, which is the space that we are after, while $V_a^-$ is the span of $|n\rangle_a$ with $n < 0$. For this auxiliary space we will implicitly assume that the trace in (3.6) is only over $V_a^+$.

Let us write $|n\rangle\langle n'|_a := |n\rangle_{a\,a}\langle n'|$ for the matrix basis. Consider the following operators:

$$\mathbf{W}_a = \sum_{n=-\infty}^{\infty} q^n |n\rangle\langle n|_a, \quad \mathbf{X}_a = \sum_{n=-\infty}^{\infty} |n+1\rangle\langle n|_a. \tag{4.1}$$

If we think of $V_a$ as the sequence space $\ell^2$ by identifying $|n\rangle_a$ with the sequence $\delta_n$ with entries $(\delta_n)_i = \delta_{ni}$ then $\mathbf{X}_a$ is the right shift. The two operators (4.1) form a Weyl pair, $\mathbf{W}_a\mathbf{X}_a = q\,\mathbf{X}_a\mathbf{W}_a$. In other words, $\mathbf{W}_a$ counts the weight and $\mathbf{X}_a$ raises it. We will also use the adjoint $\mathbf{X}_a^\dagger = \sum_n |n\rangle\langle n+1|_a$. On $V_a$ it is the inverse of $\mathbf{X}_a$. In particular $\mathbf{X}_a^\dagger$ commutes with $\mathbf{X}_a$ on $V_a$.

The half-infinite space $V_a^+$ is preserved by both of (4.1), but not by $\mathbf{X}_a^\dagger$. The role of the latter is taken over by the adjoint of the restriction $\mathbf{X}_a|_{V_a^+}$, which we denote by

$$\mathbf{Y}_a = \sum_{n=0}^{\infty} |n\rangle\langle n+1|_a. \tag{4.2}$$

On $V_a^+$ we have $\mathbf{Y}_a\mathbf{X}_a = \mathbb{1}$ while $\mathbf{X}_a\mathbf{Y}_a = \sum_{n>0}|n\rangle\langle n|_a \neq \mathbb{1}$. Together, (4.1) and (4.2) can be used to give a highest-weight representation of $U_q(\mathfrak{sl}_2)$ on $V_a^+$ with spin $s \in \mathbb{C}$:

$$\mathbf{K}_a = q^{-s}\,\mathbf{W}_a = \sum_{n=0}^{\infty} q^{n-s}|n\rangle\langle n|_a,$$

$$\text{on } V_a^+: \quad \mathbf{S}_a^+ = \frac{q^{2s+1}\mathbf{W}_a^{-1} - q^{-2s-1}\mathbf{W}_a}{q - q^{-1}}\mathbf{X}_a = \sum_{n=0}^{\infty} [2s-n]_q |n+1\rangle\langle n|_a, \tag{4.3}$$

$$\mathbf{S}_a^- = \mathbf{Y}_a\frac{\mathbf{W}_a - \mathbf{W}_a^{-1}}{q - q^{-1}} = \sum_{n=0}^{\infty} [n+1]_q |n\rangle\langle n+1|_a.$$

See also Appendix A.2.

Now we turn to the Lax operator $\mathbf{L}_{aj}(u,s)$ associated to a site $j$. Let us introduce two 'spectral parameters' $x$ and $y$ as simple combinations of $u$ and $s$,

$$x := u + \frac{2s+1}{2}\eta, \qquad y := u - \frac{2s+1}{2}\eta, \tag{4.4}$$

that will be convenient when deriving the transfer matrix fusion relations. Starting with the big auxiliary space $V_a$ the Lax operator can be decomposed into a product of operators separating the dependence on these spectral parameters (see also appendix B of Ref. [62]):

$$\mathbf{L}_{aj}(u,s) = \frac{1}{2}\begin{pmatrix} \mathbf{X}_a^\dagger & 0 \\ 0 & 1 \end{pmatrix}_j \mathbf{u}_j(x)\begin{pmatrix} \mathbf{W}_a & 0 \\ 0 & \mathbf{W}_a^{-1} \end{pmatrix}_j \mathbf{v}_j(y)^{\mathrm{T}}\begin{pmatrix} \mathbf{X}_a & 0 \\ 0 & 1 \end{pmatrix}_j, \tag{4.5}$$

where the two by two matrices $\mathbf{u}(x)$ and $\mathbf{v}(y)$ are defined as

$$\mathbf{u}(x) = \begin{pmatrix} 1 & -1 \\ -e^{-x+\eta/2} & e^{x-\eta/2} \end{pmatrix}, \qquad \mathbf{v}(y) = \begin{pmatrix} e^{y-\eta/2} & e^{-y+\eta/2} \\ 1 & 1 \end{pmatrix}. \tag{4.6}$$

The factorisation (4.5) coincides with the one introduced in [61]. To understand (4.5) we compute the product on the right-hand side. The result is

$$\mathbf{L}_{aj}(x,y) = \frac{1}{2}\begin{pmatrix} e^{y+\eta/2}\mathbf{W}_a - e^{-y-\eta/2}\mathbf{W}_a^{-1} & \mathbf{X}_a^\dagger(\mathbf{W}_a - \mathbf{W}_a^{-1}) \\ (e^{x-y}\mathbf{W}_a^{-1} - e^{-x+y}\mathbf{W}_a)\mathbf{X}_a & e^{x-\eta/2}\mathbf{W}_a^{-1} - e^{-x+\eta/2}\mathbf{W}_a \end{pmatrix}_j, \tag{4.7}$$

where we simplified the top-left entry using $\mathbf{W}_a\mathbf{X}_a = q\,\mathbf{X}_a\mathbf{W}_a$ and $\mathbf{X}_a^\dagger\mathbf{X}_a = 1$. Importantly, the four matrix elements preserve the subspace $V_a^+$. This is obvious for all but the top-right entry,

for which the point is that $\mathbf{X}_a^\dagger (\mathbf{W}_a - \mathbf{W}_a^{-1})|n\rangle_a = (q^n - q^{-n})|n-1\rangle_a$ has vanishing prefactor when $n = 0$. Thus the effect of the restriction is to replace $\mathbf{X}_a^\dagger$ by $\mathbf{Y}_a$. In view of (4.3) and $q = e^\eta$ we recognise the entries of (4.7) as

$$e^u \mathbf{K}_a = e^{y+\eta/2} \mathbf{W}_a, \qquad e^{-u} \mathbf{K}_a = e^{-x+\eta/2} \mathbf{W}_a,$$

$$\text{on } V_a^+ : \qquad 2\sinh(\eta) \mathbf{S}_a^+ = (e^{x-y} \mathbf{W}_a^{-1} - e^{-x+y} \mathbf{W}_a) \mathbf{X}_a, \tag{4.8}$$

$$2\sinh(\eta) \mathbf{S}_a^- = \mathbf{Y}_a (\mathbf{W}_a - \mathbf{W}_a^{-1}).$$

This shows that the right-hand side of (4.5) is the same as that in (3.2). The point of this discussion is to show that (4.5) can be used instead of (3.2) when computing the transfer matrix provided that we restrict the trace to $V_a^+$. This will be useful in Section 4.2.

## 4.2 Intertwiners

For convenience let us denote the product on the right-hand side of (4.5) by $\mathbf{L}_{aj}(\mathbf{u}_x, \mathbf{v}_y)$. Exchanging $\mathbf{u}_x$ and $\mathbf{v}_y$ we instead obtain

$$\mathbf{L}_{aj}(\mathbf{v}_y, \mathbf{u}_x) = \frac{1}{2} \begin{pmatrix} e^{y+\eta/2} \mathbf{W}_a - e^{-y-\eta/2} \mathbf{W}_a^{-1} & \mathbf{X}_a^\dagger (e^{x-y} \mathbf{W}_a^{-1} - e^{-x+y} \mathbf{W}_a) \\ (\mathbf{W}_a - \mathbf{W}_a^{-1}) \mathbf{X}_a & e^{x-\eta/2} \mathbf{W}_a^{-1} - e^{-x+\eta/2} \mathbf{W}_a \end{pmatrix}_j \tag{4.9}$$

$$= \mathbf{L}_{aj}(\mathbf{u}_x, \mathbf{v}_y)^{\mathrm{T}}.$$

In the second line the transpose is both in the auxiliary space and in the physical space; note that (on the auxiliary space) $\mathbf{X}_a^{\mathrm{T}} = \mathbf{X}_a^\dagger$ while the diagonal operator $\mathbf{W}_a^{\mathrm{T}} = \mathbf{W}_a$ is symmetric.

This time the matrix elements clearly preserve $V_a^-$, which allows us to consider their action on the quotient $V/V_a^- \cong V_a^+$. Practically this just means that we treat all $|n\rangle_a$ with $n < 0$ as zero. We can thus view (4.9) as acting on $V_a^+$ by substituting $\mathbf{Y}_a$ for $\mathbf{X}_a^\dagger$ in (4.9). Then we construct the monodromy matrix as in (3.5) and finally take the trace over $V_a^+$ as in (3.6) to obtain the transfer matrix. Let us show that the result is the same when we first take the product of (4.9) as in (3.5) and then restrict to $V_a^+$ to take the trace. Let $\mathbf{P}_a^+$ denote the orthogonal projection of $V_a$ onto $V_a^+$. Consider $\mathbf{P}_a^+$ times a product of the matrix elements of (4.9). Since the latter preserve $V_a^-$ we can replace each factor in the product by $\mathbf{P}_a^+$ times that factor without changing the result. Therefore, the restricted trace of such a product in $V_a^+$ amounts to trace the product of the projected matrix elements and the effect is to replace $\mathbf{X}_a^\dagger$ by $\mathbf{Y}_a$ in (4.9).

It is straightforward to intertwine $\mathbf{L}_{aj}(\mathbf{u}_x, \mathbf{v}_y)$ with $\mathbf{L}_{aj}(\mathbf{v}_y, \mathbf{u}_x)$ when both are viewed as acting on $V_a^+$:

$$\mathbf{F}_a(x, y) \mathbf{L}_{aj}(\mathbf{u}_x, \mathbf{v}_y) = \mathbf{L}_{aj}(\mathbf{v}_y, \mathbf{u}_x) \mathbf{F}_a(x, y), \tag{4.10}$$

where the solution for the intertwiner is

$$\mathbf{F}_a(x, y) = \mathbf{F}_a(x - y) = \sum_{n=0}^\infty \begin{bmatrix} (x-y-\eta)/\eta \\ n \end{bmatrix}_q^{-1} |n\rangle\langle n|_a \begin{bmatrix} (x-y-\eta)/\eta \\ n \end{bmatrix}_q$$

$$= \begin{bmatrix} 2s \\ n \end{bmatrix}_q = \prod_{k=1}^n \frac{\sinh[(2s+1-k)\eta]}{\sinh(k\eta)}. \tag{4.11}$$

Note that $\mathbf{F}_a(x, y)$ only depends on $x - y = (2s+1)\eta$, and not on the original spectral parameter $u$. It is well defined for generic values of this quantity, namely for $x - y \notin \eta\mathbb{Z} \oplus 2\pi\mathrm{i}\mathbb{Z}$. To verify that $\mathbf{F}_a(x, y)$ does the job compare (4.7) and (4.9). It is clear that the intertwining relation holds for the entries on the diagonal. For the remaining two entries (which are related by transposition) the relation follows from

$$\begin{bmatrix} 2s \\ n+1 \end{bmatrix}_q [n+1]_q = \begin{bmatrix} 2s \\ n \end{bmatrix}_q [2s-n]_q.$$

Now we introduce a copy $V_b \cong V_a$ with its own spectral parameter $u'$ and spin $s' \in \mathbb{C}$, or equivalently $x', y'$ as in (4.4). Consider the product $\mathbf{L}_{aj}(\mathbf{u}_x, \mathbf{v}_y) \mathbf{L}_{bj}(\mathbf{v}_{y'}, \mathbf{u}_{x'})$. We can construct an intertwiner $\mathbf{G}_{ab}(y, y')$ that interchanges $\mathbf{v}_y$ and $\mathbf{v}_{y'}$ in this product:

$$\mathbf{G}_{ab}(y, y') \mathbf{L}_{aj}(\mathbf{u}_x, \mathbf{v}_y) \mathbf{L}_{bj}(\mathbf{v}_{y'}, \mathbf{u}_{x'}) = \mathbf{L}_{aj}(\mathbf{u}_x, \mathbf{v}_{y'}) \mathbf{L}_{bj}(\mathbf{v}_y, \mathbf{u}_{x'}) \mathbf{G}_{ab}(y, y'). \tag{4.12}$$

To solve this we first consider the big space $V_a \otimes V_b$ and write the Lax operators as products like in (4.5). We look for $\mathbf{G}_{ab}(y, y') = g_{y,y'}(\mathbf{X}_a \mathbf{X}_b^\dagger)$ in the form of a power series $g_{y,y'}$ in $\mathbf{Z}_{ab} := \mathbf{X}_a \mathbf{X}_b^\dagger$. This commutes with $\mathbf{X}_a^\dagger$ on the left and with $\mathbf{X}_b$ on the right. Further using $\mathbf{Z}_{ab} \mathbf{W}_a^{\pm 1} = q^{\mp 1} \mathbf{W}_a^{\pm 1} \mathbf{Z}_{ab}$ and $\mathbf{W}_b^{\pm 1} \mathbf{Z}_{ab} = q^{\mp 1} \mathbf{Z}_{ab} \mathbf{W}_b^{\pm 1}$ (4.12) reduces to

$$\begin{pmatrix} g_{y,y'}(q^{-1}\mathbf{Z}_{ab}) & 0 \\ 0 & g_{y,y'}(q\,\mathbf{Z}_{ab}) \end{pmatrix}_j \mathbf{v}_j(y)^{\mathrm{T}} \begin{pmatrix} \mathbf{Z}_{ab} & 0 \\ 0 & 1 \end{pmatrix}_j \mathbf{v}_j(y')$$
$$= \mathbf{v}_j(y')^{\mathrm{T}} \begin{pmatrix} \mathbf{Z}_{ab} & 0 \\ 0 & 1 \end{pmatrix}_j \mathbf{v}_j(y) \begin{pmatrix} g_{y,y'}(q^{-1}\mathbf{Z}_{ab}) & 0 \\ 0 & g_{y,y'}(q\,\mathbf{Z}_{ab}) \end{pmatrix}_j. \tag{4.13}$$

Using (4.6) this reduces to the functional equation

$$g_{y,y'}(q z)(1 + z\, e^{-y+y'}) = g_{y,y'}(q^{-1}z)(1 + z\, e^{y-y'}), \tag{4.14}$$

which is solved by

$$g_{y,y'}(z) = \sum_{n=0}^{\infty} \begin{bmatrix} (y - y')/\eta \\ n \end{bmatrix}_q z^n, \qquad \begin{bmatrix} (y - y')/\eta \\ n \end{bmatrix}_q = \prod_{k=1}^{n} \frac{\sinh[y - y' - (k-1)\eta]}{\sinh(k\eta)}. \tag{4.15}$$

Since $\mathbf{X}_a \mathbf{X}_b^\dagger$ preserves the subspace $V_a^+ \otimes V_b^-$ the intertwiner does so too. Restricting to $V_a^+$ and passing to the quotient $V_b/V_b^- \cong V_b^+$, we replace $\mathbf{X}_b^\dagger$ by $\mathbf{Y}_b$. Then the relation (4.12) is obeyed on $V_a^+ \otimes V_b^+$ with

$$\mathbf{G}_{ab}(y, y') = \sum_{n=0}^{\infty} \begin{bmatrix} (y - y')/\eta \\ n \end{bmatrix}_q (\mathbf{X}_a \mathbf{Y}_b)^n. \tag{4.16}$$

Similarly, we define an intertwiner $\mathbf{H}_{ab}(x, x')$ which interchanges $\mathbf{u}_x$ and $\mathbf{u}_{x'}$:

$$\mathbf{H}_{ab}(x, x') \mathbf{L}_{aj}(\mathbf{v}_y, \mathbf{u}_x) \mathbf{L}_{bj}(\mathbf{u}_{x'}, \mathbf{v}_{y'}) = \mathbf{L}_{aj}(\mathbf{v}_y, \mathbf{u}_{x'}) \mathbf{L}_{bj}(\mathbf{u}_x, \mathbf{v}_{y'}) \mathbf{H}_{ab}(x, x'). \tag{4.17}$$

This time the roles of $V_a$ and $V_b$ are reversed and we seek a power series in $\mathbf{X}_a^\dagger \mathbf{X}_b$. Proceeding as before, now taking the quotient $V_a/V_a^- \cong V_a^+$ and restricting $V_b$ to $V_b^+$, we find

$$\mathbf{H}_{ab}(x, x') = \sum_{n=0}^{\infty} \begin{bmatrix} (x - x')/\eta \\ n \end{bmatrix}_q (\mathbf{Y}_a \mathbf{X}_b)^n. \tag{4.18}$$

### 4.3 Two-parameter transfer matrix

Now we reparametrise the transfer matrix with infinite-dimensional complex spin-$s$ auxiliary space as a *two*-parameter transfer matrix:

$$\mathbb{T}(x, y, \phi) := \mathbf{T}_s^{\mathrm{hw}}(u, \phi). \tag{4.19}$$

The important parameters are $x, y$ and $u, s$, related by (4.4); the twist $\phi$ is just a spectator. Due to the existence of the intertwiners this two-parameter transfer matrix satisfies

$$\mathbb{T}(x, y, \phi) \mathbb{T}(x', y', \phi) = \mathbb{T}(x', y, \phi) \mathbb{T}(x, y', \phi) = \mathbb{T}(x, y', \phi) \mathbb{T}(x', y, \phi), \tag{4.20}$$

and in particular forms a family of commuting operators,

$$\mathbb{T}(x,y,\phi)\,\mathbb{T}(x',y',\phi) = \mathbb{T}(x',y',\phi)\,\mathbb{T}(x,y,\phi). \tag{4.21}$$

The proofs of (4.20) are routine, using a variation of the argument that establishes (3.7). Since it might nevertheless be instructive to see it in the present setting let us show the first equality in (4.20). We start with site $j$, for which the preceding implies

$$\mathbf{F}_a(x',y)^{-1}\,\mathbf{H}_{ab}(x,x')\,\mathbf{F}_a(x,y)\,\mathbf{L}_{aj}(\mathbf{u}_x,\mathbf{v}_y)\,\mathbf{L}_{bj}(\mathbf{u}_{x'},\mathbf{v}_{y'})$$
$$= \mathbf{L}_{aj}(\mathbf{u}_{x'},\mathbf{v}_y)\,\mathbf{L}_{bj}(\mathbf{u}_x,\mathbf{v}_{y'})\,\mathbf{F}_a(x',y)^{-1}\,\mathbf{H}_{ab}(x,x')\,\mathbf{F}_a(x,y).$$

By the 'train argument' this readily extends to the two-parameter monodromy matrix:

$$\mathbf{F}_a(x',y)^{-1}\,\mathbf{H}_{ab}(x,x')\,\mathbf{F}_a(x,y)\,\mathbb{M}_a(x,y)\,\mathbb{M}_b(x',y')$$
$$= \mathbb{M}_a(x',y)\,\mathbb{M}_b(x,y')\,\mathbf{F}_a(x',y)^{-1}\,\mathbf{H}_{ab}(x,x')\,\mathbf{F}_a(x,y). \tag{4.22}$$

Now assume that $\mathbf{H}_{ab}(x,x')$ is invertible; this is true so long as $x - x' \notin -\eta\,\mathbb{Z}_{\geq 0} \oplus 2\pi\mathrm{i}\,\mathbb{Z}$. We multiply from the left by $\mathbf{F}_a(x,y)^{-1}\,\mathbf{H}_{ab}(x,x')^{-1}\,\mathbf{F}_a(x',y)$ and take the trace over $V_a^+ \otimes V_b^+$. By the cyclicity of the trace the conjugation by $\mathbf{F}_a(x',y)\,\mathbf{H}_{ab}(x,x')^{-1}\,\mathbf{F}_a(x,y)^{-1}$ drops out on the right-hand side, and we arrive at the desired equality.

More precisely, $\mathbf{F}_a$ is well defined for $s \notin \frac{1}{2}\mathbb{Z}_{\geq 0} \oplus 2\pi\mathrm{i}\,\mathbb{Z}$, and we furthermore need $x - x' \notin -\eta\,\mathbb{Z}_{\geq 0} \oplus 2\pi\mathrm{i}\,\mathbb{Z}$ to ensure that $\mathbf{H}_{ab}$ is invertible. Thus the preceding argument establishes the first equality in (4.20) only for almost all values of $x, y$. However, the Lax operator, and therefore the two-parameter transfer matrix, are continuous in these two spectral parameters. Thus the conclusion holds in full generality by continuity. (The situation is analogous in the standard proof of commutativity of ordinary transfer matrices from the RLL relations; there the R matrix is only invertible for almost all values of the spectral parameter.)

The exchange of $y$ and $y'$ is shown analogously. Let us note that, together, the intertwiners give rise to an R matrix

$$\mathbf{R}_{ab}(x,y;x',y') := \mathbf{P}_{ab}\,\mathbf{F}_b(x,y)^{-1}\,\mathbf{G}_{ab}(y,y')\,\mathbf{F}_b(x,y') \times \mathbf{F}_a(x',y)^{-1}\,\mathbf{H}_{ab}(x,x')\,\mathbf{F}_a(x,y), \tag{4.23}$$

where $\mathbf{P}_{ab}$ is the permutation operator between the auxiliary spaces $V_a^+$ and $V_b^+$. Using the properties of the intertwiners one can show that this is the R matrix for which the Lax operator (4.5) satisfies the RLL relation,

$$\mathbf{R}_{ab}(x,y;x',y')\,\mathbf{L}_{aj}(\mathbf{u}_x,\mathbf{v}_y)\,\mathbf{L}_{bj}(\mathbf{u}_{x'},\mathbf{v}_{y'}) = \mathbf{L}_{bj}(\mathbf{u}_{x'},\mathbf{v}_{y'})\,\mathbf{L}_{aj}(\mathbf{u}_x,\mathbf{v}_y)\,\mathbf{R}_{ab}(x,y;x',y'). \tag{4.24}$$

Via the train argument this gives a direct proof of the commutativity (4.21). A more detailed investigation of the properties of this R matrix is beyond the scope of the present paper.

## 4.4 Factorisation of two-parameter transfer matrix

The property (4.20) implies that the two-parameter transfer matrix $\mathbb{T}(x,y,\phi)$ can be factorised into two parts that only depend on the spectral parameter $x$ or $y$, respectively. Namely,

$$\mathbb{T}(x,y,\phi) = \mathbf{Q}_{y_0}(x,\phi)\,\mathbf{P}_{x_0,y_0}(y,\phi), \tag{4.25}$$

where

$$\mathbf{Q}_{y_0}(x,\phi) := \mathbb{T}(x,y_0,\phi), \qquad \mathbf{P}_{x_0,y_0}(y,\phi) := \mathbb{T}(x_0,y,\phi)\,\mathbb{T}(x_0,y_0,\phi)^{-1}, \tag{4.26}$$

provided that $\mathbb{T}(x_0, y_0, \phi)$ is invertible. We will return to this invertibility (for the truncated case) in Section 7.4. It further allows us to change the value of $y_0$ at will: note that

$$\mathbf{Q}_{y_1}(x, \phi) = \mathbf{Q}_{y_0}(x, \phi)\, \mathbb{T}(x_0, y_1, \phi)\, \mathbb{T}(x_0, y_0, \phi)^{-1}. \tag{4.27}$$

Let us therefore suppress the dependence on $y_0$. A similar argument applies to $\mathbf{P}_{x_0, y_0}(y, \phi)$, whose dependence on $x_0, y_0$ will from now on be suppressed too.

According to (4.20) the operators (4.26) commute with themselves (at different values of the spectral parameter) and with each other:

$$\begin{aligned}
\left[\mathbf{Q}(x, \phi), \mathbf{Q}(x', \phi)\right] &= 0, & x, x' &\in \mathbb{C}, \\
\left[\mathbf{Q}(x, \phi), \mathbf{P}(y, \phi)\right] &= 0, & x, y &\in \mathbb{C}, \\
\left[\mathbf{P}(y, \phi), \mathbf{P}(y', \phi)\right] &= 0, & y, y' &\in \mathbb{C}.
\end{aligned} \tag{4.28}$$

# 5 Matrix TQ relation and transfer matrix fusion relation

It is time to harvest the fruits of our labour. We will show that $\mathbf{Q}$ from (4.26) is nothing but Baxter's Q operator, satisfying the matrix TQ relation with twist $\phi$, see (5.10). Moreover, $\mathbf{P}$ obeys a very similar matrix 'TP relation', see (5.11). In particular, in the periodic case ($\phi = 0$) $\mathbf{Q}, \mathbf{P}$ are 2 linearly independent solutions of the matrix TQ relation. We will furthermore derive the transfer matrix fusion relations as well as an interpolation formula that expresses the half-integer spin transfer matrix in terms of Q operators. As in the previous section $q$ is arbitrary.

## 5.1 Decomposition of highest-weight transfer matrix

In order to demonstrate that the Q operator in (4.26) is indeed the same as Baxter's Q operator, satisfying matrix TQ relation, we shall use a decomposition [40, 60, 63–66] of the two-parameter transfer matrix $\mathbb{T}(x, y, \phi)$, when specialising the complex spin to $2s \in \mathbb{Z}_{\geq 0}$. Recall from Section 4.1 that the auxiliary space is spanned by $|n\rangle$ for $n \geq 0$; this is what we denoted by $V_a^+$ in Section 4.1. When $2s \in \mathbb{Z}_{\geq 0}$ we can decompose it as $V_{a'}^+ \oplus V_{a''}$, where $V_{a'}^+$ is the span of all $|n\rangle$ with $n > 2s$, while the finite-dimensional piece $V_{a''}$ is spanned by $|2s\rangle, \cdots, |1\rangle, |0\rangle$. The latter is certainly preserved by the diagonal operator $\mathbf{K}_a$ as well as by $\mathbf{S}_a^-$, whose block-triangular form is shown in Fig. 2. Since $2s \in \mathbb{Z}_{\geq 0}$ the operator $\mathbf{S}_a^+$ preserves $V_{a''}$ too. Indeed, as $[2s-n]_q = 0$ for $n = 2s$ the coefficient of $|2s+1\rangle\langle 2s|_a$ in $\mathbf{S}_a^+$ vanishes: this entry is marked in red in Fig. 2. That is, all of $\mathbf{K}_a, \mathbf{S}_a^\pm$ are of block lower triangular form. The $(2s+1)\times(2s+1)$ blocks that act on $V_{a''}$ differ from the unitary spin-$s$ representation by a simple gauge transformation (conjugation).

Now we go towards the transfer matrix. Since the Lax operator is built from $\mathbf{K}_a, \mathbf{S}_a^\pm$, see (3.2), for $2s \in \mathbb{Z}_{\geq 0}$ it assumes a block lower triangular form with respect to the decomposition $V_{a'}^+ \oplus V_{a''}$ too. Let us indicate its block structure, paralleling that in Fig. 2, by

$$\mathbf{L}_{aj} = \begin{pmatrix} \mathbf{L}_{a'j} & 0 \\ \mathbf{L}_{a'''j} & \mathbf{L}_{a''j} \end{pmatrix}. \tag{5.1}$$

Here we can think of $\mathbf{L}_{a'j}$ as a square infinite matrix acting on $V_{a'}^+$, $\mathbf{L}_{a''j}$ as a square matrix on $V_{a''}$, and (for want of a better notation) $\mathbf{L}_{a'''j}$ as a rectangular matrix sending $V_{a'}^+$ to $V_{a''}$; all with entries that are operators acting at site $j$. For the monodromy matrix we consider more sites. Note that the blocks on the diagonal only 'talk' amongst themselves:

$$\mathbf{L}_{aj}\mathbf{L}_{ak} = \begin{pmatrix} \mathbf{L}_{a'j}\mathbf{L}_{a'k} & 0 \\ \mathbf{L}_{a'''j}\mathbf{L}_{a'k} + \mathbf{L}_{a''j}\mathbf{L}_{a'''k} & \mathbf{L}_{a''j}\mathbf{L}_{a''k} \end{pmatrix}. \tag{5.2}$$

$$
\left(\begin{matrix} \ddots & & & & & \vdots \\ & 0 & 0 & 0 & 0 & 0 \\ & * & 0 & 0 & 0 & 0 \\ & 0 & * & 0 & 0 & 0 \\ & 0 & 0 & * & 0 & 0 \\ \cdots & 0 & 0 & 0 & * & 0 \end{matrix}\right) \updownarrow \; 2s+1
\qquad
\left(\begin{matrix} \ddots & & & & & \vdots \\ & 0 & * & 0 & 0 & 0 \\ & 0 & 0 & * & 0 & 0 \\ & 0 & 0 & 0 & 0 & 0 \\ & 0 & 0 & 0 & 0 & * \\ \cdots & 0 & 0 & 0 & 0 & 0 \end{matrix}\right) \updownarrow \; 2s+1
$$

Figure 2: The decomposition of $\mathbf{S}_s^-$ (left) and $\mathbf{S}_s^+$ (right) in the infinite-dimensional highest-weight auxiliary space $V_{a'}^+ \oplus V_{a''}$ for $s \in \frac{1}{2}\mathbb{Z}_{\geq 0}$. The $*$ represent non-zero entries. The square orange (pink) block acts on $V_{a'}^+$ (resp. $V_{a''}$), while the rectangular blue block maps $V_{a'}^+$ to $V_{a''}$. Note that we order the basis decreasingly, $\cdots, |1\rangle, |0\rangle$, cf. Appendix A.2.

Thus the monodromy matrix inherits the block triangular form

$$
\mathbf{M}_a = \begin{pmatrix} \mathbf{M}_{a'} & 0 \\ \mathbf{M}_{a'''} & \mathbf{M}_{a''} \end{pmatrix}. \tag{5.3}
$$

By taking the trace we obtain the transfer matrix

$$
\mathbf{T}_s^{\mathrm{hw}} = \mathrm{tr}_a \, \mathbf{M}_a = \mathrm{tr}_{a'} \, \mathbf{M}_{a'} + \mathrm{tr}_{a''} \, \mathbf{M}_{a''}. \tag{5.4}
$$

Since the unitary spin-$s$ representation differs from that on $V_{a''}$ by a gauge transformation, $\mathrm{tr}_{a''} \mathbf{M}_{a''}$ is nothing but the transfer matrix $\mathbf{T}_s$ for spin $s \in \frac{1}{2}\mathbb{Z}_{\geq 0}$. As $V_{a'}^+ \cong V_a^+$ moreover $\mathrm{tr}_{a'} \mathbf{M}_{a'}$ is another complex-spin highest-weight transfer matrix! Accounting for the twist and the correct value of the new complex spin we arrive at the decomposition

$$
\mathbf{T}_s^{\mathrm{hw}}(u,\phi) = e^{\mathrm{i}(2s+1)\phi} \, \mathbf{T}_{-s-1}^{\mathrm{hw}}(u,\phi) + \mathbf{T}_s(u,\phi). \tag{5.5}
$$

## 5.2 Generalised Wronskian and matrix TQ relation

By (4.4) and (4.19) we can rewrite (4.25) as

$$
\mathbf{T}_s^{\mathrm{hw}}(u,\phi) = \mathbf{Q}\Big(u + \frac{2s+1}{2}\eta, \phi\Big) \mathbf{P}\Big(u - \frac{2s+1}{2}\eta, \phi\Big). \tag{5.6}
$$

In these terms the decomposition (5.5) reads

$$
\begin{aligned}
\mathbf{T}_s(u,\phi) = {}& \mathbf{Q}\Big(u + \frac{2s+1}{2}\eta, \phi\Big) \mathbf{P}\Big(u - \frac{2s+1}{2}\eta, \phi\Big) \\
& - e^{(2s+1)\mathrm{i}\phi} \, \mathbf{Q}\Big(u - \frac{2s+1}{2}\eta, \phi\Big) \mathbf{P}\Big(u + \frac{2s+1}{2}\eta, \phi\Big).
\end{aligned} \tag{5.7}
$$

This is the *generalised Wronskian relation*. Let us consider some examples.

For $s = 0$ the operator $\mathbf{T}_0(u,\phi)$ is a scalar, which is independent of the twist according to our choice of the latter (see Appendix B.1). Thus it can be denoted as $T_0(u) = \sinh^N(u)$. We therefore obtain the Wronskian relation

$$
T_0(u) = \mathbf{Q}\Big(u + \frac{\eta}{2}, \phi\Big) \mathbf{P}\Big(u - \frac{\eta}{2}, \phi\Big) - e^{\mathrm{i}\phi} \, \mathbf{Q}\Big(u - \frac{\eta}{2}, \phi\Big) \mathbf{P}\Big(u + \frac{\eta}{2}, \phi\Big). \tag{5.8}
$$

Note that $T_0(u)$ is independent of the twist $\phi$ while $\mathbf{Q}$ and $\mathbf{P}$ in the right-hand side do depend on $\phi$.

When $s = 1/2$ we obtain a relation for the fundamental (six-vertex) transfer matrix:

$$\mathbf{T}_{1/2}(u, \phi) = \mathbf{Q}(u + \eta, \phi)\mathbf{P}(u - \eta, \phi) - e^{2i\phi}\,\mathbf{Q}(u - \eta, \phi)\mathbf{P}(u + \eta, \phi). \tag{5.9}$$

Multiplying both sides by $\mathbf{Q}(u, \phi)$ and using (5.8) to get rid of the P operator we find

$$\mathbf{T}_{1/2}(u, \phi)\mathbf{Q}(u, \phi) = T_0(u - \eta/2)\mathbf{Q}(u + \eta, \phi) + e^{i\phi}\,T_0(u + \eta/2)\mathbf{Q}(u - \eta, \phi). \tag{5.10}$$

We have recovered Baxter's matrix TQ relation [9]!

If we instead multiply by $\mathbf{P}(u, \phi)$ and use (5.8) to eliminate the Q operator we analogously obtain a matrix 'TP relation'

$$\mathbf{T}_{1/2}(u, \phi)\mathbf{P}(u, \phi) = e^{i\phi}\,T_0(u - \eta/2)\mathbf{P}(u + \eta, \phi) + T_0(u + \eta/2)\mathbf{P}(u - \eta, \phi). \tag{5.11}$$

Note the different positions at which the twist $e^{i\phi}$ appears in (5.10) and (5.11). More precisely, the TP relation for the rescaled operator $e^{i\phi u/\eta}\,\mathbf{P}(u, \phi)$ is nothing but the TQ relation with opposite twist $-\phi$. In the periodic case, $\phi = 0$, the Q and P operators are two solutions to the matrix TQ relation (5.10) that are linearly independent as their Wronskian, the right-hand side of (5.8), is nonzero ($T_0 \neq 0$).

## 5.3 Transfer matrix fusion relations

Interestingly, the decomposition of the two-parameter transfer matrix $\mathbb{T}(x, y, \phi)$ further allows us to derive the transfer matrix fusion relations [56, 57, 67, 68]. As the name suggests these are typically derived by fusion in the auxiliary space (tensoring auxiliary spaces and projecting onto any irrep via suitably related spectral parameters). In the present setup the simple derivation goes as follows.[7] We multiply both sides of (5.7) from the left by $\mathbf{T}_{1/2}(u \pm \frac{2s+1}{2}\eta, \phi)$ and apply (5.10)–(5.11). Collecting terms with the same $T_0$ and using (5.7) for each we obtain

$$\begin{aligned}
\mathbf{T}_{1/2}\Big(u \pm \frac{2s+1}{2}\eta, \phi\Big)\mathbf{T}_s(u, \phi) &= e^{i\phi}\,T_0\big(u \pm (s+1)\eta\big)\mathbf{T}_{s-1/2}\Big(u \mp \frac{\eta}{2}, \phi\Big) \\
&+ T_0\big(u \pm s\,\eta\big)\mathbf{T}_{s+1/2}\Big(u \pm \frac{\eta}{2}, \phi\Big).
\end{aligned} \tag{5.12}$$

These two equations, one for the upper signs and one for the lower signs, are the transfer matrix fusion relations. Taken together for all half-integer values of the spin $s$ the functional form of these relations, i.e. the analogous relations for the eigenvalues $T_s$, comprises a system of difference relations called a *T-system*. Together with a reformulation known as the *Y-system* it is of vital significance for physical applications like the thermodynamic Bethe ansatz. See Ref. [69] for a thorough review and further references.

## 5.4 Interpolation formula

When $s \in \frac{1}{2}\mathbb{Z}_{\geq 0}$ the transfer matrix $\mathbf{T}_s$ can be expressed in terms of $\mathbf{Q}$; let us derive this in our framework. Rewrite (5.7) as

$$\frac{\mathbf{T}_s(u)}{\mathbf{Q}\big(u + \frac{2s+1}{2}\eta\big)\mathbf{Q}\big(u - \frac{2s+1}{2}\eta\big)} = \frac{\mathbf{P}\big(u - \frac{2s+1}{2}\eta\big)}{\mathbf{Q}\big(u - \frac{2s+1}{2}\eta\big)} - e^{(2s+1)i\phi}\,\frac{\mathbf{P}\big(u + \frac{2s+1}{2}\eta\big)}{\mathbf{Q}\big(u + \frac{2s+1}{2}\eta\big)}. \tag{5.13}$$

The meaning of the fractions for multiplication with the inverse is unambiguous since the operators involved all commute. Specialising to $s = 0$ we likewise rewrite (5.8) as

$$\frac{T_0(u)}{\mathbf{Q}(u + \eta/2)\mathbf{Q}(u - \eta/2)} = \frac{\mathbf{P}(u - \eta/2)}{\mathbf{Q}(u - \eta/2)} - e^{i\phi}\,\frac{\mathbf{P}(u + \eta/2)}{\mathbf{Q}(u + \eta/2)}. \tag{5.14}$$

---

[7] The reader might find it convenient to abbreviate $f^{(\alpha)} := f(u + \alpha\eta)$ in these computations.

We translate the arguments of (5.14) by $k\eta$, multiply by $e^{ik\phi}$ and sum over $k$ from 0 to $2s$. The telescoping sum on the right-hand side yields the right-hand side of (5.13). In this way we obtain the interpolation-type formula [56, 57]

$$
\begin{aligned}
\mathbf{T}_s(u) = {} & \mathbf{Q}\left(u + \frac{2s+1}{2}\eta\right)\mathbf{Q}\left(u - \frac{2s+1}{2}\eta\right) \\
& \times \sum_{k=0}^{2s} e^{ik\phi}\,\frac{T_0\big(u + (k-s)\,\eta\big)}{\mathbf{Q}\big(u + (k-s+1/2)\eta\big)\,\mathbf{Q}\big(u + (k-s-1/2)\eta\big)}\,.
\end{aligned}
\tag{5.15}
$$

## 5.5 Structure of the eigenvalues of Q and P

Let us now study the properties of the Q and P operators, and their eigenvalues, on their respective spectral parameters. It is convenient to use multiplicative spectral parameters $r := e^x$ and $t := e^y$. As always we start from the Lax operator, which we here write as

$$
\mathbf{L}_{aj}(x,y) = \begin{pmatrix} \mathbf{L}_a^{11}(x,y) & \mathbf{L}_a^{12}(x,y) \\ \mathbf{L}_a^{21}(x,y) & \mathbf{L}_a^{22}(x,y) \end{pmatrix}_j .
\tag{5.16}
$$

We start with the dependence on $t$. Let us say that a Laurent polynomial $f(t)$ is a 'trigonometric polynomial of degree $n$' if $t^n f(t)$ is a polynomial in $t^2$ of degree $n$. From (4.7) we see that $\mathbf{L}_a^{11}, \mathbf{L}_a^{21}$ are trigonometric polynomials of degree one in $t$ while $\mathbf{L}_a^{12}, \mathbf{L}_a^{22}$ are independent of $t$. It follows that on a vector with $S^z = N/2 - M$ the two-parameter monodromy matrix $\mathbb{M}_a(x,y,\phi)$ acts by a matrix whose entries are trigonometric polynomials of degree $N - M$ in $t$. Thus the same holds for the two-parameter transfer matrix $\mathbb{T}(x,y,\phi)$. Regarding $r$ we see that $\mathbf{L}_a^{11}, \mathbf{L}_a^{12}$ are both degree zero while $\mathbf{L}_a^{21}, \mathbf{L}_a^{22}$ are trigonometric polynomials in $r$ of degree one. This means that when acting to the *left* on (the dual of) the $M$-particle subspace, $\mathbb{M}_a(x,y)$ has entries that are trigonometric polynomials in $r$ of degree $M$. As before this carries over to $\mathbb{T}(x,y,\phi)$. Since the latter moreover preserves the value of $M$ this remains true when acting to the *right* as well. The upshot is the operator $\mathbb{T}(x,y,\phi)$ acts on the $M$-particle subspace, with $S^z = N/2 - M$ fixed, by a matrix with entries that are trigonometric polynomials of degree $M$ in $r$ and degree $N - M$ in $t$. In terms of the Q and P operators the conclusion is that, on this subspace, $\mathbf{Q}(x,\phi)$ acts by a matrix consisting of trigonometric polynomials in $r$ of degree $M$, and $\mathbf{P}(y,\phi)$ likewise by a matrix of trigonometric polynomials in $t$ of degree $N - M$. These properties are inherited by the eigenvalues since the (joint) eigenvectors are independent of $r, t$ in view of the commutativity (4.28).

Let us make this a little more concrete. By the commutativity of all transfer matrices the joint eigenvalues of the Q and P operator are given by the algebraic Bethe ansatz. For any on-shell Bethe state, i.e. (3.11) — see also the remark following that equation — subject to the Bethe equations (2.7), we have

$$
\mathbf{Q}(u,\phi)\,|\{u_m\}_{m=1}^M\rangle = Q(u, \{u_m\}_{m=1}^M, \phi)\,|\{u_m\}_{m=1}^M\rangle\,.
\tag{5.17}
$$

Baxter realised that the TQ equation determines the eigenvalues $Q$ in familiar terms. Namely, by the above it is a trigonometric polynomial of degree $M$ in $t := e^u$. Let us denote the zeroes by $t_m$:

$$
Q(u,\phi) = \text{cst} \times \prod_{m=1}^{M}\left(t_m^{-1} t - t_m t^{-1}\right).
\tag{5.18}
$$

According to the matrix TQ relation (5.10) the eigenvalues obey the functional TQ relation

$$
T_{1/2}(u,\phi)Q(u,\phi) = T_0(u - \eta/2)Q(u + \eta, \phi) + e^{i\phi}\,T_0(u + \eta/2)Q(u - \eta, \phi).
\tag{5.19}
$$

Picking $u$ so that $t = t_m$ is a zero of $Q$, the left-hand side of (5.19) vanishes. Recalling that $T_0(u) = \sinh^N(u)$ the result reduces to the Bethe equations (2.7) in multiplicative form, allowing us to identify $t_m = e^{u_m}$ as the multiplicative version of the Bethe roots. That is, the zeroes of the eigenvalues of the Q operator are precisely the Bethe roots [9].

This observation is used in the numerical recipe for finding the Bethe roots in Appendix C.1. In the presence of any Bethe root at infinity, which does occur for XXZ as reviewed in Section 2.2, the form of the eigenvalues has to be modified a little. Indeed,

$$
\begin{aligned}
u_m \to +\infty : &\qquad t_m \to \infty\,, &\qquad t_m^{-1}\,t - t_m\,t^{-1} \to t^{-1}\,, \\
u_m \to -\infty : &\qquad t_m \to 0\,, &\qquad t_m^{-1}\,t - t_m\,t^{-1} \to t\,.
\end{aligned}
\tag{5.20}
$$

Therefore, Bethe roots at infinity show up in the eigenvalues of the Q operator: if we rearrange the $u_j$ so that the infinite roots are last then

$$
Q(u,\phi) = \mathrm{cst} \times t^{n_{-\infty}-n_{+\infty}} \times \prod_{m=1}^{M-n_{-\infty}-n_{+\infty}} \left( t_m^{-1}\,t - t_m\,t^{-1} \right).
\tag{5.21}
$$

One can similarly show that the eigenvalues of the P operator $\mathbf{P}$ are of the form (5.21) but with $M$ replaced by $N - M$. The functional version of the TP relations (5.11) gives rise to Bethe-type equations for the $N - M$ zeroes of the eigenvalues $P(v, \phi)$ for an $M$-particle eigenvector:

$$
\left( \frac{\sinh(v_n + \eta/2)}{\sinh(v_n - \eta/2)} \right)^N \prod_{n'(\neq n)}^{N-M} \frac{\sinh(v_n - v_{n'} - \eta)}{\sinh(v_n - v_{n'} + \eta)} = e^{\mathrm{i}\phi}\,.
\tag{5.22}
$$

These are precisely the Bethe equations for a Bethe state $|\{v_n\}_{n=1}^{N-M}\rangle$ of the XXZ model with opposite twist $-\phi$. Note that the algebraic Bethe ansatz (3.11) only uses the pseudovacuum $|\uparrow \cdots \uparrow\rangle$ and the B operator $\mathbf{B}(u)$, which is independent of the twist $\phi$, cf. (3.9). Therefore, the off-shell Bethe state $|\{v_m\}_{m=1}^{N-M}\rangle$ does not depend on the twist either; the latter only enters on shell, i.e. upon imposing the Bethe equations. When we impose (5.22) the Bethe vector $|\{v_n\}_{n=1}^{N-M}\rangle$ is not an eigenstate of $\mathbf{T}_s(u, \phi)$, but rather of $\mathbf{T}_s(u, -\phi)$. In particular, in the periodic case ($\phi = 0$) it can be interpreted as a Bethe vector beyond the equator (if $M < N/2$, so that $N - M > N/2$). A detailed calculation is presented in Appendix C.3.

## 6 Truncated transfer matrix at root of unity

Now we specialise the anisotropy parameter to a root of unity (2.3). Here the infinite-dimensional auxiliary space $V_a^+$ has a finite-dimensional subspace $\tilde{V}_a$, whose size depends on the root of unity, and which is preserved by $U_q(\mathfrak{sl}_2)$. Truncating to $\tilde{V}_a$ allows for another decomposition of the two-parameter transfer matrix that leads to a proof of a conjecture of [32, 34, 41] and to truncated fusion relations.

Importantly, the truncation enables us to construct the Q operators explicitly, only using finite-dimensional matrices at all intermediate steps. In practice we can do this for all eigenvectors a spin chain with $N \leq 16$ and thus obtain the full spectrum of the XXZ spin chain with arbitrary twist $\phi$, revealing the conditions for the appearance of exponential degeneracies that have been observed before [20]. As we will see later this has significant consequences for the thermodynamic limit.

$$
\left(
\begin{array}{ccccc}
\ddots & & & \vdots & \\
 0 & 0 & 0 & 0 & 0 \\
 * & 0 & 0 & 0 & 0 \\
 0 & {\color{red}0} & 0 & 0 & 0 \\
 0 & 0 & * & 0 & 0 \\
 \cdots \; 0 & 0 & 0 & * & 0
\end{array}
\right) \updownarrow \ell_2
\qquad
\left(
\begin{array}{ccccc}
\ddots & & & \vdots & \\
 0 & * & 0 & 0 & 0 \\
 0 & 0 & * & 0 & 0 \\
 0 & 0 & 0 & * & 0 \\
 0 & 0 & 0 & 0 & * \\
 \cdots \; 0 & 0 & 0 & 0 & 0
\end{array}
\right) \updownarrow \ell_2
$$

Figure 3: The decomposition of $\mathbf{S}_a^-$ (left) and $\mathbf{S}_a^+$ (right) at root of unity with $\ell_2 = 3$, where $*$ represents non-zero elements of the matrices. As in Fig. 2 the bottom-right entry corresponds to $|0\rangle\langle 0|$.

## 6.1 Truncation and intertwiners at root of unity

At root of unity $\eta = i\pi\frac{\ell_1}{\ell_2}$ the matrix elements of the Lax operator (4.7) in the auxiliary space acquire the periodicity:

$$
{}_a\langle k+\ell_2|\mathbf{L}_{aj}|m+\ell_2\rangle_a = \varepsilon \times {}_a\langle k|\mathbf{L}_{aj}|m\rangle_a \,, \tag{6.1}
$$

where we recall that $\varepsilon = q^{\ell_2} = e^{i\pi\ell_1}$. As the periodicity suggests, only a finite part of the Lax operator is really relevant: we can truncate the auxiliary space to a finite-dimensional subspace. This goes via a variation of the construction from Section 5.1, as follows.

We decompose the infinite-dimensional highest-weight $U_q(\mathfrak{sl}_2)$-module $V_a^+$ as $V_{a'}^+ \oplus \tilde{V}_{a''}$, where $V_{a'}^+$ is the span of $|n\rangle_a$ with $n \geq \ell_2$ and $\tilde{V}_{a''}$ be the span of $|n\rangle_a$ for $0 \leq n \leq \ell_2 - 1$. At root of unity we have $[\ell_2]_q = 0$ so one of the entries of $\mathbf{S}_a^-$ vanishes as illustrated in Fig. 3. This time all generators of $U_q(\mathfrak{sl}_2)$ preserve the infinite-dimensional subspace $V_{a'}^+$. We are interested in the finite-dimensional subspace $\tilde{V}_{a''}$. Like in Section 4.2 we can get there by taking the quotient $\tilde{V}_{a''} \cong V_a^+/V_{a'}^+$. In this way we get a finite-dimensional representation of $U_q(\mathfrak{sl}_2)$ on $\tilde{V}_{a''}$, see also Appendix A.2. More concretely, all of $\mathbf{K}_a, \mathbf{S}_a^\pm$ are block upper triangular with respect to the decomposition $V_a^+ = V_{a'}^+ \oplus \tilde{V}_{a''}$, see again Fig. 3. This property is inherited by the Lax operator

$$
\mathbf{L}_{aj} = \begin{pmatrix} \mathbf{L}_{a'j} & \mathbf{L}_{a''j} \\ 0 & \tilde{\mathbf{L}}_{a''j} \end{pmatrix}, \tag{6.2}
$$

where $\mathbf{L}_{a'j}$ can be viewed as an infinite square matrix on $V_{a'}^+$, $\mathbf{L}_{a''j}$ as an $\ell_2 \times \infty$ rectangular matrix mapping $\tilde{V}_{a''}$ to $V_{a'}^+$, and $\tilde{\mathbf{L}}_{a''j}$ as an $\ell_2 \times \ell_2$ matrix on $\tilde{V}_{a''}$. The entries of each of these are $2 \times 2$ matrices acting at site $j$. The truncation to the $\ell_2$-dimensional space $\tilde{V}_{a''}$ amounts to treating all $|n\rangle_a$ with $n \geq \ell_2$ as zero.

Now focus on the $\ell_2$-dimensional auxiliary space $\tilde{V}_a$, where we drop the double prime. To describe the truncated Lax operator $\tilde{\mathbf{L}}_{aj}$ more explicitly define

$$
\tilde{\mathbf{W}}_a = \sum_{n=0}^{\ell_2-1} q^n |n\rangle\langle n|_a, \quad \tilde{\mathbf{X}}_a = \sum_{n=0}^{\ell_2-2} |n+1\rangle\langle n|_a, \quad \tilde{\mathbf{Y}}_a = \sum_{n=0}^{\ell_2-2} |n\rangle\langle n+1|_a. \tag{6.3}
$$

Replacing all operators on the auxiliary space in the factorised formula (4.5) by these truncated versions yields an expression with the same structure as (4.7):

$$
\tilde{\mathbf{L}}_{aj}(x,y) = \frac{1}{2} \begin{pmatrix} e^{y+\eta/2}\tilde{\mathbf{W}}_a - e^{-y-\eta/2}\tilde{\mathbf{W}}_a^{-1} & \tilde{\mathbf{Y}}_a(\tilde{\mathbf{W}}_a - \tilde{\mathbf{W}}_a^{-1}) \\ (e^{x-y}\tilde{\mathbf{W}}_a^{-1} - e^{-x+y}\tilde{\mathbf{W}}_a)\tilde{\mathbf{X}}_a & e^{x-\eta/2}\tilde{\mathbf{W}}_a^{-1} - e^{-x+\eta/2}\tilde{\mathbf{W}}_a \end{pmatrix}_j, \tag{6.4}
$$

which coincides with the right-hand side of (3.2). This time, however, all four matrix entries are finite matrices, with size depending on the value $\ell_2$ of the root of unity. Notice that the truncation allows us to keep $s \in \mathbb{C}$ arbitrary.

The truncated intertwiners are

$$
\begin{aligned}
\tilde{\mathbf{F}}_a(x, y) &= \sum_{n=0}^{\ell_2-1} \left[ \begin{matrix} (x - y - \eta)/\eta \\ n \end{matrix} \right]_q^{-1} |n\rangle\langle n|_a, \\
\tilde{\mathbf{G}}_{ab}(y, y') &= \sum_{n=0}^{\ell_2-1} \left[ \begin{matrix} (y - y')/\eta \\ n \end{matrix} \right]_q \left( \tilde{\mathbf{X}}_a \tilde{\mathbf{Y}}_b \right)^n, \\
\tilde{\mathbf{H}}_{ab}(x, x') &= \sum_{n=0}^{\ell_2-1} \left[ \begin{matrix} (x - x')/\eta \\ n \end{matrix} \right]_q \left( \tilde{\mathbf{Y}}_a \tilde{\mathbf{X}}_b \right)^n.
\end{aligned}
\tag{6.5}
$$

The intertwining relations are as before, where the truncation $\tilde{\mathbf{L}}_{aj}(\mathbf{v}_y, \mathbf{u}_x) = \tilde{\mathbf{L}}_{aj}(\mathbf{u}_x, \mathbf{v}_y)^{\mathsf{T}}$ to $\tilde{V}_a$ arises by restriction (rather than taking a quotient).

## 6.2 Decomposition of two-parameter transfer matrix at root of unity

The truncated Lax operator (6.4) can be used to construct monodromy matrices that yield truncated two-parameter transfer matrices $\tilde{\mathbb{T}}(x, y) = \tilde{\mathbf{T}}_s(u, \phi)$ where $x, y$ were defined in (4.4). These truncated two-parameter transfer matrices coincide with the "$Q_s(\exp \eta u)$" obtained by Korff in Ref. [34], who conjectured that they form a two-parameter family of commuting matrices. Since $s \in \mathbb{C}$ is unrestricted, the spectral parameters $x, y$, which were defined in (4.4), are now really just two free parameters. Using (6.5) as before we see that it obeys exchange relations just as in (4.20):

$$
\tilde{\mathbb{T}}(x, y, \phi) \tilde{\mathbb{T}}(x', y', \phi) = \tilde{\mathbb{T}}(x', y, \phi) \tilde{\mathbb{T}}(x, y', \phi) = \tilde{\mathbb{T}}(x, y', \phi) \tilde{\mathbb{T}}(x', y, \phi),
\tag{6.6}
$$

and in particular forms a family of commuting operators,

$$
\tilde{\mathbb{T}}(x, y, \phi) \tilde{\mathbb{T}}(x', y', \phi) = \tilde{\mathbb{T}}(x', y', \phi) \tilde{\mathbb{T}}(x, y, \phi).
\tag{6.7}
$$

As in Section 4.4 the truncated two-parameter transfer matrix therefore factorises as

$$
\tilde{\mathbb{T}}(x, y, \phi) = \tilde{\mathbf{Q}}_{y_0}(x, \phi) \tilde{\mathbf{P}}_{x_0, y_0}(y, \phi),
\tag{6.8}
$$

where the truncated Q and P operators are defined as

$$
\tilde{\mathbf{Q}}_{y_0}(x, \phi) := \tilde{\mathbb{T}}(x, y_0, \phi), \qquad \tilde{\mathbf{P}}_{x_0, y_0}(y, \phi) := \tilde{\mathbb{T}}(x_0, y, \phi) \tilde{\mathbb{T}}(x_0, y_0, \phi)^{-1}.
\tag{6.9}
$$

We once more suppress the dependence on $x_0, y_0$; we will get back to this in the remark in Section 7.4.

In terms of $\tilde{\mathbf{T}}_s(u, \phi)$ the factorisation takes the form

$$
\tilde{\mathbf{T}}_s(u, \phi) = \tilde{\mathbf{Q}}\left( u + \frac{2s+1}{2}\eta, \phi \right) \tilde{\mathbf{P}}\left( u - \frac{2s+1}{2}\eta, \phi \right).
\tag{6.10}
$$

## 6.3 Truncated Wronskian and TQ relations

Repeating the arguments from Section 5.1 for an arbitrary complex spin $s \in \mathbb{C}$ we have the decomposition of the highest-weight transfer matrix at root of unity

$$
\mathbf{T}_s^{\mathrm{hw}}(u, \phi) = e^{i\ell_2\phi} \, \mathbf{T}_{s-\ell_2}^{\mathrm{hw}}(u, \phi) + \tilde{\mathbf{T}}_s(u, \phi),
\tag{6.11}
$$

where up to the twist factor, $\mathbf{T}_{s-\ell_2}^{\mathrm{hw}}(u,\phi)$ coincides with the matrix $\mathbf{T}_s^{\mathrm{hw}}(u,\phi)$ restricted to the basis $|m+\ell_2\rangle$ with $m \geq 0$. With (6.1), we simplify the decomposition (6.11) into

$$\tilde{\mathbf{T}}_s(u,\phi) = \left(1 - \varepsilon^N e^{\mathrm{i}\ell_2\phi}\right)\mathbf{T}_s^{\mathrm{hw}}(u,\phi). \tag{6.12}$$

When $2s \in \mathbb{Z}_{\geq 0}$, the decompositions (5.5) and (6.12) yield a decomposition of $\mathbf{T}_s$ in terms of $\tilde{\mathbf{T}}_s$:

$$\begin{aligned}
\left(1 - \varepsilon^N e^{\mathrm{i}\ell_2\phi}\right)\mathbf{T}_s(u,\phi) &= \left(1 - \varepsilon^N e^{\mathrm{i}\ell_2\phi}\right)\left(\mathbf{T}_s^{\mathrm{hw}}(u,\phi) - e^{\mathrm{i}(2s+1)\phi}\,\mathbf{T}_{-s-1}^{\mathrm{hw}}(u,\phi)\right) \\
&= \tilde{\mathbf{T}}_s(u,\phi) - e^{\mathrm{i}(2s+1)\phi}\,\tilde{\mathbf{T}}_{-s-1}(u,\phi).
\end{aligned} \tag{6.13}$$

This argument relies on the convergence of the trace defining $\mathbf{T}_s^{\mathrm{hw}}(u,\phi)$ and requires $|e^{\mathrm{i}\phi}| < 1$. In Appendix D we give another proof of this relation valid for an arbitrary twist.

From (6.10) we obtain the *truncated Wronksian relation*

$$\begin{aligned}
\left(1 - \varepsilon^N e^{\mathrm{i}\ell_2\phi}\right)\mathbf{T}_s(u,\phi) &= \tilde{\mathbf{Q}}\left(u + \frac{2s+1}{2}\eta,\phi\right)\tilde{\mathbf{P}}\left(u - \frac{2s+1}{2}\eta,\phi\right) \\
&\quad - e^{\mathrm{i}(2s+1)\phi}\,\tilde{\mathbf{Q}}\left(u - \frac{2s+1}{2}\eta,\phi\right)\tilde{\mathbf{P}}\left(u + \frac{2s+1}{2}\eta,\phi\right).
\end{aligned} \tag{6.14}$$

When $1 - \varepsilon^N e^{\mathrm{i}\ell_2\phi} = 0$, i.e. for *commensurate twist*

$$\begin{aligned}
\varepsilon^N &= +1: & \phi &= \frac{(2n-2)\pi}{\ell_2}, \\
\varepsilon^N &= -1: & \phi &= \frac{(2n-1)\pi}{\ell_2},
\end{aligned} \qquad 1 \leq n \leq \ell_2, \tag{6.15}$$

the left-hand side of (6.14) vanishes. Comparing (6.15) to the conditions (2.15) for the existence of Bethe roots at infinity we see that when (6.15) is satisfied there exist certain numbers $M$ of down spins for which (2.15) is satisfied too.

Proceeding exactly as before we obtain TQ and TP relations that look the same as (5.10)–(5.11), now involving truncated matrices

$$\mathbf{T}_{1/2}(u,\phi)\tilde{\mathbf{Q}}(u,\phi) = T_0(u-\eta/2)\tilde{\mathbf{Q}}(u+\eta,\phi) + e^{\mathrm{i}\phi}\,T_0(u+\eta/2)\tilde{\mathbf{Q}}(u-\eta,\phi), \tag{6.16}$$

and

$$\mathbf{T}_{1/2}(u,\phi)\tilde{\mathbf{P}}(u,\phi) = e^{\mathrm{i}\phi}\,T_0(u-\eta/2)\tilde{\mathbf{P}}(u+\eta,\phi) + T_0(u+\eta/2)\tilde{\mathbf{P}}(u-\eta,\phi). \tag{6.17}$$

## 6.4 Truncated fusion relations

Proceeding as in Section 5.3, but using (6.10) instead of (5.7), we readily obtain fusion-like relations for $\tilde{\mathbf{T}}_s$:

$$\begin{aligned}
\mathbf{T}_{1/2}\left(u \pm \frac{2s+1}{2}\eta,\phi\right)\tilde{\mathbf{T}}_s(u,\phi) &= e^{\mathrm{i}\phi}\,T_0\left(u \pm (s+1)\eta\right)\tilde{\mathbf{T}}_{s-1/2}\left(u \mp \frac{\eta}{2},\phi\right) \\
&\quad + T_0(u \pm s\eta)\tilde{\mathbf{T}}_{s+1/2}\left(u \pm \frac{\eta}{2},\phi\right).
\end{aligned} \tag{6.18}$$

In analogy to the cases of general $q$, we will call these the *truncated fusion relations*. However, we stress that in the present case the internal auxiliary spaces have the same dimensions $\ell_2$ for all truncated transfer matrices in (6.18), unlike for the fusion relations (5.12).

## 6.5 Interpolation formula: proof of a conjecture

Fabricius and McCoy [32], Korff [34], and more recently De Luca *et al.* [41] — see Eq. (S22) in the supplementary material of Ref. [41] — conjectured that the complex spin transfer matrix eigenvalues can be expressed in terms of $\tilde{\mathbf{Q}}$, similarly for the situation of half-integral spin representations from Section 5.4. In our notation, after introducing the dependence on the twist, the formula reads

$$
\begin{aligned}
\tilde{\mathbf{T}}_s(u) = {} & \tilde{\mathbf{Q}}\left(u + \frac{2s+1}{2}\eta\right)\tilde{\mathbf{Q}}\left(u - \frac{2s+1}{2}\eta\right) \\
& \times \sum_{k=0}^{\ell_2-1} e^{ik\phi}\, \frac{T_0\bigl(u+(k-s)\,\eta\bigr)}{\tilde{\mathbf{Q}}\bigl(u+(k-s-\frac{1}{2})\eta\bigr)\tilde{\mathbf{Q}}\bigl(u+(k-s+\frac{1}{2})\eta\bigr)}\,.
\end{aligned}
\tag{6.19}
$$

We can prove this using arguments like those leading to (5.15). Rewrite (6.10) as

$$
\frac{\tilde{\mathbf{T}}_s(u)}{\tilde{\mathbf{Q}}\bigl(u+(s+1/2)\eta\bigr)\tilde{\mathbf{Q}}\bigl(u-(s+1/2)\eta\bigr)} = \frac{\tilde{\mathbf{P}}\bigl(u-(s+1/2)\eta\bigr)}{\tilde{\mathbf{Q}}\bigl(u-(s+1/2)\eta\bigr)}\,,
\tag{6.20}
$$

where we introduce a common factor on both sides for convenience. We also cast (6.14) specialised to $s = 0$ in the form

$$
\left(1-\varepsilon^N e^{i\ell_2\phi}\right)\frac{T_0(u)}{\tilde{\mathbf{Q}}(u+\eta/2)\tilde{\mathbf{Q}}(u-\eta/2)} = \frac{\tilde{\mathbf{P}}(u-\eta/2)}{\tilde{\mathbf{Q}}(u-\eta/2)} - e^{i\phi}\frac{\tilde{\mathbf{P}}(u+\eta/2)}{\tilde{\mathbf{Q}}(u+\eta/2)}\,.
\tag{6.21}
$$

Translating $u$ by $(k-s)\eta$, multiplying both sides by $e^{ik\phi}$ and summing over $k$ from 0 to $\ell_2-1$, we get

$$
\begin{aligned}
\left(1-\varepsilon^N e^{i\ell_2\phi}\right) & \sum_{k=0}^{\ell_2-1} e^{ik\phi}\,\frac{T_0\bigl(u+(k-s)\,\eta\bigr)}{\tilde{\mathbf{Q}}\bigl(u+(k-s+1/2)\eta\bigr)\tilde{\mathbf{Q}}\bigl(u+(k-s-1/2)\eta\bigr)} \\
& = \frac{\tilde{\mathbf{P}}\bigl(u-(s+1/2)\eta\bigr)}{\tilde{\mathbf{Q}}\bigl(u-(s+1/2)\eta\bigr)} - e^{i\ell_2\phi}\frac{\tilde{\mathbf{P}}\bigl(u+(\ell_2-s-1/2)\eta\bigr)}{\tilde{\mathbf{Q}}\bigl(u+(\ell_2-s-1/2)\eta\bigr)}\,.
\end{aligned}
\tag{6.22}
$$

From the parity of $\mathbf{Q}$ and $\mathbf{P}$ — see the discussion following (5.16) — it follows that the second ratio in the second line simplifies to $\varepsilon^N$ times the first ratio in that line. As long as the twist is *not* commensurate, i.e. does not obey (6.15), we can cancel $1-\varepsilon^N e^{i\ell_2\phi}$ on both sides of the equation. In fact, since the two sides of (6.22) are continuous in the twist, the result holds for any twist. (An alternative proof, that does not rely on this cancellation or completeness, can be obtained using the decomposition (6.11) like before rather than the simplification (6.12).) So we have

$$
\sum_{k=0}^{\ell_2-1} e^{ik\phi}\,\frac{T_0\bigl(u+(k-s)\,\eta\bigr)}{\tilde{\mathbf{Q}}\bigl(u+(k-s+1/2)\eta\bigr)\tilde{\mathbf{Q}}\bigl(u+(k-s-1/2)\eta\bigr)} = \frac{\tilde{\mathbf{P}}\bigl(u-(s+1/2)\eta\bigr)}{\tilde{\mathbf{Q}}\bigl(u-(s+1/2)\eta\bigr)}\,.
\tag{6.23}
$$

Multiplying both sides by $\tilde{\mathbf{Q}}\left(u+\frac{2s+1}{2}\eta\right)$, and using (6.10) we arrive at (6.19).

## 6.6 Structure of eigenvalues of $\tilde{\mathbf{Q}}$ and $\tilde{\mathbf{P}}$

Similar to the discussion in Section 5.5 the eigenvalues of the truncated Q operator,

$$
\tilde{\mathbf{Q}}(u,\phi)|\{v_m\}_{m=1}^M\rangle = \tilde{Q}(u,\{v_m\}_{m=1}^M,\phi)|\{v_m\}_{m=1}^M\rangle\,,
\tag{6.24}
$$

can be expressed as

$$\tilde{Q}(u, \{v_m\}_{m=1}^{M}, \phi) = \text{cst} \times \prod_{m=1}^{M} \left( t_m^{-1} t - t_m t^{-1} \right), \tag{6.25}$$

with $v_m = \log t_m$ obeying the Bethe equations (2.7) due to the truncated TQ equations (6.16). In particular the eigenvalues $Q(u, \phi)$ and $\tilde{Q}(u, \phi)$ of a given eigenvector have the same zeroes.

Since we are at root of unity, the eigenvalues of the truncated Q operator on the $M$-particle sector are quasiperiodic too:

$$\tilde{Q}(u \pm \ell_2 \eta, \phi) = \varepsilon^M \tilde{Q}(u, \phi). \tag{6.26}$$

Similarly it follows that

$$\tilde{\mathbf{Q}}(u \pm \ell_2 \eta, \phi) = \varepsilon^{N/2 - \mathbf{S}^z} \tilde{\mathbf{Q}}(u, \phi). \tag{6.27}$$

Likewise, we can show that the eigenvalues of the truncated P operator are of the form (5.21) but with $M$ replaced by $N - M$, and

$$\tilde{\mathbf{P}}(u \pm \ell_2 \eta, \phi) = \varepsilon^{N/2 + \mathbf{S}^z} \tilde{\mathbf{P}}(u, \phi). \tag{6.28}$$

### 6.7 Connection to the work of Frahm *et al*

Frahm, Morin-Duchesne and Pearce [70] observed that the zeroes of the eigenvalues of the Q operator appear as part of zeroes of eigenvalues of the transfer matrix whose argument is shifted as $\mathbf{T}_{(\ell_2-1)/2}(u + \ell_2 \eta/2, \phi)$ at root of unity. In our framework this is clear too. From the factorised expression (6.10) we see that the zeroes of the eigenvalues of $\tilde{\mathbf{T}}_s(u, \phi)$ are either zeroes of $\tilde{Q}\left(u + \frac{2s+1}{2}\eta, \phi\right)$ or zeroes of $\tilde{P}\left(u - \frac{2s+1}{2}\eta, \phi\right)$. Specialising $s = (\ell_2 - 1)/2$, $\tilde{\mathbf{T}}_{(\ell_2-1)/2} = \mathbf{T}_{(\ell_2-1)/2}$ and this agrees with Ref. [70].

# 7 Applications to XXZ at root of unity: general results

Now we turn to applications of the general formalism developed in Sections 3–6 to the spectrum of the XXZ spin chain (and the six-vertex model's transfer matrix) at root of unity. In this section we use the formalism to derive several general results. This and other features of the spectrum at root of unity will be illustrated by numerous explicit examples in Section 8.

## 7.1 Preliminaries

Let us first define a few useful concepts. Recall that the Q operator and all transfer matrices commute with each other. We will call a joint eigenvector of this family of operators a *state*. We will assume that the Bethe ansatz is complete, so that any state is of the form $|\{u_m\}_{m=1}^{M}\rangle$, see (3.11) and subsequent remark, with $\{u_m\}_{m=1}^{M}$ a solution to the Bethe equations (2.7).

Consider for a moment the periodic isotropic Heisenberg XXX spin chain. Here infinite rapidities (vanishing quasimomenta) can be added to the Bethe roots to get $\mathfrak{sl}_2$-descendants (see e.g. §1.1.3 in [10]). Unlike when a finite root is added, infinite rapidities do not change the values of the other Bethe roots, and the eigenvalues for the transfer matrix stays the same as well. Likewise, for the XXZ model at root of unity there are states $|\{u_m\}_{m=1}^{M}\rangle$ for which certain special roots can be added to get another state without affecting the values of $\{u_m\}_{m=1}^{M}$ or eigenvalues. We will call states that are minimal in this sense 'primitive'; it is similar to a highest-weight condition except that we do not use representation theory to characterise it. Namely, we call a state $|\{u_m\}_{m=1}^{M}\rangle$ *primitive* if no nontrivial subset of $\{u_m\}_{m=1}^{M}$ corresponds to a

physical state whose eigenvalue for $\mathbf{T}_s$ differs at most by a sign. (For the reason why we allow for a sign see Section 7.2.)

Primitive states have no FM strings. Recall that by $n_{\pm\infty}$ we denote the numbers of Bethe roots at $\pm\infty$, respectively. Typically, namely for twist $\phi \notin \{0, \pi\}$, a primitive state has no roots at infinity either.[8] In the (anti)periodic case $\phi \in \{0, \pi\}$ a primitive state may have $n_{\pm\infty} \neq 0$ as long as $n_{\mp\infty} = 0$, see (2.18) and Section 8.1.3.

Let us remark that it is possible for two primitive eigenstates to be degenerate, namely when $N$ is odd and the two states are related by the spin flip operator $\prod_j \sigma_j^x$: we will discuss this case in Section 7.4.

Any state that is not primitive is a *descendant* of some primitive state: it satisfies

$$\mathbf{Q}(u, \phi)|\{v_{m'}\}_{m'=1}^{M'}\rangle \propto \prod_{m=1}^{M} \sinh(u - u_m) \prod_{n=M+1}^{M'} \sinh(u - w_n)|\{v_{m'}\}_{m'=1}^{M'}\rangle, \qquad (7.1)$$

where $|\{u_m\}_{m=1}^{M}\rangle$ is primitive and the additional Bethe roots $\{w_n\}_{n=M+1}^{M'}$ consist of FM strings and roots located at $\pm\infty$. Away from the isotropic points descendant states only exist when condition (6.15) is satisfied.

Finally, for $\phi \notin \{0, \pi\}$ away from the (anti)periodic points, a primitive state and its descendants might be eigenstates for Hamiltonians with opposite twist, $\phi$ and $-\phi$, see Appendix C.3. The sign of the twist in $\mathbf{T}_s(u, \pm\phi)$ is fixed accordingly. We will further illustrate this in Section 8.2.

**Remark.** We have used the term 'descendant' here, because the corresponding 'descendant tower' that we will introduce in Section 8 resembles the descendant structures of Lie algebras. We use 'primitive' rather than 'highest weight' to stress that we do not use representation theory to characterise these states. (One could similarly use 'derived state' rather than 'descendant', but we will refrain from doing so. We hope that this will not cause too much confusion.) In fact, primitive states do have highest weight for the loop algebra of $\mathfrak{sl}_2$ [39], cf. [32]. From our perspective this algebraic perspective is not completely satisfactory since the loop-algebra action does not commute with the two-parameter transfer matrix or Q operator: as the example in Section 2.3 illustrates, the loop-algebra descendants are no longer eigenvectors of the latter. We do believe that there exist a deeper algebraic structure that is compatible with the Q operator, which would allow for unambiguous use of the terms 'highest-weight' and 'descendant'. We intend to investigate the underlying algebraic structure in the future.

## 7.2 Impact of FM strings on transfer-matrix eigenvalues

With this terminology let us show that a descendant state $|\{v_{m'}\}_{m'=1}^{M'}\rangle$ of a primitive state $|\{u_m\}_{m=1}^{M}\rangle$ has $\mathbf{T}_s(u, \pm\phi)$-eigenvalues that differ by at most a sign. This means that the momentum of the primitive state and its descendant may differ by $\pi$ while all other (quasi-)local charges generated by $\mathbf{T}_s(u, \pm\phi)$ are identical.

Write the eigenvalues of $\mathbf{T}_{1/2}$ and the Q operator for the primitive state $|\{u_m\}_{m=1}^{M}\rangle$ as

$$\begin{aligned}
\mathbf{T}_{1/2}(u, \phi)|\{u_m\}_{m=1}^{M}\rangle &= T_{1/2}(u)|\{u_m\}_{m=1}^{M}\rangle, \\
\mathbf{Q}(u, \phi)|\{u_m\}_{m=1}^{M}\rangle &= Q(u)|\{u_m\}_{m=1}^{M}\rangle.
\end{aligned} \qquad (7.2)$$

---

[8] States with $n_{\pm\infty} \neq 0$ but $n_{\mp\infty} = 0$ do occur at twist $\phi \notin \{0, \pi\}$. In Section 8.2 we will show that they have the same $\mathbf{T}_s$-eigenvalues as certain primitive states at twist $-\phi$ that have the same finite Bethe roots, and of which they should be considered descendants.

For simplicity we assume that $|\{u_m\}_{m=1}^M\rangle$ does not contain any Bethe roots at $\pm\infty$. By Section 5.5 the eigenvalue of the Q operator is a trigonometric polynomial of degree $M$,

$$Q(u) \propto \prod_{m=1}^{M} (t_m^{-1} t - t_m t^{-1}), \qquad t = e^u, \tag{7.3}$$

and satisfies the functional TQ relation

$$T_{1/2}(u)Q(u) = T_0(u - \eta/2)Q(u + \eta) + e^{i\phi} T_0(u + \eta/2)Q(u - \eta). \tag{7.4}$$

A descendant $|\{v_{m'}\}_{m'=1}^{M'}\rangle$ of the primitive state $|\{u_m\}_{m=1}^M\rangle$ is also an eigenvector of the Q operator,

$$\mathbf{Q}(u, \pm\phi)|\{v_{m'}\}_{m'=1}^{M'}\rangle = Q'(u)|\{v_{m'}\}_{m'=1}^{M'}\rangle, \tag{7.5}$$

where for $\phi \notin \{0, \pi\}$ the sign of the twist $\pm\phi$ has to be chosen to match that of the Hamiltonian, see Section 8.2. From the definition (2.19) of an FM string we note that the eigenvalue of the Q operator on the descendant state is

$$Q'(u) \propto Q(t) \, t^{n_{-\infty} - n_{+\infty}} \prod_{m=1}^{n_{\mathrm{FM}}} (t^{\ell_2} - e^{2\ell_2 \alpha_m^{\mathrm{FM}}} t^{-\ell_2}), \qquad t = e^u, \tag{7.6}$$

where $n_{\pm\infty}$ and $n_{\mathrm{FM}}$ are the numbers of roots at $\pm\infty$ and FM strings, respectively, of the descendant state. Observe that $Q'(u)$ satisfies the TQ relation

$$\varepsilon^{n_{\mathrm{FM}}} T_{1/2}(u)Q'(u) = T_0(u - \eta/2)Q'(u + \eta) + e^{i\phi} T_0(u + \eta/2)Q'(u - \eta), \tag{7.7}$$

where we recall that $\varepsilon = q^{\ell_2} \in \{\pm 1\}$. Hence, the eigenvalue of $\mathbf{T}_{1/2}$ for the descendant state $|\{v_{m'}\}_{m'=1}^{M'}\rangle$ is

$$\mathbf{T}_{1/2}(u, \pm\phi)|\{v_{m'}\}_{m'=1}^{M'}\rangle = \varepsilon^{n_{\mathrm{FM}}} T_{1/2}(u)|\{v_{m'}\}_{m'=1}^{M'}\rangle. \tag{7.8}$$

Since the $\mathbf{T}_{1/2}$-eigenvalue of the descendant state differs by at most a sign from that of the primitive state, the descendant has the same local charges generated by $\mathbf{T}_{1/2}$ as the primitive state, except that its momentum might differ by $\pi$. More generally, from the transfer matrix fusion relations (5.12) we obtain

$$\begin{aligned} \mathbf{T}_s(u, \phi)|\{u_m\}_{m=1}^M\rangle &= T_s(u)|\{u_m\}_{m=1}^M\rangle, \\ \mathbf{T}_s(u, \pm\phi)|\{v_{m'}\}_{m'=1}^{M'}\rangle &= \varepsilon^{2s\, n_{\mathrm{FM}}} T_s(u)|\{v_{m'}\}_{m'=1}^{M'}\rangle. \end{aligned} \tag{7.9}$$

We will give several explicit examples of the descendant states associated with primitive states, and the descendant towers that they form, in Sections 8.1–8.2.

## 7.3 FM strings at commensurate twist

In Section 6.3 we saw that, at root of unity, when the twist $\phi$ is commensurate as in (6.15) the truncated Wronskian relations (6.14) trivialise in the sense that the eigenvalues of $\tilde{\mathbf{Q}}$ and $\tilde{\mathbf{P}}$ are proportional to each other. This can only be achieved when the eigenvalues of $\tilde{\mathbf{P}}$ are related to those of $\tilde{\mathbf{Q}}$ through adding or subtracting Bethe roots at infinity or FM strings, cf. (6.14): eigenstates with FM strings must occur when the twist is commensurate. Conversely, if FM strings are present amongst the zeroes of the eigenvalues of $\tilde{\mathbf{Q}}$ or $\tilde{\mathbf{P}}$ the truncated Wronskian relations (6.14) vanish, since for $s = 0$ the left-hand side of (6.14) does not contain zeroes of FM strings while the right-hand side does. This means that descendants containing FM strings can only exist when the twist $\phi$ is commensurate. In conclusion, FM strings can only, and necessarily do, occur at commensurate twist.

From (6.12), whenever $\tilde{\mathbf{T}}_s(u, \phi)$ has nonzero eigenvalues it follows that the eigenvalues $\mathbf{T}_s^{\mathrm{hw}}(u, \phi)$ must have poles if $1 - \varepsilon^N e^{i\ell_2 \phi} = 0$ to compensate the vanishing prefactor. This corroborates Refs. [25, 28] in which it was shown that terms proportional to $\log t$ can arise when solving the functional TQ relations, yielding eigenvalues that are *quasi-polynomials* in $t$ [28]. As it turns out, the appearance of these quasi-polynomials is also closely related to the FM strings and their string centres, cf. (6.15). A detailed discussion of this will appear in a forthcoming article [71].

When (6.15) is satisfied, a quantisation condition for the centres of FM strings based on (6.23) was obtained in Refs. [32, 34, 36]. Let us review their arguments. For an eigenstate $|\{u_m\}_{m=1}^M\rangle$ we have

$$\tilde{\mathbf{Q}}(u, \phi)|\{u_m\}_{m=1}^M\rangle = Q(u)|\{u_m\}_{m=1}^M\rangle,$$
$$\tilde{\mathbf{P}}(u, \phi)|\{u_m\}_{m=1}^M\rangle = P(u)|\{u_m\}_{m=1}^M\rangle, \tag{7.10}$$

where the eigenvalues are trigonometric polynomials in $t = e^u$. Let us decompose the Q function in to a 'regular' part $Q_{\mathrm{r}}(u)$, consisting of $n_{\mathrm{r}}$ factors whose zeroes are Bethe roots that are neither FM strings nor infinite, a 'singular' part $Q_{\mathrm{s}}(u)$, consisting of $n_{\mathrm{FM}}$ FM strings, along with factors accounting for Bethe roots at infinity as in (5.21):

$$Q(u) = Q_{\mathrm{r}}(u) Q_{\mathrm{s}}(u) t^{n_{-\infty} - n_{+\infty}}, \qquad t = e^u,$$
$$Q_{\mathrm{r}}(u) \propto \prod_{m=1}^{n_{\mathrm{r}}} \left( t_m^{-1} t - t_m t^{-1} \right), \qquad Q_{\mathrm{s}}(u) \propto \prod_{m=1}^{n_{\mathrm{FM}}} \left( t_m^{-\ell_2} t^{\ell_2} - t_m^{\ell_2} t^{-\ell_2} \right), \tag{7.11}$$

where the number of down spins $M = n_{\mathrm{r}} + \ell_2 n_{\mathrm{FM}} + n_{+\infty} + n_{-\infty}$. The proportionality signs in (7.11) indicate equality up to $t$-independent factors. Because the Wronskian (6.14) vanishes, the eigenvalue $P(u)$ has the same regular zeroes as $Q(u)$:

$$P(u) = Q_{\mathrm{r}}(u) P_{\mathrm{s}}(u) t^{\bar{n}_{-\infty} - \bar{n}_{+\infty}}, \qquad P_{\mathrm{s}}(u) \propto \prod_{m=1}^{\bar{n}_{\mathrm{FM}}} \left( \bar{t}_m^{-\ell_2} t^{\ell_2} - \bar{t}_m^{\ell_2} t^{-\ell_2} \right), \tag{7.12}$$

where $\bar{n}_{\pm\infty}$ and $\bar{n}_{\mathrm{FM}}$ are the number of Bethe roots at $\pm\infty$ and FM strings, respectively, present amongst the zeroes of the P operator. The total number of zeroes of $Q(u)$ and $P(u)$ is equal to the system size $N$:

$$N = 2 n_{\mathrm{r}} + \ell_2 n_{\mathrm{FM}} + \ell_2 \bar{n}_{\mathrm{FM}} + n_{+\infty} + n_{-\infty} + \bar{n}_{+\infty} + \bar{n}_{-\infty}. \tag{7.13}$$

Consider the functional form of the interpolation formula (6.23),

$$P(u) = Q(u) \sum_{k=0}^{\ell_2 - 1} e^{ik\phi} \frac{T_0\big(u + (k + 1/2)\eta\big)}{Q(u + k\eta) Q\big(u + (k+1)\eta\big)}. \tag{7.14}$$

Using (7.11)–(7.12) and the fact that $Q_{\mathrm{s}}(u + \eta) = \varepsilon^{n_{\mathrm{FM}}} Q_{\mathrm{s}}(u)$ this can be rewritten as

$$\varepsilon^{n_{\mathrm{FM}}} Q_{\mathrm{s}}(u) P_{\mathrm{s}}(u) t^{n_{-\infty} - n_{+\infty} + \bar{n}_{-\infty} - \bar{n}_{+\infty}}$$
$$= \sum_{k=0}^{\ell_2 - 1} e^{ik\phi} \frac{T_0\big(u + (k + 1/2)\eta\big)}{Q_{\mathrm{r}}(u + k\eta) Q_{\mathrm{r}}\big(u + (k+1)\eta\big) e^{i(2k+1)(n_{-\infty} - n_{+\infty})\eta}}. \tag{7.15}$$

This is a 'quantisation condition' for the string centres of FM strings. Indeed, (7.15) implies that for any two states belonging to the same descendant tower, i.e. sharing the same regular part of their Q functions, the combination $Q_{\mathrm{s}}(u) P_{\mathrm{s}}(u)$ contains the same zeroes. In other words, the string centres of FM strings for any state belonging to a descendant tower are determined solely by the regular part of the Q functions. Moreover, string centres of FM strings are *free*

within a descendant tower in the sense that adding or removing FM strings whose string centres are given by the zeroes of $Q_s(u)P_s(u)$ results in other eigenstates within the same descendant tower. We can add FM strings from the zeroes of $Q_s(u)P_s(u)$ to the primitive state in order to generate the descendant states. We present explicit examples of the resulting tower structures in Sections 8.1–8.2.

### 7.4 Primitive degenerate eigenstates

When the system size $N$ is odd and condition (6.15) for a commensurate twist is satisfied, it is possible to have two primitive eigenstates with degenerate eigenvalues of $\mathbf{T}_s$ [72]. Namely, the two eigenstates are related by reversing all spins,

$$|\{v_{m'}\}_{m'=1}^{N-M}\}\rangle = \prod_{j=1}^{N} \sigma_j^x \, |\{u_m\}_{m=1}^{M}\}\rangle, \tag{7.16}$$

and, most importantly, their Bethe roots $\{v_{m'}\}_{m'=1}^{N-M}$ and $\{u_m\}_{m=1}^{M}$ are completely different: both are primitive states. This is only possible when the parities of $M$ and $N-M$ differ, i.e. the system size $N$ is odd.

When this happens, the eigenvalues of $\tilde{\mathbf{T}}_s$ are zero for both states,

$$\tilde{\mathbf{T}}_s(u, \phi)|\{u_m\}_{m=1}^{M}\}\rangle = \tilde{\mathbf{T}}_s(u, -\phi)|\{v_{m'}\}_{m'=1}^{N-M}\}\rangle = 0. \tag{7.17}$$

In this case, the matrix $\tilde{\mathbb{T}}(x, y, \phi) = \tilde{\mathbf{T}}_s(u, \phi)$ can not be inverted for any $x, y \in \mathbb{C}$. Despite this, the decomposition (6.20) still applies, where rather than via (6.9) the P operator is defined through the TQ relation for the state beyond the equator. However, it is no longer possible determine the eigenvalues of the Q operator $\tilde{\mathbf{Q}}$ for the degenerate primitive states. In practice this is not a problem since, as both states are primitive, we can use the numerical recipe in Appendix C.1 to determine the zeroes of the Q functions (Bethe roots) for both states by solving the functional TQ relation numerically.

**Curiosity at supersymmetric point.** In general the number of degenerate primitive eigenstates at root of unity and commensurate twist increases as the (odd) system size $N$ grows. The exception to this rule is the special point $\eta = \frac{2\pi}{3}$, $\phi = 0$ (or $\eta = \frac{\pi}{3}$, $\phi = \pi$, cf. Appendix C.2), for which there are only two degenerate primitive eigenstates for any odd $N$. (There are no degenerate primitive eigenstate when $N$ is even.) These values of $\eta$ correspond to the supersymmetric point $\Delta = -\frac{1}{2}$. For commensurate twist $\phi = 0$ and odd system size $N$, the antiferromagnetic ground states are always doubly degenerate, and have been well studied. We call them *Razumov–Stroganov (RS) states*, from the conjecture made in Ref. [73] and later proven in Ref. [74]. RS states are closely related to lattice supersymmetry [75,76]. (Note that in our convention for the Hamiltonian the RS states are the *highest* excited states in the spectrum.)

Interestingly, RS states are the only two primitive states that have degenerate eigenvalues for $\mathbf{T}_{1/2}$ at $\Delta = -1/2$, for any odd $N$. Let us denote them by $|\mathrm{RS}_1\rangle$, in the sector with $M = (N-1)/2$ down spins, and $|\mathrm{RS}_2\rangle = \prod_{j=1}^{N} \sigma_j^x \, |\mathrm{RS}_1\rangle$, with $M = (N+1)/2$. Their eigenvalues for $\mathbf{T}_{1/2}$ are

$$\mathbf{T}_{1/2}(u, 0)|\mathrm{RS}_1\rangle = \sinh^N(u)|\mathrm{RS}_1\rangle, \quad \mathbf{T}_{1/2}(u, 0)|\mathrm{RS}_2\rangle = \sinh^N(u)|\mathrm{RS}_2\rangle. \tag{7.18}$$

The Bethe roots can be obtained using numerical method in Appendix C.1. For instance, for $N = 5$ we have

$$|\mathrm{RS}_1\rangle = |\{u_m\}_{m=1}^{2}\rangle,$$

$$u_1 = \frac{\log(11 - \sqrt{21}) - \log 10}{2} + \frac{\mathrm{i}\pi}{2}, \quad u_2 = \frac{\log(11 + \sqrt{21}) - \log 10}{2} + \frac{\mathrm{i}\pi}{2}, \tag{7.19}$$

and

$$|\text{RS}_2\rangle = |\{v_{m'}\}_{m'=1}^3\rangle,$$
$$v_1 = \frac{i\pi}{2}, \quad v_2 = \frac{\log(2-\sqrt{3})}{2} + \frac{i\pi}{2}, \quad v_3 = \frac{\log(2+\sqrt{3})}{2} + \frac{i\pi}{2}. \tag{7.20}$$

One naturally wonders what the structure of the truncated two-parameter transfer matrix $\tilde{\mathbb{T}}(x, y, \phi)$ is at $\Delta = 1/2$, and how it relates to the RS states. We will postpone this question to future work.

## 7.5   Q functions for fully polarised states

Since all transfer matrices and the Hamiltonian commute with the total magnetisation $\mathbf{S}^z$, the two fully polarised states $|\uparrow\uparrow\cdots\uparrow\rangle$ and $|\downarrow\downarrow\cdots\downarrow\rangle$ are eigenstates of all transfer matrices, and therefore of the Q and P operators. Let us study this in more detail.

For the pseudovacuum $|\uparrow\uparrow\cdots\uparrow\rangle$ on top of which magnon excitations are built it is easy to see that for any system size $N$ and twist $\phi$,

$$\tilde{\mathbf{Q}}(u,\phi)|\uparrow\uparrow\cdots\uparrow\rangle = Q_{\Uparrow}(u,\phi)|\uparrow\uparrow\cdots\uparrow\rangle, \qquad Q_{\Uparrow}(u,\phi) = \text{cst}, \tag{7.21}$$

with Q function that does not depend on $u$ or $\phi$.

From the definition of the two-parameter transfer matrix (3.6) and (6.4) we have

$$\tilde{\mathbf{Q}}(u,\phi)|\downarrow\downarrow\cdots\downarrow\rangle = Q_{\Downarrow}(u,\phi)|\downarrow\downarrow\cdots\downarrow\rangle, \tag{7.22}$$

where the Q function is

$$\begin{aligned}
Q_{\Downarrow}(u,\phi) &= \langle\downarrow\downarrow\cdots\downarrow|\tilde{\mathbb{T}}(u,0,\phi)|\downarrow\downarrow\cdots\downarrow\rangle \\
&= \text{tr}_a\Big[\big(e^{u-\eta/2}\tilde{\mathbf{W}}_a^{-1} - e^{-u+\eta/2}\tilde{\mathbf{W}}_a\big)^N \tilde{\mathbf{E}}(\phi)\Big] \\
&= \sum_{k=0}^{\ell_2-1}\big(q^{-k-1/2}t - q^{k+1/2}t^{-1}\big)^N e^{ik\phi}, \qquad t = e^u,
\end{aligned} \tag{7.23}$$

which is consistent with (7.15). (Note that one needs to compare $Q_{\Downarrow}(u,\phi)$ with $P_{\Uparrow}(u,-\phi)$.)

From the final expression in (7.23) we see that

$$\begin{aligned}
Q_{\Downarrow}(u+\eta,\phi) = e^{i\phi}\Bigg[ &\sum_{k=0}^{\ell_2-2}\big(q^{-k-1/2}t - q^{k+1/2}t^{-1}\big)^N e^{ik\phi} \\
&+ \big(q^{-\ell_2+1/2}t - q^{\ell_2-1/2}t^{-1}\big)^N e^{i(\ell_2-1)\phi}\big(\varepsilon^N e^{-i\ell_2\phi}\big)\Bigg],
\end{aligned} \tag{7.24}$$

and since $\varepsilon = q^{\ell_2} = \pm 1$, we conclude that if $1 - \varepsilon^N e^{i\ell_2\phi} = 0$ then

$$Q_{\Downarrow}(u+\eta,\phi) = e^{i\phi}Q_{\Downarrow}(u,\phi). \tag{7.25}$$

This is precisely the commensurate-twist condition (6.15) for the appearance of FM strings. When $\phi = 0$ and the condition (6.15) is satisfied $Q_{\Downarrow}(u,\phi)$ is a trigonometric polynomial in $z = t^{\xi\ell_2} = e^{\xi\ell_2 u}$ with $\xi = 1$ ($\xi = 2$) if $\ell_1$ is even (odd). In that case the Bethe roots, i.e. the zeroes of (7.23), consist of only FM strings and pairs $+\infty, -\infty$.

# 8  Applications to XXZ at root of unity: examples

The physical Hilbert space $(\mathbb{C}^2)^{\otimes N}$ consists of all joint eigenstates of the transfer matrices. Assuming the completeness of the Bethe ansatz these states can be distinguished by their Bethe roots. Degeneracies are known to occur amongst the eigenvalues of the transfer matrices $\mathbf{T}_s$ when the condition (6.15) for commensurate twist is satisfied [20, 30, 31]. At first sight, this might resemble the degeneracies due to the $SU(2)$, or $\mathfrak{sl}_2 = (\mathfrak{su}_2)_{\mathbb{C}}$, symmetry at the isotropic point. Yet such degeneracies are not expected away from the isotropic point. In this section we will show with several concrete examples how to construct the descendant towers (Hasse diagrams) that link all different eigenstates with the same eigenvalues (possibly up to a sign for $\mathbf{T}_{1/2}$, cf. Section 7.2), for the transfer matrices. Moreover, as was observed by Fabricius and McCoy [32], we will illustrate in Section 8.1.1 that these degeneracies grow exponentially. This has significant consequences in the thermodynamic limit, which will be discussed in Section 10.2.

The discussions in this section are less rigorous than the previous sections. We have used the following method. In order to study the spectrum of XXZ spin chain at root of unity we need to know the Bethe roots associated to the eigenstates of the model, which is equivalent to knowing the Q functions, i.e. the eigenvalues of the Q operator, for these eigenstates. Making use of the decomposition of the truncated two-parameter transfer matrix $\tilde{\mathbf{Q}}(x, \phi) = \tilde{\mathbb{T}}(x, 0, \phi)$, we construct the Q operator for the XXZ spin chain at root of unity explicitly for system size $N \leq 16$. This is possible thanks to the truncation of the auxiliary space at root of unity. All the explicit examples in Section 8.1–8.2 are obtained through this procedure. In many instances, analytic expressions can be obtained using symbolic algebra software.

## 8.1  Descendant towers in periodic case

Let us first consider twist $\phi = 0$ vanishes, i.e. periodic boundary conditions, and illustrate how descendant states can be found from a primitive state together with FM strings or pairs $+\infty, -\infty$. The result will be a *descendant tower*, consisting of all eigenstates with degenerate (possibly up to a sign for $s = 1/2$) eigenvalues of $\mathbf{T}_s$ for all $2s \in \mathbb{Z}_{>0}$. At $\phi = 0$ every eigenstate $|\{u_m\}_{m=1}^M\}\rangle$ ($M < \frac{N}{2}$) has at least one eigenstate with the same eigenvalue for $\mathbf{T}_s$, namely the spin-flipped state *beyond the equator*,

$$|\{v_{m'}\}_{m'=1}^{N-M}\}\rangle \propto \prod_{j=1}^N \sigma_j^x \, |\{u_m\}_{m=1}^M\}\rangle \,. \tag{8.1}$$

We will refer to $|\{u_m\}_{m=1}^M\}\rangle$ and $|\{v_{m'}\}_{m'=1}^{N-M}\}\rangle$ as states on the opposite side of the equator with respect to each other. As we will illustrate, these two states are closely related when (6.15) is satisfied, forming the top and bottom of a descendant tower that includes various intermediate states.

In this section we will examine the case with system size $N = 12$, anisotropy $\Delta = \frac{1}{2}$ ($\eta = \frac{i\pi}{3}$, so $\ell_1 = 1$ and $\ell_2 = 3$), and twist $\phi = 0$. Similar constructions of descendant towers apply to the antiperiodic case $\phi = \pi$, which we shall not discuss separately.

### 8.1.1  Descendant towers of FM strings and their 'free-fermion' nature

We start with an example of a descendant tower that is generated solely by adding FM strings to a primitive state. Consider the simplest primitive state: the pseudovacuum state $|\{\varnothing\}\rangle = |\uparrow\uparrow\cdots\uparrow\rangle$, with no magnons ($M_0 = 0$ spins down). The corresponding state beyond the equator, $\prod_{j=1}^{12} \sigma_j^x \, |\{\varnothing\}\rangle = |\downarrow\downarrow\cdots\downarrow\rangle$ has Q function given by (7.23):

$$Q(t) \propto t^{12} + 220 \, t^6 + 924 + 220 \, t^{-6} + t^{-12} \,. \tag{8.2}$$

It is easy to verify (7.25). The zeroes of this Q function[9] are of the form

$$\alpha_1^{\text{FM}} = \alpha_1 + i\frac{\pi}{6}, \quad \alpha_2^{\text{FM}} = \alpha_2 + i\frac{\pi}{6}, \quad \alpha_3^{\text{FM}} = \alpha_3 + i\frac{\pi}{6}, \quad \alpha_4^{\text{FM}} = \alpha_4 + i\frac{\pi}{6}, \quad (8.3)$$

with numerical values of the real parts of the Bethe roots given by

$$\alpha_1 = -0.89566465, \quad \alpha_2 = -0.23210918, \quad \alpha_3 = 0.23210918, \quad \alpha_4 = 0.89566465. \quad (8.4)$$

Thus the state corresponding to $|\{\varnothing\}\rangle$ beyond the equator is itself a Bethe vector:

$$\prod_{j=1}^{12} \sigma_j^x \, |\{\varnothing\}\rangle = |\{v_{m'}\}_{m'=1}^{12}\rangle =: |\{\alpha_1^{\text{FM}}, \alpha_2^{\text{FM}}, \alpha_3^{\text{FM}}, \alpha_4^{\text{FM}}\}\rangle. \quad (8.5)$$

Now let us for a moment consider the free-fermion case $\Delta = 0$ ($\eta = i\pi/2$), $\phi = 0$, which was an important example for [37]. In that case the $S$-matrix reduces to $S = -1$, so the Bethe equations decouple with Bethe roots taking values of the form $\alpha_n \pm i\pi/2$ with $\alpha_n \in [0, 2\pi]$. Reality of the energy requires any solution to consists of complex-conjugate pairs $\alpha_n + i\pi/2, \alpha_n - i\pi/2$. The very simple resulting spectrum can be understood as consisting of FM strings of length $\ell_2 = 2$, and the free-fermion nature is visible in that any string can be added or removed without affecting the location of the other strings. See e.g. [36] for the details. More generally, for any root of unity, (7.15) implies that FM strings behave like *free fermions* within a descendant tower. For a primitive state $|\{u_m\}_{m=1}^M\rangle$, if the state $|\{u_m\}_{m=1}^M \cup \{\alpha_1^{\text{FM}}, \alpha_2^{\text{FM}}\}\rangle$ belongs to the descendant tower, then so does $|\{u_m\}_{m=1}^M \cup \{\alpha_n^{\text{FM}}\}\rangle$ for $n \in \{1, 2\}$. That is, FM strings are 'transparent': they do not scatter with (feel or influence the values of) other roots.

This observation immediately yields the following descendant tower. Given a primitive state, construct the corresponding eigenstate beyond the equator and find its Bethe roots as above. The descendant tower is obtained from the primitive state by adding FM strings one by one, with magnetisation $M$ jumping by $\ell_2$ each time, until we reach the eigenstate beyond the equator at the bottom. For example, the descendant tower obtained in this way from the pseudovacuum that we considered above is illustrated in Fig. 4. This structure is easily verified by explicitly constructing the Q operator, and one sees that the eigenvalues of the transfer matrices $\mathbf{T}_s$ are degenerate (up to a sign) for all descendant states.

In particular it follows that the number of descendant states within the magnetisation sector with number of down spins fixed to $M = M_0 + n\ell_2$ is equal to $\binom{n_{\text{FM}}}{n}$, i.e. grows binomially in the number $n_{\text{FM}}$ of FM strings for the state beyond the equator. By adding up all descendant states at the occurring values of $M$ we find that the total number within the descendant tower is

$$n_{\text{total}} = \sum_{n=0}^{n_{\text{FM}}} \binom{n_{\text{FM}}}{n} = 2^{n_{\text{FM}}}, \quad (8.6)$$

in agreement with the loop-algebra prediction, cf. Eq. (1.19) of Ref. [32]. The the descendant tower is exponentially large in the number $n_{\text{FM}}$ of FM strings present in the state beyond the equator corresponding to the primitive state.

### 8.1.2 Descendant towers with pairs of roots at infinity

Next we turn to an example of a descendant tower that is generated by adding FM strings as well as pairs $+\infty, -\infty$ to the Bethe roots of a primitive state. Recall that Bethe roots at infinity correspond to applications of the lowering operators of $U_q(\mathfrak{sl}_2)$ (see Appendix A.1).

---

[9] Recall that each pair $\pm t$ of zeroes of $Q$ corresponds to one Bethe root, cf. Footnote 3 on p. 6

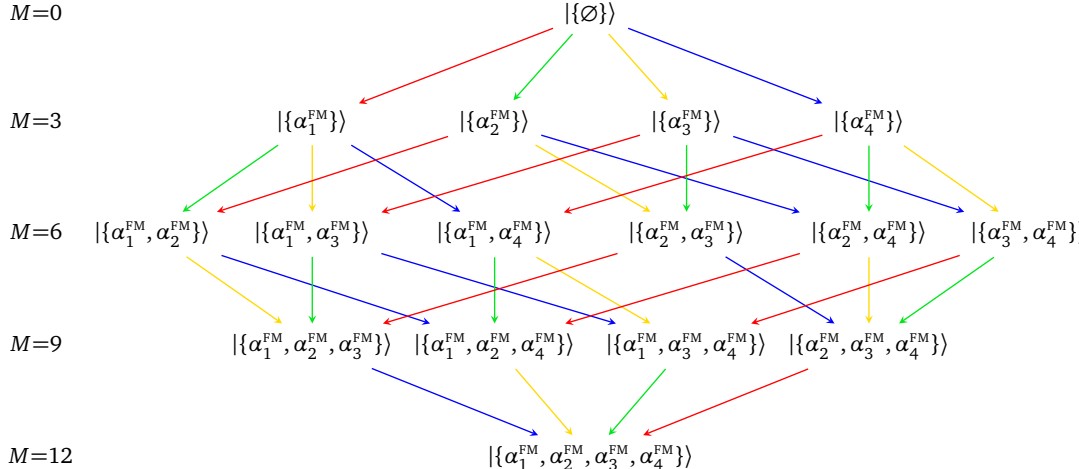

Figure 4: Illustration of a descendant tower from the pseudovacuum for $N = 12$, $\Delta = 1/2$, $\phi = 0$. Arrows of different colours correspond to the addition of different FM strings.

At root of unity $\ell_2$ applications of either of $\mathbf{S}^\pm$ or $\bar{\mathbf{S}}^\pm$ already gives zero, so the total numbers of Bethe roots at $+\infty$ and $-\infty$ must be smaller than $\ell_2$: we need $n_{\pm\infty} < \ell_2$. As we will illustrate, pairs of Bethe roots at $+\infty, -\infty$ play a similar role as FM strings.

We continue with the example $N = 12$, $\Delta = \frac{1}{2}$ ($\eta = \frac{i\pi}{3}$), $\phi = 0$. Let us now start from a primitive state with one down spin ($M_0 = 1$), $|\{u_1\}\rangle$, say for the solution to Bethe equations (2.7) given by

$$u_1 = \frac{1}{2} \log \frac{\sqrt{3}+1}{\sqrt{3}-1}. \tag{8.7}$$

Consider again the corresponding state beyond the equator, $|\{v_{m'}\}_{m'=1}^{11}\rangle \propto \prod_{j=1}^{12} \sigma_j^x |\{u_1\}\rangle$. We obtain its eigenvalue for the Q operator by solving (7.15), yielding the Q function

$$Q(t) \propto \prod_{m'=1}^{11} \left(t - e^{2v_{m'}} t^{-1}\right) = \left(t - \frac{\sqrt{3}+1}{\sqrt{3}-1} t^{-1}\right)\left(t^6 - \frac{59}{74}(-8 + 3\sqrt{3}) + \frac{91 - 48\sqrt{3}}{37} t^{-6}\right). \tag{8.8}$$

The zeroes of this Q function are $u_1$, two FM strings, along with (cf. Section 5.5) two pairs of roots $+\infty, -\infty$:

$$\begin{aligned} v_1 &= u_1, \\ v_{3(n-1)+k+1} &= \alpha_n^{\text{FM}} + \frac{i(k-2)\pi}{3}, \qquad 1 \leq n \leq n_{\text{FM}} = 2, \quad 1 \leq k \leq \ell_2 = 3, \\ v_8 &= v_{10} = +\infty, \quad v_9 = v_{11} = -\infty. \end{aligned} \tag{8.9}$$

Here the FM strings are centred at

$$\alpha_1^{\text{FM}} = \alpha_1 + i\frac{\pi}{6}, \quad \alpha_2^{\text{FM}} = \alpha_2 + i\frac{\pi}{6}, \tag{8.10}$$

with real parts that have numerical values

$$\alpha_1 = -0.38464681, \quad \alpha_2 = 0.12649136, \tag{8.11}$$

satisfying the quantisation condition (7.15). Note that $n_{\pm\infty} = 2 < \ell_2 = 3$.

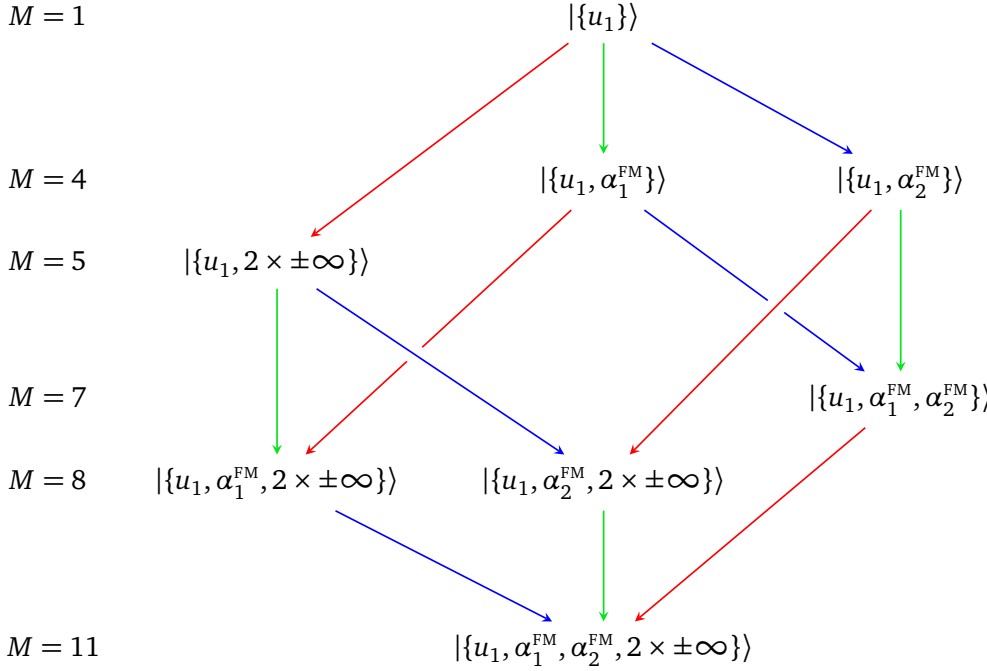

Figure 5: The descendant tower of $|\{u_1\}\rangle$ with finite $u_1$ for $N = 12$, $\Delta = 1/2$, $\phi = 0$.

Unlike for FM strings, Bethe roots at infinity cannot added one by one: $\{u_1, \pm\infty\}$ does not satisfy the Bethe equations, as it violates the condition (2.12). On the other hand, using (A.7) we can easily check that

$$|\{u_1, 2 \times \pm\infty\}\rangle := |\{u_1, +\infty, -\infty, +\infty, -\infty\}\rangle \propto (\mathbf{S}^-)^2 (\bar{\mathbf{S}}^-)^2 |\{u_1\}\rangle \tag{8.12}$$

is an eigenstate with the same eigenvalues for $\mathbf{T}_s$ as $|\{u_1\}\rangle$. That is, a pair of Bethe roots $+\infty, -\infty$ can be viewed as a 'bound state': these two roots always appear together in order to satisfy condition (2.15).[10] Like FM strings, pairs $+\infty, -\infty$ do not scatter with other magnons. Therefore, when the corresponding descendant state beyond the equator contains one pair of infinite Bethe roots as in our example, the total number of states in the descendant tower is

$$n_{\text{total}} = \sum_{m=0}^{n_{\text{FM}}+1} \binom{n_{\text{FM}} + 1}{m} = 2^{n_{\text{FM}}+1}, \tag{8.13}$$

as illustrated in Fig. 5.

Almost all (primitive) states at $M = 1$ give rise to a descendant tower of this form, only differing in the locations of the FM string centres. There are two exceptions, to which we turn next.

### 8.1.3 Mirroring descendant towers

There is one more interesting feature present in the spectrum for $N = 12$, $\Delta = \frac{1}{2}$ ($\eta = \frac{i\pi}{3}$), $\phi = 0$. At $M = 1$ down spin we find from (2.15) that there are two eigenstates, namely $|\{+\infty\}\rangle$ and $|\{-\infty\}\rangle$, which have an infinite Bethe root. These are primitive: their $\mathbf{T}_s$-eigenvalues are different from those of the pseudovacuum, reflecting that we are not at the

---

[10] Another viewpoint is that, unlike in the isotropic case, the coproduct of $U_q(\mathfrak{sl}_2)$ is 'chiral': the operators $\mathbf{S}^\pm$ and $\bar{\mathbf{S}}^\pm$ from (2.5) break the left-right (parity) symmetry of the Hamiltonian. The combination $\mathbf{S}^- \bar{\mathbf{S}}^-$, however, is compatible with parity invariance.

isotropic point. We will concentrate on $|\{+\infty\}\rangle$ in this section; the situation for $|\{-\infty\}\rangle$ is analogous.

In general, in the (anti)periodic case $\phi \in \{0, \pi\}$, the existence of an eigenstate of the form $|\{u_m\}_{m=1}^{M} \cup \{n \times +\infty\}\rangle$ for some $n > 0$ implies the existence of another eigenstate $|\{u_m\}_{m=1}^{M} \cup \{(\ell_2-n) \times -\infty\}\rangle$ with the same eigenvalues of $\mathbf{T}_s$ (up to a sign), whose Bethe roots can be found from the Bethe equations (2.7). (Likewise, the presence of a state $|\{u_m\}_{m=1}^{M} \cup \{n \times -\infty\}\rangle$ implies that of $|\{u_m\}_{m=1}^{M} \cup \{(\ell_2-n) \times +\infty\}\rangle$.)

One can verify that the state beyond the equator corresponding to $|\{+\infty\}\rangle$ does *not* belong to the same descendant tower as $|\{+\infty\}\rangle$, unlike for the examples presented in the previous sections. The eigenvalue for the Q operator on the corresponding state beyond the equator, $|\{v_{m'}\}_{m'=1}^{11}\}\rangle \propto \prod_{j=1}^{12} \sigma_j^x |\{+\infty\}\rangle$, is obtained through (7.15), yielding

$$Q(t) \propto \prod_{m'=1}^{11} \left(t - e^{2v_{m'}} t^{-1}\right) = t^{11} + \frac{165}{4} t^5 + 66 \, t^{-1} + \frac{11}{2} \, t^{-7}. \tag{8.14}$$

Its zeroes contain three FM strings and two Bethe roots at $-\infty$,

$$
\begin{aligned}
v_1 = v_2 &= -\infty, \\
v_{3(n-1)+k+2} = \alpha_n^{\mathrm{FM}} &+ \frac{i(k-2)\pi}{3}, \qquad 1 \le n \le n_{\mathrm{FM}} = 3, \quad 1 \le k \le \ell_2 = 3,
\end{aligned}
\tag{8.15}
$$

where the FM strings have centres

$$\alpha_n^{\mathrm{FM}} = \alpha_n + i\frac{\pi}{6}, \quad \alpha_1 = -0.404723313, \quad \alpha_2 = 0.075767627, \quad \alpha_3 = 0.613080368, \tag{8.16}$$

which satisfy the quantisation condition (7.15).

The resulting descendant-tower structure is as follows. The primitive state $|\{+\infty\}\rangle$ gives rise to descendants via the addition of the above FM strings. This yields a descendant tower that accounts for half of the states with the same eigenvalues (possibly up to a sign) of the transfer matrices $\mathbf{T}_s$. The other half of the states form a 'mirroring tower', obtained from the first tower by spin reversal. At the top of the mirroring tower we find the primitive eigenstate $|\{2 \times -\infty\}\rangle$, whose eigenvalues for $\mathbf{T}_s$ are the same as those of $|\{+\infty\}\rangle$ except for a sign for $s = 1/2$. Spin reversal relates states on opposite sides of the equator as

$$\prod_{j=1}^{12} \sigma_j^x |\{u_m\}_{m=1}^{M}\rangle \propto |\{v_{m'}\}_{m'=1}^{12-M}\rangle = |\{+\infty, 2 \times -\infty, \alpha_1^{\mathrm{FM}}, \alpha_2^{\mathrm{FM}}, \alpha_3^{\mathrm{FM}}\} \setminus \{u_m\}_{m=1}^{M}\rangle. \tag{8.17}$$

See Fig. 6 for an illustration.

## 8.2 Descendant towers for nonzero commensurate twist

Now we turn to commensurate twist (6.15) with $\phi \notin \{0, \pi\}$ away from the (anti)periodic case. It will be instructive to consider the two commensurate twists $\pm\phi$ simultaneously. Like with the mirror-pair of descendant towers in Section 8.1.3 we need to consider two copies of descendant towers. This time, however, the two copies include states at twist $\phi$ as well as states at *opposite* twist $-\phi$. Indeed, in Appendix C.3 we show that the transfer matrices $\mathbf{T}_s(u, \phi)$ with twist $\phi$ are related to those with opposite twist by flipping all spins,

$$\prod_{j=1}^{N} \sigma_j^x \, \mathbf{T}_s(u, \phi) \prod_{j=1}^{N} \sigma_j^x = e^{2si\phi} \, \mathbf{T}_s(u, -\phi). \tag{8.18}$$

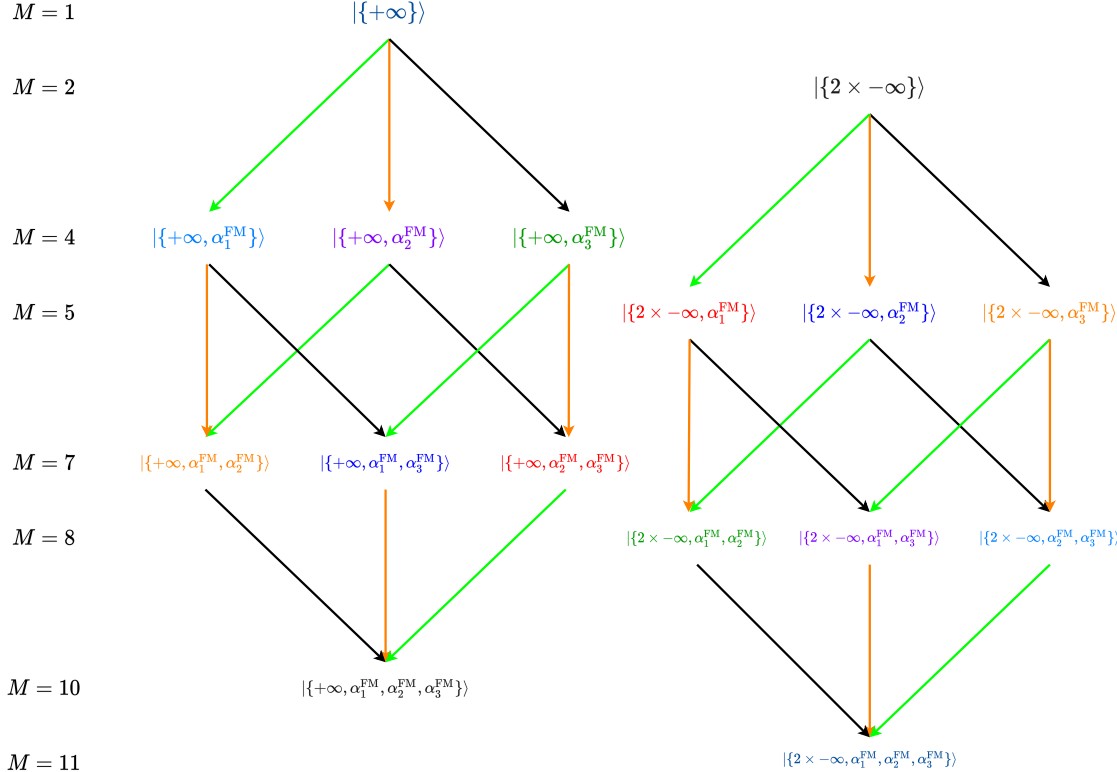

Figure 6: Illustration of descendant towers from degenerate primitive states at $M_0 = 1, 2$ with Bethe roots at infinity for $N = 12$, $\Delta = 1/2$, $\phi = 0$. Global spin reversal acts by reflection through the middle of the figure: the states with the same colour are pairs corresponding to each other beyond the equator.

Thus $\prod_{j=1}^{N} \sigma_j^x |\{u_m\}_{m=1}^M\rangle$ is an eigenstate of (the Hamiltonian, and more generally) the transfer matrices $\mathbf{T}_s(u, -\phi)$ whenever $|\{u_m\}_{m=1}^M\rangle$ is so for $\mathbf{T}_s(u, \phi)$, cf. (C.17). (Observe that $\phi = 0, \pi$ are the only two values for which $\phi = -\phi \bmod 2\pi$.)

We study the example $N = 6$, $\Delta = \frac{1}{2}$ (so $\eta = i\pi/3$) and commensurate twist $\phi = \pm\frac{2\pi}{3}$. We wish to understand the descendant tower for primitive states with $M_0 = 1$ down spin. These 'twisted magnons' are discussed in detail in Appendix B.2. Consider $|\{u_1\}\rangle$ with $u_1 = \arctan\frac{\tan(\pi/18)}{\sqrt{3}}$, which is an eigenvector for the Hamiltonian with twist $\phi = \frac{2\pi}{3}$. Its quasimomentum, see Eq. (B.9), is

$$\widetilde{p} = i\log\frac{\sinh(u_1 - \eta/2)}{\sinh(u_1 + \eta/2)} = \frac{6\pi - \phi}{6} = \frac{8\pi}{9}. \tag{8.19}$$

Next consider the primitive state $|\{u_2\}\rangle'$ with $u_2 = -\arctan\frac{\tan(\pi/18)}{\sqrt{3}}$, where we decorate the ket by a prime to indicate that it is an eigenstate for the Hamiltonian with twist $\phi' = -\frac{2\pi}{3}$.[11] This second state has opposite quasimomentum

$$\widetilde{p}' = i\log\frac{\sinh(u_2 - \eta/2)}{\sinh(u_2 + \eta/2)} = \frac{-6\pi - \phi'}{6} = -\frac{8\pi}{9} = \frac{10\pi}{9} \bmod 2\pi. \tag{8.20}$$

---

[11] With our conventions (see Section 3) the off-shell Bethe vectors are independent of the twist, and the dependence on $\phi$ only enters via the Bethe equations. The notation $|\{\dots\}\rangle'$ therefore is not really necessary, as the values of the Bethe roots implicitly determine the sign of the twist, yet we believe that it helps clarifying the discussion.

These two primitive states have the same eigenvalues for the energy (cf. Appendix B.2)

$$
\begin{aligned}
H(\phi)|\{u_1\}\rangle &= E\ |\{u_1\}\rangle, \\
H(-\phi)|\{u_2\}\rangle' &= E'\ |\{u_2\}\rangle',
\end{aligned}
\qquad E = E' = -\cos\frac{\pi}{9} - \frac{1}{2}, \tag{8.21}
$$

and, as $\varepsilon^{n_{\text{FM}}} = (-1)^0 = 1$ in (7.8), the same 'twisted' momentum $p = \tilde{p} + \phi/N = \tilde{p}' - \phi/N = \pi \bmod 2\pi$. We note, however, that these two states are *not* degenerate for the transfer matrices. For example, the fundamental transfer matrix $\mathbf{T}_{1/2}$ acts by

$$
\mathbf{T}_{1/2}(u,\phi)|\{u_1\}\rangle = \frac{e^{i\phi/2}}{64}\left(a_1\,t^6 + a_2\,t^4 + a_3\,t^2 + a_4 + a_5\,t^{-2} + a_6\,t^{-4} + a_7\,t^{-6}\right)|\{u_1\}\rangle, \tag{8.22}
$$

in terms of $t = e^u$ as usual, while

$$
\mathbf{T}_{1/2}(u,\phi')|\{u_2\}\rangle' = \frac{e^{i\phi'/2}}{64}\left(a_7\,t^6 + a_6\,t^4 + a_5\,t^2 + a_4 + a_3\,t^{-2} + a_2\,t^{-4} + a_1\,t^{-6}\right)|\{u_2\}\rangle' \tag{8.23}
$$

has the opposite order of the coefficients. The coefficients read

$$
\begin{aligned}
a_1 &= -2, \qquad a_2 = a_5 = 9 - 6\sin\frac{\pi}{18} - 6\cos\frac{\pi}{9}, \\
a_3 &= -6\left(1 - 3\cos\frac{\pi}{9} + \cos\frac{2\pi}{9} - 4\sin\frac{\pi}{18}\right), \\
a_4 &= -1 + 18\cos\frac{2\pi}{9} - 18\sin\frac{\pi}{18}, \\
a_6 &= -6\left(1 + \cos\frac{2\pi}{9} - \sin\frac{\pi}{18}\right), \qquad a_7 = 1.
\end{aligned} \tag{8.24}
$$

This reproduces the above values of $\tilde{p}, \tilde{p}'$, and thus $p$, through (B.9) and $E, E'$ via (B.6).

To construct the descendant towers we turn to the corresponding states beyond the equator. For $|\{u_1\}\rangle$ this is $\prod_{j=1}^{6}\sigma_j^x\,|\{u_1\}\rangle \propto |\{u_1, -\infty, \alpha_1^{\text{FM}}\}\rangle'$, whose eigenvalue for the Q operator $\tilde{\mathbf{Q}}(t,\phi')$ is

$$
\begin{aligned}
Q'(t) &\propto t\left(t - e^{2u_1}t^{-1}\right)\left(t^3 - e^{6\alpha_1^{\text{FM}}}t^{-3}\right) \\
&\approx t^5 - 1.2266816\,t^3 + 3.4456224\,t^{-1} - 4.2266816\,t^{-3}.
\end{aligned} \tag{8.25}
$$

The state beyond the equator corresponding to $|\{u_2\}\rangle'$ is $|\{u_2, +\infty, \alpha_2^{\text{FM}}\}\rangle$, with eigenvalue for the Q operator $\tilde{\mathbf{Q}}(t,\phi)$ given by

$$
\begin{aligned}
Q(t) &\propto t^{-1}\left(t - e^{2u_2}t^{-1}\right)\left(t^3 - e^{6\alpha_2^{\text{FM}}}t^{-3}\right) \\
&\approx t^3 - 0.8152075\,t + 0.2902233\,t^{-3} - 0.2365922\,t^{-5}.
\end{aligned} \tag{8.26}
$$

Here the string centres of the two FM strings are

$$
\alpha_1^{\text{FM}} = 0.20618409 + \frac{i\pi}{6}, \qquad \alpha_2^{\text{FM}} = -0.20618409 + \frac{i\pi}{6}. \tag{8.27}
$$

The descendant towers can now readily be constructed using the 'free fermion'-like property as in Section 8.1.1. The resulting tower structure is depicted in Fig. 7, where in general states come in pairs that have the same (possibly up to a sign, as always) eigenvalues for all transfer matrices $e^{\mp is\phi}\,\mathbf{T}_s(u, \pm\phi)$, where the sign in $\pm\phi$ is determined by the Hamiltonian for which the state is an eigenvector.

A key difference with the (anti)periodic case is that, in order to construct the descendant tower by considering the state beyond the equator corresponding to the primitive state, we are led to consider the systems with twists $\phi$ and $\phi' = -\phi$ simultaneously. The states *within*

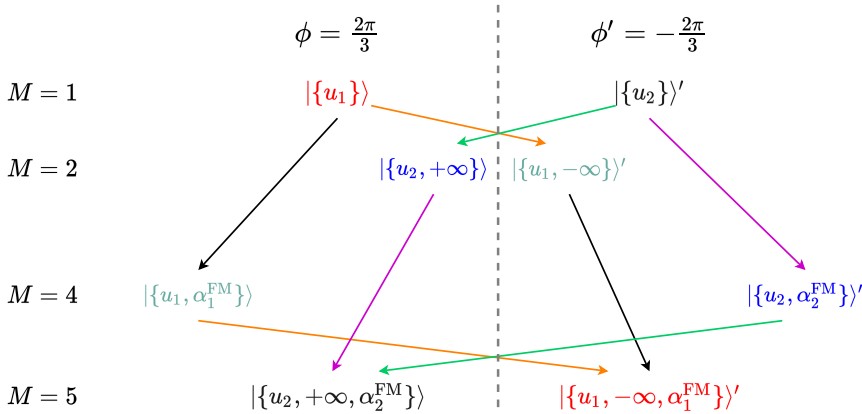

Figure 7: Illustration of descendant towers from the primitive states $|\{u_1\}\rangle$ and $|\{u_2\}\rangle'$ for $N = 6$, $\Delta = 1/2$ and commensurate twist $\phi = \frac{2\pi}{3}$ and $\phi' = -\phi$. Like in Fig. 6 states of the same colour correspond to each other via global spin reversal.

either tower in Fig. 7, e.g. $|\{u_1\}\rangle$ and $|\{u_1, -\infty\}\rangle'$, have degenerate (possibly up to a sign) eigenvalues for all $e^{\mp is\phi} \mathbf{T}_s(u, \pm\phi)$, with sign of the twist determined by the states. On the other hand, *between* the two different towers, the eigenstates are only degenerate for the momentum (possibly up to a shift by $\pi$) and energy: as we saw for the two primitive states this does *not* extend to the higher charges generated by the transfer matrices. This is the second important difference with the (anti)periodic case from Section 8.1.3, where all eigenstates in Fig. 6 have degenerate (up to a possible sign) eigenvalues for $\mathbf{T}_s$. To conclude this discussion we remark that the descendant-tower structure for twists $\pm\phi \notin \{0, \pi\}$ can get more complicated for roots of unity with $\ell_2 > 3$.

## 8.3 Full spectrum at root of unity: an example

Although we have not been able to find an algorithmic description of the structure of the spectrum for given values of the spin-chain parameters $N, \Delta, \phi$, our numerical investigations of numerous examples suggest that the full spectrum can be understood case by case in terms of the descendant towers that we have described in this section. Let us demonstrate this for a system with $N = 10$, $\Delta = \frac{1}{2}$ ($\eta = \frac{i\pi}{3}$) and $\phi = 0$. The resulting description of the full spectrum is given in Table 1.

We stress that, due to the vanishing Wronskian relation (6.14) and the 'free fermion'-like behaviour of FM strings, each primitive state is at the top of a descendant tower, with descendants that are obtained by adding FM strings or pairs of roots $+\infty, -\infty$. By (2.18) the states come in three categories, classified by the number of Bethe roots located at infinity:

  i) $n_{+\infty} = n_{-\infty}$ ($= 0$ for primitive states),

 ii) $n_{+\infty} > n_{-\infty} = 0$,

iii) $n_{-\infty} > n_{+\infty} = 0$.

More precisely, the conditions (2.15) and (2.17) allow for $n_{\pm\infty} = M + 1 \bmod 3$. Since of course $n_{+\infty} + n_{-\infty} \leq M$ infinite roots can first occur at $M = 3$ ($n_{\pm\infty} = 1$), then at $M = 4$ ($n_{\pm\infty} = 2$), not at $M = 5$ ($n_{\pm\infty} = 0$), after which the pattern repeats.

Start with $|\uparrow \cdots \uparrow\rangle = |\{\varnothing\}\rangle$. From the zeroes of the Q function (7.23) we find (cf. Footnote 9 on p. 34) that the corresponding state beyond the equator can be written as $|\downarrow \cdots \downarrow\rangle \propto$

$|\{\alpha_1^{\text{FM}}, \alpha_2^{\text{FM}}, 2 \times \pm\infty\}\rangle$, with FM strings of length $\ell_2 = 3$. We proceed as in Section 8.1.2. Draw the candidate descendant tower using the 'free-fermion' property. Note that (2.15) and (2.17) exclude all potential descendant states in this tower with one pair $+\infty, -\infty$ of infinite roots: all four infinite roots must be added at the same time. The primitive state at $M = 0$ thus gives rise to a descendant tower with two descendants at $M = 3$ and at $M = 7$, and one descendant at each of $M = 4, 6, 10$.

All ten states $|\{u_1\}\rangle$ at $M = 1$ must be primitive. Their Bethe root are finite in view of (2.15) and (2.17). The corresponding states beyond the equator have room for two FM strings and one pair of infinite roots ($n_{+\infty} = n_{-\infty} = 1$, allowed at $M = 9$). The descendant tower is as in Fig. 5, except that there is just a single pair of infinite roots. All intermediate descendants occur: the pair of infinite roots at $M = 3, 6, 9$ is allowed by (2.15) and (2.17). Each $M = 1$ vector thus sits at the top of a descendant tower containing one descendant at each of $M = 3, 7$ and two descendants at $M = 4, 6$.

The 45 states at $M = 2$ are also primitive, and must have finite Bethe roots. The corresponding states beyond the equator allow for two FM strings. By the free-fermion property each descendant tower contains four states.

Next we turn to $M = 3$, where $120 - 12 = 108$ primitive states remain. Here it starts to become a little more complicated to predict the structure by hand, as infinite Bethe roots ($n_{\pm\infty} = 1$) may occur. Due to parity invariance there must be equally many states in the classes (ii) and (iii). It is possible to find out how many such states occur. Indeed, suppose that $n_{+\infty} = 1$, $n_{-\infty} = 0$ (the opposite case is treated analogously). Then two Bethe roots remain to be determined. Their Bethe equations effectively acquire a twist due to the presence of the infinite root, cf. the discussion preceding (2.12). One can solve these Bethe equations, and delete all solutions that themselves contain infinite roots. Alternatively one can directly use the eigenvalues of the truncated two-parameter transfer matrix $\tilde{\mathbb{T}}(x, y, \phi)$ to compute the eigenvalues of the Q operator $\tilde{\mathbf{Q}}$ numerically. In either case, the corresponding states beyond the equator need four more Bethe roots. For the primitive states with $n_{+\infty} = n_{-\infty} = 0$ these extra roots come from $n_{+\infty} = n_{-\infty} = 2$, while for the primitive states with $n_{\pm\infty} = 1$ and $n_{\mp\infty} = 0$ these descendants have $n_{\pm\infty} = 2$, $n_{\mp\infty} = 0$ plus one FM string.

One proceeds analogously for the $210 - 12 = 198$ primitive states left at $M = 4$. The corresponding states beyond the equator need two more Bethe roots, which come from adding a pair $+\infty, -\infty$ for the primitive states in class (i). The situation is a bit more tricky for the primitive states in the other classes. For class (ii) they are of the form $|\{u_1, u_2, +\infty, +\infty\}\rangle$, but the corresponding states beyond the equator are only allowed to have one infinite root. In this case one of the infinite roots is removed to make place for an FM string: the descendants are $|\{u_1, u_2, \alpha^{\text{FM}}, +\infty\}\rangle$.

The $252 - 90 = 162$ primitive states left at the equator must fall in class (i).

The resulting spectrum is summarised in Table 1, where '$\times k$' counts the different FM string configurations (choices of $\alpha_n^{\text{FM}}$) for the descendant states at a given magnetisation. For example, the entry $1 \times 2 + 10$ comes from $|\{\alpha_n^{\text{FM}}\}\rangle$ for $n = 1, 2$ (descendants from $M = 0$) along with $|\{u_1, \pm\infty\}\rangle$ for the ten different Bethe roots $u_1$ at $M = 1$. The entry $1 + 10 \times 2$ similarly represents $|\{2 \times \pm\infty\}\rangle$ and $|\{u_1, \alpha_n^{\text{FM}}\}\rangle$ for $n = 1, 2$, respectively.

## 8.4 Deformations of FM strings

At the start of Section 7.3 we showed that FM strings occur precisely when the system is at root of unity with commensurate twist (6.15). Let us further substantiate this by studying what happens to the Bethe roots constituting an FM string under small changes to the twist $\phi$ or the anisotropy parameter $\eta$. (Observe that such deformations immediately break the relation (6.15), as $\ell_2$ fluctuates wildly when the value of $\eta$ varies through $i\mathbb{Q} \subset i\mathbb{R}$.)

Table 1: The full spectrum for $N = 10$, $\Delta = \frac{1}{2}$ ($\eta = \frac{i\pi}{3}$) and $\phi = 0$, organised in terms of primitive and descendant states with $n_{+\infty} = n_{-\infty}$, $n_{+\infty} > n_{-\infty} = 0$, $n_{-\infty} > n_{+\infty} = 0$.

| $M$ | total # | # primimitive states | | | # descendant states | | |
|---|---|---|---|---|---|---|---|
| | | $n_{\pm\infty} = 0$ | $n_{+\infty} > 0$ | $n_{-\infty} > 0$ | $n_{+\infty} = n_{-\infty}$ | $n_{+\infty} > 0$ | $n_{-\infty} > 0$ |
| 0 | 1 | 1 | 0 | 0 | 0 | 0 | 0 |
| 1 | 10 | 10 | 0 | 0 | 0 | 0 | 0 |
| 2 | 45 | 45 | 0 | 0 | 0 | 0 | 0 |
| 3 | 120 | 40 | 34 | 34 | $1 \times 2 + 10$ | 0 | 0 |
| 4 | 210 | 121 | 34 | 34 | $1 + 10 \times 2$ | 0 | 0 |
| 5 | 252 | 162 | 0 | 0 | $45 \times 2$ | 0 | 0 |
| 6 | 210 | 0 | 0 | 0 | $1 + 10 \times 2 + 121$ | 34 | 34 |
| 7 | 120 | 0 | 0 | 0 | $1 \times 2 + 10 + 40$ | 34 | 34 |
| 8 | 45 | 0 | 0 | 0 | 45 | 0 | 0 |
| 9 | 10 | 0 | 0 | 0 | 10 | 0 | 0 |
| 10 | 1 | 0 | 0 | 0 | 1 | 0 | 0 |

Turning on a small twist $\phi$ (Appendix E.1) leads to smooth changes of the Bethe roots associated to FM strings, and the same is true for the eigenvalues of the Q operator. This is closely related to the string hypothesis in the thermodynamic limit, as explained in Section 10.2: under small variations of the twist an FM string decomposes into (string deviations of) two ordinary Bethe strings with slightly different string centres. See also our conjecture in Section 10.2.

Instead, even a tiny change of $\eta$ away from the root-of-unity values completely changes the Bethe roots associated to the FM strings (Appendix E.2): unlike before, FM strings cannot be continuously reconstructed when $\eta$ is deformed, and the eigenvalues of the Q operator change drastically. This is expected from the representation theory of $U_q(\mathfrak{sl}_2)$, which is very sensitive to the precise root of unity. However, as one would expect on physical grounds, physical quantities such as the eigenvalues of the transfer matrices $\mathbf{T}_s$ *do* change continuously as $\eta$ is varied. Further investigations are required in order to fully understand this behaviour.

### 8.5 Connection to the work of Fabricius and McCoy

In [31] Fabricius and McCoy derived an equation for the string centre of FM strings at zero twist, see (1.11) therein, by linearising the Bethe equations (2.7) around an arbitrary root of unity. In other words, these string centres are obtained by continuity from their values in an infinitesimal neighbourhood of the root of unity. Let us explain why this is not in contradiction with the results from Section 8.4 and Appendix E.2.

By 'Bethe roots' we mean the zeroes of the eigenvalue of Q operator constructed from the two-parameter transfer matrix. We thus determine the FM string centres at commensurate twist (6.15) from the zeroes of the Q operator. Such zeroes associated with FM strings are among the zeroes of the Drinfeld polynomial (parameters of the evaluation representation). Note that the Drinfeld polynomial $Y(v)$ in (1.42) of Ref. [32], cf. (4.2) therein and [39], is proportional to $Q_s P_s$ in our notation, as can be seen by comparing (1.42) in [32] with our (7.15).

Next, our roots associated with FM strings do *not* satisfy (1.11) in [31]. This can most easily been seen when $N$ is a multiple of $\ell_2$ and $\ell_1$ is even (i.e. $L$ a multiple of $N$ and $r$ even in the notation of [31]), so that (1.11) from [31] reduces to (1.10) therein. In that case the denominator of $Y(v)$ is one, and the zeroes of $Y(v)$ coincide (up to a simple change of notation)

with the zeroes of (7.23), which we identify with the roots constituting the FM string. This would coincide with the solutions of [31] if one would ignore the second factor in parenthesis in (1.10).

The solutions to (1.11) in [31] can be used to construct the eigenvectors of **H**. Indeed, Fabricius and McCoy proposed a creation operator for FM strings, see (1.38) and (1.41)–(1.42) in [32]. In examples we find that, taking the spectral parameter therein to by a solution to (1.11) in [31], the result is an eigenvector of **H** and $\mathbf{T}_{1/2}$, but *not* of the two-parameter transfer matrix or Q operator. (Note that we cannot take the spectral parameter to be the FM string centres as found by our methods, cf. Section 8.5, since these are zeroes of $Y(v)$ which appears as a denominator in (1.38) from [32].) Instead, the FM string creation operator from [32] yields a linear combination of the eigenvectors of the Q operator. In Appendix E.2 we illustrate this with an example.

In the next section we propose creation and annihilation operators for FM strings that do yield eigenvectors of the Q operator.

# 9 Conjectures for FM creation and annihilation operators

Vernier *et al.* [36] proposed to use semicyclic representations in order to construct degenerate states with different magnetisation. These representations have also been used to construct quasilocal charges [77]. Building on these ideas we present conjectures for explicit constructions of the FM string creation and annihilation operators that commute with the XXZ transfer matrix while changing the magnetisation in steps of $\ell_2$.

## 9.1 Case $q^{\ell_2} = +1$

We start with roots of unity obeying $\varepsilon = q^{\ell_2} = +1$. Pick the semicyclic representation for the auxiliary space: $V_a^{\text{sc}}$ is $\ell_2$ dimensional, has spin $s \in \mathbb{C}$ and furthermore depends on a parameter $\beta \in \mathbb{C}$, see Appendix A.2. Replace $\mathbf{X}_a$ from (4.1), (6.3) by $\mathbf{X}_a^{\text{sc}} = \mathbf{X}_a + \beta \, x \, y / (x - y) |0\rangle\langle l - 1|_a$, which still obeys the commutation relation $\mathbf{W}_a \mathbf{X}_a^{\text{sc}} = q \mathbf{X}_a^{\text{sc}} \mathbf{W}_a$. Substituting this in (4.7) modifies the matrix entry $\mathbf{S}_a^+ = \mathbf{L}_a^{21} / \sinh \eta$ to

$$\mathbf{S}_a^{+,\text{sc}} = \mathbf{S}_a^+ + \beta \, |0\rangle\langle \ell_2 - 1|_a \, , \tag{9.1}$$

while keeping the other three matrix elements in (4.7) unchanged. In this way we obtain a Lax operator $\mathbf{L}_{aj}^{\text{sc}}$, and the usual construction (3.5)–(3.6) gives the transfer matrix

$$\mathbf{T}_s^{\text{sc}}(u, \phi, \beta) = \text{tr}_a \big[ \mathbf{L}_{aN}^{\text{sc}}(u, \beta) \cdots \mathbf{L}_{a1}^{\text{sc}}(u, \beta) \, \mathbf{E}_a(\phi) \big], \tag{9.2}$$

depending on $s \in \mathbb{C}$ and $\beta \in \mathbb{C}$. Here the twist $\mathbf{E}_a(\phi)$ acts on $V_a^{\text{sc}}$ by the usual expression (B.1). The matrix elements of (9.2) change the magnetisation $(N - 2M)/2$ by $-n\ell_2$ and are proportional to $\beta^n$ for positive $n \in \mathbb{Z}_{>0}$. For example, in a chain of length $N = \ell_2$, the expansion of the transfer matrix contains a term $\text{tr}_a[(\mathbf{S}_a^{+,\text{sc}})^N] \prod_{j=1}^N \sigma_j^-$ that changes the magnetisation by $-\ell_2$, creating $\ell_2$ magnon excitations.

The matrices $\mathbf{T}_s^{\text{sc}}(u, \phi)$ do *not* commute with each other at different values of $u$, but *do* commute with $\mathbf{T}_{1/2}(u, \phi)$ when the twist is commensurate. To see this, note that the construction of the semicyclic Lax operator guarantees that the following version of the RLL relation (3.3) with (six-vertex) R-matrix (3.4) holds true for any $\beta$:

$$\mathbf{R}_{jk}(u - v) \mathbf{L}_{aj}^{\text{sc}}(u, \beta) \mathbf{L}_{ak}^{\text{sc}}(v, \beta) = \mathbf{L}_{ak}^{\text{sc}}(v, \beta) \mathbf{L}_{aj}^{\text{sc}}(u, \beta) \mathbf{R}_{jk}(u - v). \tag{9.3}$$

This is an identity of operators on $V_j \otimes V_k \otimes V_a^{\text{sc}}$. Let us reinterpret it by thinking of the spin-1/2 space labelled by $k$ as an auxiliary space, $V_k \rightsquigarrow V_b$. The symmetry property $\mathbf{R}_{jk} = \mathbf{R}_{kj}$

of (3.4) allows us to view the R-matrix as a Lax matrix $\mathbf{L}_{bj}$, whereas $\mathbf{L}_{ak}^{\mathrm{sc}}$ takes the role of an R-matrix $\mathbf{R}_{ab}^{\mathrm{sc}}$. Reversing the two sides of (9.3) and changing $v \mapsto u - v$ we arrive at an RLL relation on $V_a^{\mathrm{sc}} \otimes V_b \otimes V_j$:

$$\mathbf{R}_{ab}^{\mathrm{sc}}(u-v,\beta)\mathbf{L}_{aj}^{\mathrm{sc}}(u,\beta)\mathbf{L}_{bj}(v) = \mathbf{L}_{bj}(v)\mathbf{L}_{aj}^{\mathrm{sc}}(u,\beta)\mathbf{R}_{ab}^{\mathrm{sc}}(u-v,\beta). \tag{9.4}$$

Here $a$ corresponds to the semicyclic auxiliary space, $b$ to a spin-1/2 auxiliary space, and $k$ to a spin-1/2 physical space. The train argument implies that the twisted semicyclic transfer matrix (9.2) commutes with the twisted fundamental transfer matrix (3.5)–(3.6), *provided* $\mathbf{R}_{ab}^{\mathrm{sc}}$ commutes with the tensor product of the twists matrices. This requires the twist to be commensurate, $e^{i\ell_2\phi} = 1$, cf. (6.15).

By fusion (9.2) further commutes with $\mathbf{T}_{s'}(u,\phi)$ for any $2s' \in \mathbb{Z}_{>0}$:

$$\left[\mathbf{T}_s^{\mathrm{sc}}(u,\phi,\beta), \mathbf{T}_{s'}(v,\phi)\right] = 0, \qquad s \in \mathbb{C}, \quad \beta \in \mathbb{C}, \quad 2s' \in \mathbb{Z}_{>0}. \tag{9.5}$$

Since the twisted semicyclic transfer matrix $\mathbf{T}_s^{\mathrm{sc}}(u,\phi,\beta)$ changes the magnetisation of an eigenstate of the Q operator in steps of $\ell_2$, it mixes states that are degenerate for $\mathbf{T}_{s'}(u,\phi)$ within each descendant tower.

One can use $\mathbf{T}_s^{\mathrm{sc}}(u,\phi,\beta)$ to construct eigenstates of the Q operator itself. Because the part of $\mathbf{T}_s^{\mathrm{sc}}(u,\phi,\beta)$ of first order in $\beta$ changes the magnetisation of eigenstates of the Q operator by $-\ell_2$, we make

**Conjecture 1.** When $\varepsilon = q^{\ell_2} = +1$ the linearisation in $\beta$ of (9.2) at $s = (\ell_2-1)/2$,

$$\mathbf{B}^{\mathrm{FM}}(u) := \partial_\beta \mathbf{T}_s^{\mathrm{sc}}(u,\phi,\beta)\big|_{\beta=0,\, 2s=\ell_2-1}, \tag{9.6}$$

is the creation operator for the FM string $\{\alpha^{\mathrm{FM}}\} = \{u, u + \frac{i\pi}{\ell_2}, \cdots, u + i\pi\frac{\ell_2-1}{\ell_2}\}$. The spectral parameter can be taken to be any Bethe root from the FM string, e.g. $u$ as in (9.6).

Note that at $\beta = 0$ and $2s = \ell_2 - 1$ the semicyclic Lax operator coincides with the Lax operator whose auxiliary space is the $\ell_2$-dimensional highest-weight representation. By (9.2) the operator (9.6) can thus be expressed more explicitly as

$$\mathbf{B}^{\mathrm{FM}}(u) = \sum_{j=1}^{N} \mathrm{tr}_a\left[\mathbf{L}_{aN}(u)\cdots\mathbf{L}_{a,j+1}(u)\, \mathbf{e}_a^{0,\ell_2-1}\sigma_j^-\, \mathbf{L}_{a,j-1}(u)\cdots\mathbf{L}_{a1}(u)\, \mathbf{E}_a(\phi)\right], \tag{9.7}$$

where we write $\mathbf{e}_a^{n,n'} = |n\rangle\langle n'|_a$ for the matrix units on $V_a$.

The construction of the FM-string creation operator (9.6) is just like that of the generating function of the quasilocal Y charges proposed in Ref. [77], except that the transfer matrices are evaluated at different values of $s$. Therefore $\mathbf{B}^{\mathrm{FM}}(u)$, like those Y charges, commutes with $\mathbf{T}_s(u)$ with $2s \in \mathbb{Z}_{>0}$ but not with $\tilde{\mathbf{T}}_{s'}(u)$ when $2s' \in \mathbb{C} \setminus \mathbb{Z}_{>0}$.

*Example.* We have verified our conjecture in many examples. To illustrate this consider $N = 6$, $\Delta = -\frac{1}{2}$ ($\eta = \frac{2i\pi}{3}$) and $\phi = 0$. The descendant tower of the pseudovacuum $|\{\varnothing\}\rangle = |\uparrow\uparrow\uparrow\uparrow\uparrow\uparrow\rangle$ contains three descendants:

$$|\{\alpha_1^{\mathrm{FM}}\}\rangle, \quad |\{\alpha_2^{\mathrm{FM}}\}\rangle, \quad |\{\alpha_1^{\mathrm{FM}}, \alpha_2^{\mathrm{FM}}\}\rangle, \tag{9.8}$$

with FM strings centred at

$$\alpha_1^{\mathrm{FM}} = -\frac{\log(10+3\sqrt{11})}{6}, \quad \alpha_2^{\mathrm{FM}} = +\frac{\log(10+3\sqrt{11})}{6}. \tag{9.9}$$

One can verify that (9.6) does indeed create these FM strings:

$$|\{\alpha_n^{\mathrm{FM}}\}\rangle \propto \mathbf{B}^{\mathrm{FM}}(\alpha_n^{\mathrm{FM}})|\{\varnothing\}\rangle, \quad n = 1,2, \tag{9.10}$$

and the consistency condition $|\{\alpha_1^{\mathrm{FM}}, \alpha_2^{\mathrm{FM}}\}\rangle \propto \mathbf{B}^{\mathrm{FM}}(\alpha_1^{\mathrm{FM}})|\{\alpha_2^{\mathrm{FM}}\}\rangle = \mathbf{B}^{\mathrm{FM}}(\alpha_2^{\mathrm{FM}})|\{\alpha_1^{\mathrm{FM}}\}\rangle$ holds, even though the $\mathbf{B}^{\mathrm{FM}}$ do not commute in general. More generally, whenever two FM strings with centres $\alpha_n^{\mathrm{FM}}, \alpha_{n'}^{\mathrm{FM}}$ occur among the descendants of some primitive state $|\{u_m\}_m\rangle$ we find that $\mathbf{B}^{\mathrm{FM}}(\alpha_n^{\mathrm{FM}})|\{u_m\}_m \cup \{\alpha_{n'}^{\mathrm{FM}}\}\rangle = \mathbf{B}^{\mathrm{FM}}(\alpha_{n'}^{\mathrm{FM}})|\{u_m\}_m \cup \{\alpha_n^{\mathrm{FM}}\}\rangle \propto |\{u_m\}_m \cup \{\alpha_n^{\mathrm{FM}}, \alpha_{n'}^{\mathrm{FM}}\}\rangle$. That is, our creation operator (9.6) can be used to construct the whole descendant tower described in Section 8.1.1.

Next we turn to the annihilation operator. Let us denote the parameter of the cyclic representation by $\gamma$. If we repeat the same construction with the transposed Lax matrix $\mathbf{L}_{aj}(\mathbf{v}_y, \mathbf{u}_x)$ from (4.9), this time replacing the entry $\hat{\mathbf{S}}_a^- := \mathbf{L}_a^{12}(\mathbf{v}_y, \mathbf{u}_x)/\sinh\eta$ by

$$\hat{\mathbf{S}}_a^{-,\mathrm{sc}} = \hat{\mathbf{S}}_a^- + \gamma \, |\ell_2 - 1\rangle\langle 0|_a \,, \tag{9.11}$$

we obtain a semicyclic transfer matrix

$$\hat{\mathbf{T}}_s^{\mathrm{sc}}(u, \phi, \gamma) = \mathrm{tr}_a\big[\hat{\mathbf{L}}_{aN}^{\mathrm{sc}}(u, \gamma) \cdots \hat{\mathbf{L}}_{a1}^{\mathrm{sc}}(u, \gamma) \, \mathbf{E}_a(\phi)\big] \tag{9.12}$$

that changes the magnetisation by positive multiples of $\ell_2$. Like before we arrive at

**Conjecture 2.** When $\varepsilon = q^{\ell_2} = +1$ the linearisation in $\gamma$ at $s = (\ell_2 - 1)/2$,

$$
\begin{aligned}
\mathbf{C}^{\mathrm{FM}}(u) &:= \partial_\gamma \, \hat{\mathbf{T}}_s^{\mathrm{sc}}(u, \phi, \gamma)\big|_{\gamma=0,\, 2s=\ell_2-1} \\
&= \sum_{j=1}^{N} \mathrm{tr}_a\big[\mathbf{L}_{aN}(u) \cdots \mathbf{L}_{a,j+1}(u) \, \mathbf{e}_a^{\ell_2-1,0} \sigma_j^+ \, \mathbf{L}_{a,j-1}(u) \cdots \mathbf{L}_{a1}(u) \, \mathbf{E}_a(\phi)\big],
\end{aligned} \tag{9.13}
$$

annihilates the FM string with Bethe roots $\{u, u + \frac{\mathrm{i}\pi}{\ell_2}, \cdots, u + \mathrm{i}\pi\frac{\ell_1-1}{\ell_2}\}$.

We have again checked this in many examples including the one above. We find that the consistency condition $\mathbf{C}^{\mathrm{FM}}(\alpha_n^{\mathrm{FM}}) \mathbf{B}^{\mathrm{FM}}(\alpha_{n'}^{\mathrm{FM}})|\{u_m\}_m \cup \{\alpha_n^{\mathrm{FM}}\}\rangle = |\{u_m\}_m \cup \{\alpha_{n'}^{\mathrm{FM}}\}\rangle = \mathbf{B}^{\mathrm{FM}}(\alpha_{n'}) \mathbf{C}^{\mathrm{FM}}(\alpha_n^{\mathrm{FM}})$ $|\{u_m\}_m \cup \{\alpha_n^{\mathrm{FM}}\}\rangle$ holds so long as $|\{u_m\}_m \cup \{\alpha_n^{\mathrm{FM}}\}\rangle$ and $|\{u_m\}_m \cup \{\alpha_{n'}^{\mathrm{FM}}\}\rangle$ are eigenstates of the Q operator, even though we do not know the commutation relations between $\mathbf{B}^{\mathrm{FM}}(u)$ and $\mathbf{C}^{\mathrm{FM}}(v)$ in general. We postpone such 'off-shell' relations to future investigations.

## 9.2 Case $q^{\ell_2} = -1$

When $\varepsilon = q^{\ell_2} = -1$ the RLL relation (9.4) has to be modified by including some signs:

$$\mathbf{R}_{ab}^{\mathrm{sc}}(u - v, \beta) \mathbf{L}_{aj}^{\mathrm{sc}}(u, -\beta) \mathbf{L}_{bj}(v) = \mathbf{L}_{bj}(v) \mathbf{L}_{aj}^{\mathrm{sc}}(u, \beta) \mathbf{R}_{aj}^{\mathrm{sc}}(u - v, -\beta). \tag{9.14}$$

This RLL relation can be shown by direct calculation on $V_b \otimes V_j$. The resulting monodromy matrix is 'staggered', with alternating sign of $\beta$. For general $\varepsilon = q^{\ell_2} = \pm 1$ the semicyclic monodromy matrix can thus be defined as

$$\mathbf{M}_a^{\mathrm{sc}}(u, s, \phi, \beta) := \mathbf{L}_{aN}^{\mathrm{sc}}(u, s, \varepsilon^N \beta) \cdots \mathbf{L}_{aj}^{\mathrm{sc}}(u, s, \varepsilon^j \beta) \cdots \mathbf{L}_{a1}^{\mathrm{sc}}(u, s, \varepsilon \beta) \, \mathbf{E}_a(\phi). \tag{9.15}$$

When $\varepsilon = +1$ all $\beta$s have the same sign, while for $\varepsilon = -1$ the sign alternates. Taking the trace we obtain the general expression for the semicyclic transfer matrix

$$\mathbf{T}_s^{\mathrm{sc}}(u, \phi, \beta) = \mathrm{tr}_a\big[\mathbf{L}_{aN}^{\mathrm{sc}}(u, s, \varepsilon^N \beta) \cdots \mathbf{L}_{aj}^{\mathrm{sc}}(u, s, \varepsilon^j \beta) \cdots \mathbf{L}_{a1}^{\mathrm{sc}}(u, s, \varepsilon \beta) \, \mathbf{E}_a(\phi)\big]. \tag{9.16}$$

For $\varepsilon = +1$ this reduces to (9.16); in particular, the use of the same notation as in (9.16) should not cause any confusion.

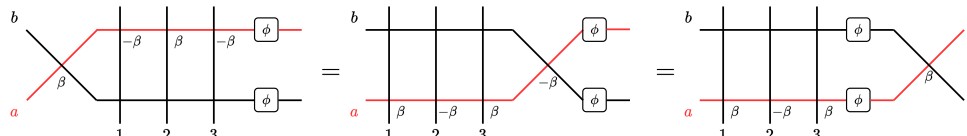

Figure 8: Graphical proof of Eq. (9.18) for an even number $N$ of sites.

Figure 9: Graphical proof of Eq. (9.18) for an odd number $N$ of sites.

Recall that for $\varepsilon = -1$ the commensurate twist $\phi$ depends on the system size $N$ through the condition $e^{i\ell_2\phi} = (-1)^N$ from (6.15). Therefore on $V_a^{sc} \otimes V_b$ we have

$$\mathbf{E}_a(\phi)\mathbf{E}_b(\phi)\mathbf{R}_{ab}^{sc}(u,\beta) = \mathbf{R}_{ab}^{sc}\big(u,(-1)^N\beta\big)\mathbf{E}_a(\phi)\mathbf{E}_b(\phi). \tag{9.17}$$

Combining (9.14) and (9.17) the train argument, illustrated in Figs. 8 and 9, yields

$$\mathbf{R}_{ab}^{sc}(u-v,\beta)\mathbf{M}_a^{sc}(u,s,\beta,\phi)\mathbf{M}_b(v,\phi) = \mathbf{M}_b(v,\phi)\mathbf{M}_a^{sc}(u,s,-\beta,\phi)\mathbf{R}_{ab}^{sc}(u-v,\beta). \tag{9.18}$$

Multiplying by $\mathbf{R}_{ab}^{sc}(u-v,\beta)^{-1} = \mathbf{R}_{ab}^{sc}(v-u+\eta,-\beta)/\sinh^2(u-v+\eta)$, provided it exists, and taking the trace over the $\ell_2$-dimensional auxiliary space we see that the semicyclic transfer matrix (9.16) commutes with the fundamental transfer matrix in the sense that

$$\mathbf{T}_s^{sc}(u,\beta,\phi)\mathbf{T}_{1/2}(v,\phi) = \mathbf{T}_{1/2}(v,\phi)\mathbf{T}_s^{sc}(u,-\beta,\phi), \qquad u,v,s \in \mathbb{C}. \tag{9.19}$$

With the aid of the fusion relation we obtain the commutation relations

$$\mathbf{T}_s^{sc}(u,\beta,\phi)\mathbf{T}_{s'}(v,\phi) = \mathbf{T}_{s'}(v,\phi)\mathbf{T}_s^{sc}\big(u,(-1)^{2s'}\beta,\phi\big), \qquad u,v,s \in \mathbb{C}, \quad 2s' \in \mathbb{Z}_{>0}. \tag{9.20}$$

In particular, the semicyclic transfer matrix commutes with $\mathbf{T}_{s'}$ for integer $s'$.

**Remark.** Another way to get this result is to evaluate the monodromy matrix $\mathbf{M}_{a'}^{sc}$ with an $2\ell_2$-dimensional semicyclic auxiliary space $V_{a'}$ when $\varepsilon = -1$. One can verify that $\mathrm{tr}_{a'}\mathbf{M}_{a'}^{sc} = \mathbf{T}_s^{sc}(u,\beta,\phi) + \mathbf{T}_s^{sc}(u,-\beta,\phi)$, while $(-1)^N \mathrm{tr}_{a'}\big[\mathbf{M}_{a'}^{sc}\mathbf{E}'\big] = \mathbf{T}_s^{sc}(u,\beta,\phi) - \mathbf{T}_s^{sc}(u,-\beta,\phi)$ if an extra twist $\mathbf{E}' = \sum_{n=0}^{2\ell_1-1}|n+\ell_2 \bmod 2\ell_2\rangle\langle n|_{a'}$ is included.

In either way we are led to

**Conjecture 3.** For any $\varepsilon = q^{\ell_2} = \pm 1$ the linearisation in $\beta$ of (9.16) at $s = (\ell_2 - 1)/2$,

$$
\begin{aligned}
\mathbf{B}^{FM}(u) &:= \partial_\beta \mathbf{T}_s^{sc}(u,\phi,\beta)\big|_{\beta=0,\,2s=\ell_2-1} \\
&= \sum_{j=1}^N \varepsilon^j \mathrm{tr}_a\big[\mathbf{L}_{aN}(u)\cdots\mathbf{L}_{a,j+1}(u)\,\mathbf{e}_a^{0,\ell_2-1}\sigma_j^-\,\mathbf{L}_{a,j-1}(u)\cdots\mathbf{L}_{a1}(u)\mathbf{E}_a(\phi)\big],
\end{aligned}
\tag{9.21}
$$

creates an FM string with Bethe roots $\{u', u' + \frac{i\pi}{\ell_2}, \cdots, u' + i\pi\frac{\ell_2-1}{\ell_2}\}$. The spectral parameter in (9.21) is related to the FM string by $u = u'$ if $\varepsilon = +1$ and $u = u' - \frac{i\pi}{2\ell_2}$ when $\varepsilon = -1$.

This conjecture has been successfully tested on all examples in Section 8, as well as for various cases when $\ell_2$ is even.

Let us focus on $\varepsilon = -1$ again. Taking the derivative of (9.20) with respect to the parameter $\beta$ we obtain the (anti)commutation relations

$$\begin{aligned} \left\{ \mathbf{B}^{\text{FM}}(u), \mathbf{T}_{s'}(v) \right\} &= 0, \qquad 2s' + 1 \in \mathbf{Z}_{>0}, \\ \left[ \mathbf{B}^{\text{FM}}(u), \mathbf{T}_{s'}(v) \right] &= 0, \qquad s' \in \mathbf{Z}_{>0}. \end{aligned} \tag{9.22}$$

The *anti*commutation of $\mathbf{B}^{\text{FM}}(u)$ with $\mathbf{T}_{1/2}(v)$ for $\varepsilon = -1$ might be surprising, but can be understood as follows. When $\varepsilon = -1$ an FM string adds $\pi$ to the momentum without affecting the energy or other charges. This requires $\mathbf{B}^{\text{FM}}(u)$ to anticommute with the translation operator, and to commute with all higher conserved charges (logarithmic derivatives of $\mathbf{T}_{1/2}$). This is guaranteed by the anticommutation.

The annihilation operator for FM strings can be defined like in (9.13). Generalise the semicyclic transfer matrix (9.12) to arbitrary $\varepsilon \in \{+1, -1\}$ by including staggered parameters $\pm \gamma$ like in (9.16). Then we propose

**Conjecture 4.** When $\varepsilon = q^{\ell_2} = \pm 1$ the operator

$$\begin{aligned} \mathbf{C}^{\text{FM}}(u) &:= \partial_\gamma \hat{\mathbf{T}}_s^{\text{sc}}(u, \phi, \gamma)\big|_{\gamma=0,\, 2s=\ell_2-1} \\ &= \sum_{j=1}^{N} \varepsilon^j \, \text{tr}_a\left[ \mathbf{L}_{aN}(u) \cdots \mathbf{L}_{a,j+1}(u) \, \mathbf{e}_a^{\ell_2-1,0} \sigma_j^+ \, \mathbf{L}_{a,j-1}(u) \cdots \mathbf{L}_{a1}(u) \, \mathbf{E}_a(\phi) \right] \end{aligned} \tag{9.23}$$

annihilates an FM string with Bethe roots $\{u', u' + \frac{i\pi}{\ell_2}, \cdots, u' + i\pi\frac{\ell_2-1}{\ell_2}\}$. The spectral parameter is again related to the FM string by $u = u'$ if $\varepsilon = 1$ and $u = u' - \frac{i\pi}{2\ell_2}$ for $\varepsilon = -1$.

## 10 Thermodynamic limit

### 10.1 FM strings and Z charges

In Section 7.2 we have seen that all states within a descendant tower have the same eigenvalues (up to a possible minus sign) for $\mathbf{T}_s$. This may lead one to wonder if those states can be distinguished at all using charges that are (quasi)local in the thermodynamic limit. The answer to this question is positive: the (exponentially many) degeneracies can be lifted by taking into account the quasilocal *Z charges* [42–45] that can be constructed at root of unity as logarithmic derivatives of the truncated transfer matrix $\tilde{\mathbf{T}}_s$ from Section 6. The quasilocality of the Z charges in the thermodynamic limit was demonstrated in Refs. [42–45]. The Z charges were used to study out-of-equilibrium phenomena such as quantum quenches in Ref. [41]. Here we focus on their role in distinguishing states that are degenerate for $\mathbf{T}_s$.

In Section 6 we already studied the key ingredient for the construction of the Z charges, viz. the truncated two-parameter transfer matrix $\tilde{\tilde{\mathbb{T}}}(x, y, \phi) = \tilde{\mathbf{T}}_s(u, \phi)$ at root of unity. Similar to Ref. [41] we define the generating function of the Z charges as

$$\mathbf{Z}(u, \phi) := -i \partial_s \log \tilde{\mathbf{T}}_s(u, \phi)\big|_{2s=\ell_2-1} = -i \tilde{\mathbf{T}}_{(\ell_2-1)/2}^{-1}(u, \phi) \, \partial_s \tilde{\mathbf{T}}_s(u, \phi)\big|_{2s=\ell_2-1}. \tag{10.1}$$

Like in (3.8) the Z charges are the coefficients of the expansion around $u = \frac{\eta}{2}$,

$$\mathbf{Z}^{(j)} := -i \frac{d^{j-1}}{du^{j-1}} \mathbf{Z}(u, \phi)\bigg|_{u=\eta/2}. \tag{10.2}$$

The Z charges are able to tell apart each member of the descendant tower. Let us illustrate this for the example $N = 6$, $\Delta = \frac{1}{2}$ ($\eta = \frac{i\pi}{3}$) and $\phi = 0$ from Section 2.3. Consider the descendant tower formed by

$$|\{\varnothing\}\rangle = |\uparrow\uparrow\uparrow\uparrow\uparrow\uparrow\rangle, \quad |\{\alpha_1^{\text{FM}}\}\rangle, \quad |\{\alpha_2^{\text{FM}}\}\rangle, \quad |\{\alpha_1^{\text{FM}}, \alpha_2^{\text{FM}}\}\rangle \propto |\downarrow\downarrow\downarrow\downarrow\downarrow\downarrow\rangle, \tag{10.3}$$

where the string centres for the two FM strings are

$$\alpha_1^{\text{FM}} = -\frac{\log(10 + 3\sqrt{11})}{6} + \frac{i\pi}{6}, \quad \alpha_2^{\text{FM}} = \frac{\log(10 + 3\sqrt{11})}{6} + \frac{i\pi}{6}. \tag{10.4}$$

The generating function (10.1) acts on the descendant tower by

$$\mathbf{Z}(u, 0)|\{\varnothing\}\rangle = \frac{12\pi \cosh(6u)}{10 - \cosh(6u)}|\{\varnothing\}\rangle,$$

$$\mathbf{Z}(u, 0)|\{\alpha_1^{\text{FM}}\}\rangle = \frac{6\sqrt{11}\pi}{10 - \cosh(6u)}|\{\alpha_1^{\text{FM}}\}\rangle,$$

$$\mathbf{Z}(u, 0)|\{\alpha_2^{\text{FM}}\}\rangle = -\frac{6\sqrt{11}\pi}{10 - \cosh(6u)}|\{\alpha_2^{\text{FM}}\}\rangle, \tag{10.5}$$

$$\mathbf{Z}(u, 0)|\{\alpha_1^{\text{FM}}, \alpha_2^{\text{FM}}\}\rangle = -\frac{12\pi \cosh(6u)}{10 - \cosh(6u)}|\{\alpha_1^{\text{FM}}, \alpha_2^{\text{FM}}\}\rangle.$$

The eigenvalues of $\mathbf{Z}(u)$ are different for each eigenstate.

Now consider the thermodynamic limit $N \to \infty$. There is a subtlety in the presence of FM strings. For instance, when $\phi = 0$ and $\eta = i\pi\ell_1/\ell_2$ with $\ell_1$ odd, (6.15) implies that for odd $N$ there are no FM strings in the spectrum, while for even $N$ there are (exponentially many) states associated with FM strings. The spectrum of systems at finite size is therefore sensitive to the parity of $N$, and it is not a priori clear whether the thermodynamic limit is well defined. However, numerics suggests that at odd $N$ there are states in the spectrum that differ by terms that vanish as $N \to \infty$ to give rise to the same asymptotic degeneracies as obtained when taking the limit via even $N$. Thus the result in the thermodynamic limit should be independent of the way in which the limit $N \to \infty$ is taken.

The role of the Z charges in separating the degeneracies in the presence of FM strings has important implications for the thermodynamic limit: Z charges have to be included when constructing the generalised Gibbs ensemble (GGE) for the XXZ spin chain at root of unity. Z charges are also important to obtain non-vanishing spin Drude weight with XXZ model at root of unity [42, 43]. This is further supported by the discussion in the remainder of this section.

## 10.2 TBA, string-charge duality, and a conjecture for string centres of FM strings

One of the cornerstones in the study of the thermodynamic properties of quantum integrable models is the thermodynamic Bethe ansatz (TBA), which has been used extensively to study out-of-equilibrium problems such as transport, quantum quenches and generalised hydrodynamics.

In order to solve the TBA equations for the XXZ spin chain at root of unity one relies on the *string hypothesis* [78], which stipulates the types of bound states that the model permits. This is a powerful tool that has yielded numerous results for thermodynamic properties of the XXZ model. A general review can be found in [79]. In the following we will use the notation of Ref. [46].

In [46], Ilievski *et al.* made the remarkable observation that the root densities of the Bethe strings (bound states) are in one-to-one correspondence with the generating functions of (quasi)local charges (3.8) of the model. This *string-charge duality* is of great use and convenient for the study of expectation values of (quasi)local charges, especially for quantum quenches [12]. The string-charge duality furthermore provides the root densities for the long-time non-equilibrium steady states [46, 80].

When we are at root of unity we have to take into account the generating function of the Z charges to obtain the root densities of the 'last two string types' in the list of Takahashi [78,

79]. This result has been derived in [41, 46].[12] Moreover, the total root densities of the 'last two string types' are the same from the TBA calculation (see Eq. (9.66) of [79]), revealing a deep connection between the two.

The 'last two string types' are related to FM strings too. In fact, two Bethe strings, one of each of the last two string types, that have coinciding real parts of their string centres (cf. Footnote 9 on p. 34) together form an FM string. For example, consider $\Delta = \frac{1}{2}$ ($\eta = \frac{i\pi}{3}$), for which the string hypothesis says that the last two string types are $(2, +)$ and $(1, -)$. Call a bound state with $n$ magnons whose Bethe roots have the same real part an '$n$-string'. Then '$(2, +)$' denotes a 2-string with even parity (centred at the real axis, i.e. a complex conjugate pair), and '$(1, -)$' a 1-string with odd parity (centred at $\mathrm{Im}\, u_m = \frac{\pi}{2} \equiv -\frac{\pi}{2}$). For $\eta = \frac{i\pi}{3}$ the imaginary parts of the 2-string with even parity are $\pm\frac{\pi}{6}$ while the imaginary part of the 1-string with odd parity is $\frac{\pi}{2}$. On the other hand, according to the examples in Section 8, FM strings for $\eta = \frac{i\pi}{3}$ can be expressed as

$$u_1 = \alpha - \frac{i\pi}{6}, \quad u_2 = \alpha + \frac{i\pi}{6}, \quad u_3 = \alpha + \frac{i\pi}{2}, \qquad \alpha \in \mathbb{R}. \tag{10.6}$$

But this can be viewed as two strings, one of type $(2, +)$ and one of type $(1, -)$, with coinciding real part of the string centre.

We expect that for any system size with commensurate twist FM strings can be decomposed in terms of the last two string types of the string hypothesis in this way, even at finite system size. Indeed, for any principal root of unity $\eta = \frac{i\pi}{\ell_2}$

the last two string types are $(\ell_2 - 1, +)$ and $(1, -)$ [78]. We have investigated FM strings for all $\ell_2 \leq 6$ at various $N$ and found that they can all be viewed as consisting of strings of the last two string types with equal real parts of the string centres. It even appears to hold for non-principal root of unity, see Appendix F for examples.

By studying the string centres of FM strings as the value of $\phi$ approaches the commensurate twist (6.15) as in Appendix E.1 we are led to

**Conjecture 5.** For any finite system size $N$ and root of unity $\eta = i\pi\frac{\ell_1}{\ell_2}$ with commensurate twist, any Fabricius–McCoy string

$$\alpha_k = \alpha^{\text{FM}} + \frac{i\pi}{\ell_2}\left(\frac{\ell_2 + 1}{2} - k\right), \quad 1 \leq k \leq \ell_2, \tag{10.7}$$

has string centre with imaginary part

$$\mathrm{Im}\, \alpha^{\text{FM}} = \begin{cases} \frac{\pi}{2\ell_2} & \text{if } \ell_1 \text{ is odd}, \\ 0 & \text{if } \ell_1 \text{ is even}. \end{cases} \tag{10.8}$$

This conjecture has been confirmed for all examples in Section 8.

In the thermodynamic limit, the conjecture (10.8) is compatible with the known TBA results [79]. Due to the exponentially large degeneracies of the descendant tower, FM strings in the thermodynamic limit will contribute to the thermodynamic quantities. Through conjecture (10.8) the density of FM strings has been included already in the root densities of the 'last two string types'. It implies that we do not need to include any new functional relation to the TBA formalism in [79].

Combining the conjecture with the knowledge that the generating function of the Z charges is crucial to obtain the root densities of 'last two string types', we conclude that the Z charges are directly related to the FM strings.

---

[12] In Ref. [41], the authors considered only principal root of unity ($\ell_1 = 1$). Similar results for arbitrary root of unity can be obtained analogously, cf. the supplementary material of [81].

**Remark.** There is a key difference between the thermodynamics of the 'descendant states' in our article and the $\mathfrak{sl}_2$-descendants in spin-1/2 Heisenberg XXX spin chain. For the latter the $\mathfrak{sl}_2$-descendant states are associated with Bethe roots at infinity (vanishing quasimomenta), and do not enter in the TBA calculations. However, for the XXZ spin chain at root of unity the 'descendant states' in our terminology (see Section 7.1) do enter the TBA calculation through the density of FM strings, which is in principle determined by the density of the last two string types.

### 10.3 FM strings and spin Drude weight

One of the most important physical consequences of the quasilocal Z charges is the non-vanishing high-temperature spin Drude weight[13] of the XXZ model at root of unity, due to the non-commutativity between the spin flip operator $\prod_j \sigma_j^x$ and the Z charges [42, 43, 45]. It can be considered as a manifestation of the exponentially many degeneracies in the thermodynamic limit.

In Appendix E.2 we demonstrate that perturbing the anisotropy parameter $\eta$ away from root of unity can change the structure of the Bethe roots dramatically, especially for states with FM strings. On the other hand, the spin Drude weight [45] is also known to change significantly under such perturbations. This hints at a connection between the existence of FM strings and the fractal structure of the spin Drude weight.

Another hint for the intimate relation between FM strings and non-vanishing spin Drude weight at root of unity comes from the domain-wall quench, i.e. the time evolution of an initial state $|\uparrow \cdots \uparrow\uparrow\downarrow\downarrow \cdots \downarrow\rangle$, for the XXZ model at root of unity. Here ballistic spin transport (non-vanishing spin Drude weight) can be treated analytically using generalised hydrodynamics [83]. The right half of the system, i.e. the fully polarised state $|\downarrow\downarrow \cdots \downarrow\rangle$, has Q function given in Section 7.5 for finite size. In the thermodynamic limit, according to the TBA, this fully polarised state consists of a filled 'Fermi sea' with Bethe roots of the last two string types [79, 83], cf. Section 10.2. In this case, each pair of Bethe strings with one string from each of the last two string types has the same real parts of the string centres. From the definition of the density of Bethe strings it follows that the densities of the last two string types must coincide. The conjecture (10.8) implies that the Bethe roots of $|\downarrow\downarrow \cdots \downarrow\rangle$ consists solely of FM strings. For the domain-wall quench the ballistic spin transport from the right half of the system is solely carried out by the quasilocal Z charges [83]. Notice that, even though the FM strings do not contribute to the dynamics, the 'FM strings' of the right half of the system are not true excitations from the perspective of the whole system. Thus they *do* contribute to the dynamics, as combinations of Bethe strings of the last two string types, resulting in the domain-wall melting phenomenon.

## 11 Conclusion

We have studied the full spectrum of the transfer matrices associated to the quantum spin-$\frac{1}{2}$ Heisenberg XXZ chain, focussing on root of unity with arbitrary twist. To this end we constructed the Baxter's Q operator, and the P operator, from the factorisation of a two-parameter transfer matrix (4.25). The eigenvalues of the Q operator, i.e. Q functions, are polynomials whose zeroes encode the physical solutions of the Bethe equations (2.7).

As a by-product of our construction we rederived the matrix TQ relation (5.10) and transfer-matrix fusion relations (5.12) from a decomposition of the two-parameter transfer matrix, pro-

---

[13] The full frequency-dependent conductivity $\sigma(\omega)$, which would be of greater experimental relevance than the Drude weight, remains challenging to compute [82].

viding a simplification of the conventional approach. At root of unity we further derived truncated transfer-matrix fusion and Wronskian relations from the two-parameter transfer matrix with auxiliary space truncated to a finite-dimensional space. We also proved an interpolation-type formula conjectured in Refs. [32, 34, 41].

Equipped with these algebraic tools we obtained analytic results about the spectrum at root of unity (Section 7). These results enabled us to demonstrate the presence of descendant towers for the XXZ model at root of unity with commensurate twist, for with explicit examples for the various scenarios that occur (Section 8). We elucidated the exponential growth of the degeneracies at root of unity. Since we can construct the Q operator explicitly at root of unity, via a trace over a finite-dimensional auxiliary space, we obtained analytic results for rather large system size compared to previous works. From the quantisation condition (7.15) we found that FM strings associated with the descendant states behave like free fermions within each descendant tower (Section 8.1.1). We have found new semicyclic transfer matrices that satisfy unconventional RLL relations, see e.g. (9.18), from which we conjectured an explicit expression for the creation and annihilation operators of FM strings (Section 9).

Even though our main results and discussions concentrate on systems with finitely many sites, we moreover compared our results with recent works on the thermodynamic limit (Section 10). We explained the relation between the truncated two-parameter transfer matrices and the quasilocal Z charges, which are of crucial importance in many applications such as quantum quenches and spin transport at root of unity. Inspired by the string-charge duality we found a connection between the last two string types à la Takahashi, the Z charge and the FM strings in the XXZ model at root of unity. This led us to a conjecture about the imaginary part of the string centres of FM strings based on the string hypothesis.

**Outlook.**  Several interesting aspects remain to be explored. First of all we hope to apply the construction of the two-parameter transfer matrix to other quantum integrable models, such as the XXZ spin chain with other boundary conditions as well as their higher-spin generalisations. It is not known how the construction should be generalised to integrable models with higher-rank symmetry, where there exist several Q operators.

We would like to understand more properties of the FM strings in the thermodynamic limit, e.g. whether quasilocality of the Z charges has a deep relation to FM strings (cf. Section 10). In addition, the semiclassical limit of the domain-wall quench in the gapless regime yields a similar ballistic spin transport behaviour [84, 85] to the quantum counterpart [83]. However, it is not clear what the semiclassical limit of the Z charges would be since the classical Landau–Lifshitz field theory does not have an underlying quantum-group structure. It would be desirable to clarify the relation between the mechanisms in both quantum and classical regime when a qualitative classical-quantum correspondence of spin transport [84] applies.

The structure of descendant towers from Section 8 implies the existence of a hidden Onsager algebra at least for a part of the physical Hilbert space [86]. At the free-fermion point $\Delta = 0$, and at $\eta = \frac{i\pi}{3}$ for the integrable spin-1 XXZ chain, the Onsager algebra plays a crucial role. The properties of the Onsager algebra may allow one to obtain results like in Ref. [36]. The relation between the Onsager generators and the semicyclic transfer matrices from Section 9 needs to be elucidated. In particular we would like to discover algebraic structures like those in [36] in the extensively studied, yet not completely understood, XXZ spin chain.

The full spectrum of the XXZ model at root of unity has potential physical applications. For instance, equipped with our results, it would be interesting to calculate the partition function of the six-vertex model at root of unity analytically using algebraic geometry, following recent works on the isotropic case [87, 88]. Furthermore, other physically relevant quantities such as the density matrix and overlaps with a generic state can be extracted from the full spectrum at finite system size, which could shed light on the studies of quantum entanglement and

non-equilibrium dynamics in exact solvable models.

# Acknowledgements

We are indebted to Jean-Sébastien Caux for collaboration in the early stages of this work and for discussions throughout the course of this work. Y.M. thanks Paul Fendley, Etienne Granet, Enej Ilievski, Tomaž Prosen, Eyzo Stouten and Eric Vernier for useful discussions and correspondences. V.P. thanks Olivier Babelon and Alexandre Lazarescu for illuminating discussions. We thank the referees for their comments.

Y.M. acknowledges the hospitality of Institut Henri Poincaré during the workshop "Systems out of equilibrium", where part of the work has been conducted. Y.M. acknowledges support from the European Research Council under ERC Advanced grant 743032 DYNAMINT. J.L. acknowledges the Australian Research Council Centre of Excellence for Mathematical and Statistical Frontiers (ACEMS) for financial support.

# A   Quantum $\mathfrak{sl}_2$

The quantum group $U_q(\mathfrak{sl}_2)$ is the unital associative algebra with generators $\mathbf{S}^+, \mathbf{S}^-$ along with $\mathbf{K}$ (which is invertible) subject to the commutation relations

$$\mathbf{K}\mathbf{S}^{\pm}\mathbf{K}^{-1} = q^{\pm 1}\,\mathbf{S}^{\pm}\,, \qquad \left[\mathbf{S}^+, \mathbf{S}^-\right] = \frac{\mathbf{K}^2 - \mathbf{K}^{-2}}{q - q^{-1}}\,. \tag{A.1}$$

We take the coproduct to be $\mathbf{S}^{\pm} \mapsto \mathbf{S}^{\pm} \otimes \mathbf{K}^{-1} + \mathbf{K} \otimes \mathbf{S}^{\pm}$ and $\mathbf{K} \mapsto \mathbf{K} \otimes \mathbf{K}$. There is a counit and antipode, see e.g. Eqs. (1.2)–(1.4) in Ref. [47]. A good (but hard to find) introduction is [89].

In this appendix we summarise the representations of $U_q(\mathfrak{sl}_2)$ that we will use. For each case it is easy to check that the commutation relations (A.1) hold and that $\mathbf{K} = \exp(\eta\,\mathbf{S}^z)$. The spin of an irrep is defined by the eigenvalue $[s]_q\,[s+1]_q$ of the quantum Casimir operator

$$\frac{1}{2}\big(\mathbf{S}^+\mathbf{S}^- + \mathbf{S}^-\mathbf{S}^+\big) + \frac{[2]_q}{2}\bigg(\frac{\mathbf{K} - \mathbf{K}^{-1}}{q - q^{-1}}\bigg)^2\,, \tag{A.2}$$

which generates the centre of $U_q(\mathfrak{sl}_2)$.

## A.1   Global representation

The physical Hilbert space $(\mathbb{C}^2)^{\otimes N}$ of the spin chain comes with two 'global' representations. When $N = 1$ the representation is given by $\mathbf{S}^{\pm} = \sigma^{\pm}$ and $\mathbf{K} = q^{\sigma^z/2}$. For $N > 2$ repeated application of the coproduct gives the (reducible) representation

$$\begin{aligned}
\mathbf{S}^{\pm} &= \sum_{j=1}^{N} q^{\sigma_1^z/2} \otimes \cdots \otimes q^{\sigma_{j-1}^z/2} \otimes \sigma_j^{\pm} \otimes q^{-\sigma_{j+1}^z/2} \otimes \cdots \otimes q^{-\sigma_N^z/2}\,, \\
\mathbf{K} &= q^{\mathbf{S}^z} = q^{\sigma_1^z/2} \otimes q^{\sigma_2^z/2} \otimes \cdots \otimes q^{\sigma_N^z/2}\,,
\end{aligned} \tag{A.3}$$

from (2.5). By reversing the factors (taking the opposite coproduct) we obtain another (reducible) representation:

$$\begin{aligned}
\bar{\mathbf{S}}^{\pm} &= \sum_{j=1}^{N} q^{-\sigma_1^z/2} \otimes \cdots \otimes q^{-\sigma_{j-1}^z/2} \otimes \sigma_j^{\pm} \otimes q^{\sigma_{j+1}^z/2} \otimes \cdots \otimes q^{\sigma_N^z/2}\,, \\
\bar{\mathbf{K}} &= q^{\bar{\mathbf{S}}^z} = q^{\sigma_1^z/2} \otimes q^{\sigma_2^z/2} \otimes \cdots \otimes q^{\sigma_N^z/2} = \mathbf{K}\,.
\end{aligned} \tag{A.4}$$

All of these operators can be obtained from the entries (3.9) of the monodromy matrix is the ('braid') limits $u \to \pm\infty$. In particular, the **B** operator from the QISM is closely related to the above spin-lowering generators. To see this we write the Lax operator (3.2) with spin-$\frac{1}{2}$ auxiliary space (see Footnote 4 on p. 10) in the form

$$\mathbf{L}_{aj}(u) = \begin{pmatrix} \mathbf{a}_j(u) & \mathbf{b}_j(u) \\ \mathbf{c}_j(u) & \mathbf{d}_j(u) \end{pmatrix}_a, \quad \mathbf{b}_j(u) = \sinh(\eta)\,\sigma_j^-, \quad \mathbf{c}_j(u) = \sinh(\eta)\,\sigma_j^+. \tag{A.5}$$

In the ('braid') limits $u \to \pm\infty$ the diagonal entries behave as

$$\mathbf{a}_j(u) \sim \pm\frac{e^{\pm u}}{2}\,q^{\sigma_j^z/2}, \qquad \mathbf{d}_j(u) \sim \pm\frac{e^{\pm u}}{2}\,q^{-\sigma_j^z/2}, \qquad u \to \pm\infty. \tag{A.6}$$

Therefore, using the definition of monodromy matrix (3.9), we have

$$\begin{aligned} \lim_{u \to -\infty} (-2\,e^u)^{N-1}\,\mathbf{B}(u) &= \sinh(\eta)\,\mathbf{S}^-, \\ \lim_{u \to +\infty} (2\,e^{-u})^{N-1}\,\mathbf{B}(u) &= \sinh(\eta)\,\bar{\mathbf{S}}^-. \end{aligned} \tag{A.7}$$

One similarly recovers the spin-raising operators from **C**, and $\mathbf{K}, \mathbf{K}^{-1}$ from either of $\mathbf{A}, \mathbf{D}$.

## A.2 Auxiliary representations

In this article we use various choices for the auxiliary space, which is always a representation of $U_q(\mathfrak{sl}_2)$. We summarise the key ingredient in this appendix.

First of all we use the finite-dimensional unitary spin-$s$ representation of $U_q(\mathfrak{sl}_2)$. Denote the (orthonormal) basis of $\mathbb{C}^{2s+1}$ by $|n\rangle$ for $n = 0, 1, \cdots, 2s$. Then the generators are given by

$$\begin{aligned} \mathbf{S}^+ &= \sum_{n=0}^{2s-1} \sqrt{[2s-n]_q[n+1]_q}\,|n+1\rangle\langle n|, \\ \mathbf{S}^- &= \sum_{n=0}^{2s-1} \sqrt{[2s-n]_q[n+1]_q}\,|n\rangle\langle n+1|, \qquad 2s \in \mathbb{Z}_{\geq 0}. \\ \mathbf{K} &= \sum_{n=0}^{2s} q^{-s+n}\,|n\rangle\langle n|, \qquad \mathbf{S}^z = \sum_{n=0}^{2s} (-s+n)\,|n\rangle\langle n|, \end{aligned} \tag{A.8}$$

When $s = 1/2$ this gives the case $N = 1$ of (A.3) when we identify $|1\rangle = |\uparrow\rangle$ and $|0\rangle = |\downarrow\rangle$. Note that we are thus led to a *decreasing* ordering of the basis: we prefer conventions such that $\mathbf{S}^\pm|n\rangle \propto |n \pm 1\rangle$ and the basis is labelled by non-negative integers; then the matrices match the standard choice provided we order the basis as $|2s\rangle, |2s-1\rangle, \cdots, |0\rangle$.

Next we use the complex spin-$s$ highest weight representation of $U_q(\mathfrak{sl}_2)$. It is defined on an infinite-dimensional Hilbert space with orthonormal basis $\langle n|$ indexed by $n \in \mathbb{Z}_{\geq 0}$. The generators are given by

$$\begin{aligned} \mathbf{S}^+ &= \sum_{n=0}^{\infty} [2s-n]_q\,|n+1\rangle\langle n|, \\ \mathbf{S}^- &= \sum_{n=0}^{\infty} [n+1]_q\,|n\rangle\langle n+1|, \qquad 2s \in \mathbb{C}. \\ \mathbf{K} &= \sum_{n=0}^{\infty} q^{-s+n}\,|n\rangle\langle n|, \qquad \mathbf{S}^z = \sum_{n=0}^{\infty} (-s+n)\,|n\rangle\langle n|, \end{aligned} \tag{A.9}$$

This is related to the transfer matrices $\mathbf{T}_s^{\mathrm{hw}}$ (4.19). In Figs. 2 and 3 we write down explicit matrices; let us stress that — albeit perhaps somewhat unusual in an infinite-dimensional context — as above we take the basis to be ordered *de*creasingly: $\cdots, |2\rangle, |1\rangle, |0\rangle$. When $s \in \frac{1}{2}\mathbb{Z}_{\geq 0}$ the infinite-dimensional highest-weight representation contains a finite-dimensional submodule. Indeed, the entry of $\mathbf{S}^+$ for $n = 2s$ vanishes (cf. Fig. 2), so the subspace labelled by $0 \leq n \leq 2s$ is preserved by all of (A.9). Thus we can truncate to a representation of dimension $2s + 1$ with generators

$$
\begin{aligned}
\mathbf{S}^+ &= \sum_{n=0}^{2s-1} [2s-n]_q \, |n+1\rangle\langle n|, \\
\mathbf{S}^- &= \sum_{n=0}^{2s-1} [n+1]_q \, |n\rangle\langle n+1|, && 2s \in \mathbb{Z}_{\geq 0}. && \text{(A.10)} \\
\mathbf{K} &= \sum_{n=0}^{2s} q^{-s+n} \, |n\rangle\langle n|, && \mathbf{S}^z = \sum_{n=0}^{2s} (-s+n) \, |n\rangle\langle n|,
\end{aligned}
$$

This representation is equivalent to (A.8) by a gauge transformation (conjugation).

More importantly, when $\eta = \mathrm{i}\pi\ell_1/\ell_2$ there exists another truncation yielding an $\ell_2$-dimensional representation, illustrated in Fig. 3. This is sometimes referred to as a *nilpotent* representation, due to the fact that $(\mathbf{S}^\pm)^{\ell_2} = 0$ in this case. The generators act on the subspace with basis $|n\rangle$ for $0 \leq n \leq \ell_2 - 1$ by

$$
\begin{aligned}
\mathbf{S}^+ &= \sum_{n=0}^{\ell_2-2} [2s-n]_q \, |n+1\rangle\langle n|, \\
\mathbf{S}^- &= \sum_{n=0}^{\ell_2-2} [n+1]_q \, |n\rangle\langle n+1|, && 2s \in \mathbb{C}. && \text{(A.11)} \\
\mathbf{K} &= \sum_{n=0}^{\ell_2-1} q^{-s+n} \, |n\rangle\langle n|, && \mathbf{S}^z = \sum_{n=0}^{\ell_2-1} (-s+n) \, |n\rangle\langle n|,
\end{aligned}
$$

There is one more truncated $\ell_2$-dimensional representation that we will use at root of unity: the *semicyclic* representation. It is similar to the truncated highest-weight representation (A.11) with an additional entry in $\mathbf{S}^+$:

$$
\begin{aligned}
\mathbf{S}^+ &= \beta \, |0\rangle\langle \ell_2 - 1| + \sum_{n=0}^{\ell_2-2} [2s-n]_q \, |n+1\rangle\langle n|, \\
\mathbf{S}^- &= \sum_{n=0}^{\ell_2-2} [n+1]_q \, |n\rangle\langle n+1|, && \begin{aligned} 2s &\in \mathbb{C}, \\ \beta &\in \mathbb{C}. \end{aligned} && \text{(A.12)} \\
\mathbf{K} &= \sum_{n=0}^{\ell_2-1} q^{-s+n} \, |n\rangle\langle n|, && \mathbf{S}^z = \sum_{n=0}^{\ell_2-1} (-s+n) \, |n\rangle\langle n|,
\end{aligned}
$$

# B  Quasiperiodicity

## B.1  Twist operator

We define the twist operator $\mathbf{E}_a(\phi)$ for the auxiliary space. Each of the $U_q(\mathfrak{sl}_2)$ representations on the auxiliary space from Appendix A.2 is expressed in terms of an orthonormal basis

$\{|d-1\rangle, \cdots, |1\rangle, |0\rangle\}$ with $d$ the dimension of the representation. Here $d = 2s+1$ for the unitary spin-$s$ representation with $s \in \frac{1}{2}\mathbb{Z}$, $d = \infty$ for the highest-weight representation with $s \in \mathbb{C}$, and $d = \ell_2$ for the truncation at root of unity. We consider diagonal twist operator $\mathbf{E}_a(\phi)$ given by

$$\mathbf{E}_a(\phi) = \sum_{n=0}^{d-1} e^{\mathrm{i}\phi n} |n\rangle\langle n|_a. \tag{B.1}$$

In view of our ordering of the basis this yields the twist from (3.9) for $s = 1/2$ ($d = 2$).

In particular, the complex spin-$s$ representation yields monodromy matrix $\mathbf{M}_s^{\mathrm{hw}}$

$$\mathbf{M}_s^{\mathrm{hw}}(u, \phi) = \mathbf{L}_{sN}(u) \cdots \mathbf{L}_{s2}(u) \mathbf{L}_{s1}(u) \mathbf{E}_s^{\mathrm{hw}}(\phi), \qquad \mathbf{E}_s^{\mathrm{hw}}(\phi) = \sum_{n=0}^{\infty} e^{\mathrm{i}\phi n} |n\rangle\langle n|_s, \tag{B.2}$$

resulting in the transfer matrix $\mathbf{T}_s^{\mathrm{hw}}(u, \phi) = \mathrm{tr}_s \mathbf{M}_s^{\mathrm{hw}}(u, \phi)$. When $|q| \le 1$ the diagonal matrix elements of $\mathbf{M}_s^{\mathrm{hw}}$ can be bounded by $A_N |q^N e^{\mathrm{i}\phi}|^n$ for some constant $A_N$, and so the trace is convergent if $|e^{\mathrm{i}\phi}| < |q|^N$. At this point we do not know if $\mathbf{T}_s^{\mathrm{hw}}$ can be analytically continued outside this disc of convergence.

The truncation at root of unity $\eta = \mathrm{i}\pi\ell_1/\ell_2$ likewise has

$$\tilde{\mathbf{M}}_s(u, \phi) = \mathbf{L}_{sN}(u) \cdots \mathbf{L}_{s2}(u) \mathbf{L}_{s1}(u) \tilde{\mathbf{E}}_s(\phi), \qquad \tilde{\mathbf{E}}_s(\phi) = \sum_{n=0}^{\ell_2-1} e^{\mathrm{i}\phi n} |n\rangle\langle n|_s, \tag{B.3}$$

and transfer matrix $\tilde{\mathbf{T}}_s(u, \phi) = \mathrm{tr}_s \tilde{\mathbf{M}}_s(u, \phi)$. In this case the trace is well defined for any value of of the twist $\phi$.

## B.2 Twisted momentum and magnons

Let us compute the first few charges (3.8) generated by the (six-vertex) transfer matrix with auxiliary space $V_a \cong \mathbb{C}^2$ of spin $\frac{1}{2}$ and twist $\mathbf{E}(\phi) = \mathrm{diag}(e^{\mathrm{i}\phi}, 1)$ as in (3.9)–(3.10). In the periodic case ($\phi = 0$) we get the cyclic (right) translation operator $\mathbf{G}(0)$,

$$\mathbf{T}(\eta/2, 0) = \sinh^N \eta \, \mathbf{G}(0), \qquad \mathbf{G}(0) = \mathbf{P}_{12\cdots N} = \mathbf{P}_{12}\mathbf{P}_{23}\cdots\mathbf{P}_{N-1,N}, \tag{B.4}$$

so the charge $\mathbf{I}^{(1)} = -\mathrm{i}\log\mathbf{G}$ is the usual momentum operator. In the quasiperiodic case the twist deforms the translation operator to

$$\mathbf{T}(\eta/2, \phi) = \sinh^N \eta \, \mathbf{G}(\phi), \qquad \mathbf{G}(\phi) = e^{\mathrm{i}\phi(\sigma_1^z+1)/2} \mathbf{G}(0) = \mathbf{G}(0) e^{\mathrm{i}\phi(\sigma_N^z+1)/2}. \tag{B.5}$$

In analogy with the periodic case write its eigenvalue as $e^{\mathrm{i}p}$; we will get back to the 'twisted momentum' $p$ soon. The next charge is the XXZ Hamiltonian (2.1) up to some constants:

$$\mathbf{I}^{(2)} = -\mathrm{i}\mathbf{T}(\eta/2, \phi)^{-1}\mathbf{T}'(\eta/2, \phi) = -\frac{2\mathrm{i}}{\sinh\eta}\left(\mathbf{H} + \frac{N\Delta}{2}\right), \tag{B.6}$$

where the prime denotes the derivative with respect to the first argument. The twist spoils homogeneity in the traditional sense: for $\phi \ne 0$ (B.6) does not commute with (B.4). However, (3.7) guarantees that (B.6) is 'twisted homogeneous' in that it commutes with (B.5). For fun let us show that this immediately gives the eigenvectors for $M = 1$.

The twisted translation (B.5) shows the quasiperiodic boundary conditions by $\mathbf{G}(\phi)^N = \exp[\mathrm{i}\phi(\mathbf{S}^z + N/2)]$. Since the transfer matrix commutes with $\mathbf{S}^z$ the 'improved' translation operator $\mathbf{G}_\star(\phi) := \exp[-\mathrm{i}\phi(\mathbf{S}^z + N/2)/N]\mathbf{G}(\phi) = \mathbf{G}(\phi)\exp[-\mathrm{i}\phi(\mathbf{S}^z + N/2)/N]$ still commutes with the Hamiltonian. The 'improved momentum' is $p_\star = p - (1 - M/N)\phi$. Formally this setting provides a fully translationally-invariant, and in particular periodic, structure:

$\mathbf{G}_\star(\phi)\,\mathbf{H}\,\mathbf{G}_\star(\phi)^{-1} = \mathbf{H}$ and $\mathbf{G}_\star(\phi)^N = \mathbb{1}$. For $M = 1$ it is easy to build the eigenvectors of $\mathbf{G}_\star$ explicitly (see Lemma 1.14 in [90] for a variant of this construction): the cyclicity of $\mathbf{G}_\star(\phi)$ ensures that with $|\Omega\rangle = |\uparrow\uparrow\cdots\uparrow\rangle$ the vector

$$\sum_{j=1}^{N} e^{\mathrm{i}p_\star j}\,\mathbf{G}_\star(\phi)^{1-j}\,\sigma_N^-\,|\Omega\rangle = e^{\mathrm{i}(p_\star-\phi)}\sum_{j=1}^{N} e^{-\mathrm{i}(p_\star-\phi/N)j}\,\sigma_j^-\,|\Omega\rangle \tag{B.7}$$

has $\mathbf{G}_\star(\phi)$-eigenvalue $e^{\mathrm{i}p_\star}$ where $p_\star = 2\pi k/N$, $0 \le k \le N-1$, is quantised. When $\phi$ is real $\mathbf{G}_\star(\phi)$ is unitary so the twisted magnons (B.7) are linearly independent and form a basis for the $M = 1$ sector, so they are also eigenvectors of (B.5) and (B.6). In terms of $\widetilde{p} := (2\pi k - \phi)/N$ the (twisted) momentum is $p = \widetilde{p} + \phi/N$, the dispersion is $\varepsilon = \cos(\widetilde{p}) - \Delta$, and the right-hand side of (B.7) looks just like an ordinary magnon, $\sum_j e^{-\mathrm{i}\widetilde{p}j}\,\sigma_j^-\,|\Omega\rangle$. (The sign in the exponential is because we work with the *right* translation operator.)

Let us finally make contact with the algebraic Bethe ansatz: for $M = 1$ (3.11) gives

$$|\{\nu\}\rangle = \mathbf{B}(\nu)\,|\Omega\rangle = \frac{\sinh\eta}{\sinh(\nu - \eta/2)}\,\sinh^N(\nu + \eta/2)\sum_{j=1}^{N} e^{-\mathrm{i}\widetilde{p}j}\,\sigma_j^-\,|\Omega\rangle, \tag{B.8}$$

where the quasimomentum

$$\widetilde{p} = \mathrm{i}\log\frac{\sinh(\nu - \eta/2)}{\sinh(\nu + \eta/2)} = \frac{2\pi k - \phi}{N} \tag{B.9}$$

solves the Bethe equation $e^{\mathrm{i}N\widetilde{p}} = e^{-\mathrm{i}\phi}$. The (twisted) momentum and energy given above match the result (2.8)–(2.9) obtained from (3.12) using (B.5) and (B.6). The difference between quasimomentum and (twisted) momentum for $M = 1$ can be avoided by taking $\widetilde{\mathbf{G}} := e^{-\mathrm{i}\phi}\,\mathbf{G}$ to be the twisted translation.

# C  Bethe roots

## C.1  Numerical recipe for finding Bethe roots

Here we review a numerical recipe, called McCoy's method, for solving the functional TQ relation. It appears to have been published first in [91]. We follow the description of Haldane [92]. A similar method was also described by Baxter [20].

The idea is that rather than solve the (coupled, nonlinear) Bethe equations one can obtain the Bethe roots by solving a few sets of linear equations. This is done by exploiting the known form of the eigenvalues of the (fundamental) transfer matrix and the Q operator, whose zeroes are the Bethe roots (see Section 5.5). The recipe goes as follows:

   i. Construct the transfer matrix $\mathbf{T}_{1/2}(u)$ at some $u \in \mathbb{C}$ (almost any value will do), and numerically diagonalise it; this is much more efficient than diagonalisation when $u$ is kept free. One obtains $2^N$ eigenstates that span the physical Hilbert space. From the Bethe ansatz we know that the eigenvectors are independent of the spectral parameter, so these will be eigenvectors for $\mathbf{T}_{1/2}(u)$ for any $u$.

   ii. The eigenvalues, however, depend on $u$. Pick one of the eigenvectors. Its eigenvalue is found by acting with $\mathbf{T}_{1/2}(u)$. This may again be done numerically by writing the eigenvalue as a Laurent polynomial in $t = e^u$ of order $N$,

$$T_{1/2}(t) = \mathrm{cst}_T \prod_{n=1}^{N} (\tau_n^{-1}\,t - \tau_n\,t^{-1}), \tag{C.1}$$

with zeroes $\tau_n$ that can be fixed by acting with $\mathbf{T}_{1/2}(u_n)$ for $N$ distinct values $u_n \in \mathbb{C}$.

iii. The corresponding Bethe roots are the zeroes of the Q operator, found by solving the functional TQ relation (5.19), i.e.

$$T_{1/2}(u, \phi) Q(u, \phi) = T_0(u - \eta/2) Q(u + \eta, \phi) + e^{i\phi} T_0(u + \eta/2) Q(u - \eta, \phi). \quad \text{(C.2)}$$

Here $T_0(u) = \sinh^N(u)$ and the eigenvalues are of the form

$$Q(t) = \text{cst}_Q \prod_{m=1}^{M} (t_m^{-1} t - t_m t^{-1}), \quad \text{(C.3)}$$

where $M$ is the number of down spins of the eigenvector under consideration. The zeroes $t_m$ can once more be found numerically by taking $t = e^u$ equal to the zeroes $\tau_n$ of $T_{1/2}$ and solving the linear problem.

The zeroes give the Bethe roots $u_m = \log t_m$. One needs to be careful to interpret the result correctly in the presence of Bethe roots at infinity: $u_m = \pm\infty$ corresponds to $t_m \in \{0, \infty\}$ so the corresponding factor in (C.3) collapses to $t^{\pm 1}$, yielding (5.21):

$$Q(t) = \text{cst}_Q \times t^{n_{-\infty} - n_{+\infty}} \prod_{n=1}^{M - n_{+\infty} - n_{-\infty}} (t_n^{-1} t - t_n t^{-1}). \quad \text{(C.4)}$$

The numerical recipe works very well for the XXZ model away from root of unity, as well as for the XXX model ($\Delta = \pm 1$). However, one cannot find all the Bethe roots for the XXZ spin chain at root of unity, precisely due to the existence of degenerate eigenstates of the transfer matrix $\mathbf{T}_{1/2}$. In that case, our construction for the Q operator still works and gives the correct results, sometimes even analytically.

## C.2  Relation between Bethe roots for anisotropies $\Delta$ and $-\Delta$

In the gapless regime ($-1 < \Delta < 1$) there is a simple relation between Bethe roots of all the physical solutions at anisotropy $\Delta$ and those at anisotropy $-\Delta$, even though the corresponding eigenstates are different since the $\mathbf{B}$-operators in the algebraic Bethe ansatz differ. We will denote the parameters of the second spin chain by primes: $\Delta' = -\Delta$ and

$$\eta = \text{arccosh}(\Delta) \in i\mathbb{R}, \qquad \eta' = \text{arccosh}(\Delta') = i\pi - \eta. \quad \text{(C.5)}$$

Consider any solution to Bethe equation (2.7) with $\eta$, system size $N$ and twist $\phi$: assume that the Bethe roots $\{u_m\}_{m=1}^{M}$ obey

$$\left( \frac{\sinh(u_m + \eta/2)}{\sinh(u_m - \eta/2)} \right)^N \prod_{n(\neq m)}^{M} \frac{\sinh(u_m - u_n - \eta)}{\sinh(u_m - u_n + \eta)} = e^{-i\phi}. \quad \text{(C.6)}$$

Then define $\{u'_m\}_{m=1}^{M}$ by

$$u'_m = -u_m - \frac{i\pi}{2}, \qquad 1 \leq m \leq M. \quad \text{(C.7)}$$

In terms of these parameters (C.6) reads

$$\left( \frac{\sinh(-u'_m + i\pi/2 + \eta/2)}{\sinh(-u'_m + i\pi/2 - \eta/2)} \right)^N \prod_{n(\neq m)}^{M} \frac{\sinh(-u'_m + u'_n - \eta)}{\sinh(-u'_m + u'_n + \eta)} = e^{-i\phi}. \quad \text{(C.8)}$$

This precisely of the form (C.6) with $\eta' = i\pi - \eta$ and twist $\phi'$ chosen such that $e^{-i\phi'} = (-1)^N e^{-i\phi}$. This shows that for each solution $\{u_m\}_{m=1}^M$ at anisotropy $\Delta$ there is a corresponding solution $\{u'_m\}_{m=1}^M$ at $\Delta' = -\Delta$ provided the twist is modified to

$$\phi' = \begin{cases} \phi & N \text{ even}, \\ \phi + \pi & N \text{ odd}. \end{cases} \tag{C.9}$$

The two eigenstates are related by the unitary gauge transformation

$$\mathbf{U} = \exp\left(i\pi \sum_{j=1}^N \frac{j}{2} \sigma_j^z\right) = e^{i\pi L(L+1)/4} \prod_{j=1}^{\lceil N/2 \rceil} \sigma_{2j-1}^z. \tag{C.10}$$

(Removing the prefactor in the expression on the right yields a simpler transformation that also does the job and is its own inverse.) It is easy to check that this transformation changes the sign of $\Delta$ in the Hamiltonian (2.1):

$$\mathbf{U}\mathbf{H}(\Delta, \phi)\mathbf{U}^{-1} = -\mathbf{H}(-\Delta, \phi'). \tag{C.11}$$

Moreover, the eigenstates are related by

$$|\{u'_m\}\rangle \propto \mathbf{U}|\{u_m\}\rangle. \tag{C.12}$$

## C.3 Relation between eigenstates with opposite twist

Recall from Section 5.5 that an $M$-particle Bethe state $|\{u_m\}_{m=1}^M\rangle$ for the XXZ model obeys

$$\begin{aligned} \mathbf{Q}(u, \phi)|\{u_m\}_{m=1}^M\rangle &= Q(u)|\{u_m\}_{m=1}^M\rangle, \\ \mathbf{P}(u, \phi)|\{u_m\}_{m=1}^M\rangle &= P(u)|\{u_m\}_{m=1}^M\rangle, \end{aligned} \tag{C.13}$$

with eigenvalue $Q(u)$ and $P(u)$ of the form

$$\begin{aligned} Q(u) &= \text{cst}_Q \times \prod_{m=1}^M \left(t_m^{-1} t - t_m t^{-1}\right), & t_m &= e^{u_m}, \\ & & t &= e^u, \\ P(u) &= \text{cst}_P \times \prod_{n=1}^{N-M} \left(\tau_n^{-1} t - \tau_n t^{-1}\right), & \tau_n &= e^{v_n}, \end{aligned} \tag{C.14}$$

and where the zeroes $u_m = \log t_m$ of $Q$ are the Bethe roots. Let us show that the $v_n$ can similarly be interpreted as the Bethe roots of the spin-flipped counterpart of $|\{u_m\}_{M=1}^M\rangle$ 'beyond the equator' with opposite twist.

Consider the transfer matrices $\mathbf{T}_s(u)$ with $s \in \frac{1}{2}\mathbb{Z}_{\geq 0}$. Under global spin inversion these operators simply behave as

$$\prod_{j=1}^N \sigma_j^x \, \mathbf{T}_s(u, \phi) \prod_{j=1}^N \sigma_j^x = e^{2si\phi} \, \mathbf{T}_s(u, -\phi). \tag{C.15}$$

To see this note that for the unitary spin-$s$ representation (A.8) the Lax operator (3.2) is invariant under total spin reversal, which acts by conjugation by $\sigma_j^x$ in the physical space and by the antidiagonal matrix $\mathbf{U} := \sum_{n=0}^{2s} |2s-n\rangle\langle n|$ in the auxiliary space. Spin reversal in the physical space is thus equivalent to spin reversal in the auxiliary space. This property is inherited by the monodromy matrix. In the periodic case it follows that spin flip in the physical space does not affect the transfer matrix, as the trace is invariant under reordering the basis. In the twisted

case we only have to correct for the twist (B.1), which is reversed: $\mathbf{U}\mathbf{E}(\phi)\mathbf{U}^{-1}=e^{2\mathrm{s}\mathrm{i}\phi}\mathbf{E}(-\phi)$. This proves (C.15).

Now consider the TQ equation (5.10) with $\phi$ inverted to $-\phi$. By conjugating both sides with the global spin-flip operator $\prod_{j=1}^{N}\sigma_{j}^{x}$, using (C.15) and multiplying both sides by $e^{\mathrm{i}\phi}$ we see that $\bar{\mathbf{Q}}(u,\phi):=\prod_{j=1}^{N}\sigma_{j}^{x}\,\mathbf{Q}(u,-\phi)\prod_{j=1}^{N}\sigma_{j}^{x}$ precisely obeys the TP equation (5.11). Moreover, comparing the eigenvalues in (C.14) shows that the eigenvalues of $\bar{\mathbf{Q}}(u,\phi)$ on $M$-particle Bethe vectors are trigonometric polynomials of degree $N-M$, just as for the P operator. It follows that the eigenvalues of $\bar{\mathbf{Q}}(u,\phi)$ are proportional to those of the P operator; in particular they have the same zeroes:

$$\bar{\mathbf{Q}}(u,\phi)\propto P(u,\phi)\propto\prod_{n=1}^{N-M}\left(\tau_{n}^{-1}\,t-\tau_{n}\,t^{-1}\right). \tag{C.16}$$

Since $\bar{\mathbf{Q}}(u,\phi)$ and $\mathbf{Q}(u,-\phi)$ have the same characteristic polynomial this shows that the $M$-particle eigenvalues of the P operator are the same as the $(N-M)$-particle eigenvalues of $\mathbf{Q}(u,-\phi)$. But we know that the latter can be interpreted as the Bethe roots. Therefore the zeroes of the P operator can be interpreted as the Bethe roots of the spin-reversed Bethe vector beyond the equator with opposite twist.

Finally notice that the Bethe vectors (3.11) are constructed using the B-operator, which is independent of the twist, see (3.9). This implies that the result of reversing all spins on an off-shell Bethe vector (for the Hamiltonian with original twist $\phi$) is

$$\prod_{l=1}^{N}\sigma_{l}^{x}\,|\{u_{m}\}_{m=1}^{M}\rangle=|\{v_{n}\}_{n=1}^{N-M}\rangle\,, \tag{C.17}$$

where the Bethe roots $v_{n}$ beyond the equator are related to the zeroes of eigenvalues of the P operator on $|\{u_{m}\}_{m=1}^{M}\rangle$. (The dependence of the on-shell Bethe vectors on the twist enters through the Bethe equations.)

## D  Alternative proof of Eq. (6.13)

In this appendix we give another proof of (6.13), i.e.

$$\tilde{\mathbf{T}}_{s}(u,\phi)-e^{\mathrm{i}(2s+1)\phi}\,\tilde{\mathbf{T}}_{-s-1}(u,\phi)=\left(1-\varepsilon^{N}e^{\mathrm{i}\ell_{2}\phi}\right)\mathbf{T}_{s}(u,\phi)\,. \tag{D.1}$$

Consider $2s\in\mathbb{Z}_{\geq0}$ with $s<\frac{\ell_{2}}{2}-1$. Proceeding as in Section 5.1 we obtain a decomposition like in (5.5):

$$\tilde{\mathbf{T}}_{s}(u,\phi)=\mathbf{T}_{s}(u,\phi)+e^{\mathrm{i}(2s+1)\phi}\,\mathbf{T}_{s}^{*}(u,\phi)\,. \tag{D.2}$$

Here $\mathbf{T}_{s}^{*}$ denotes the result of restricting the trace over the auxiliary space to the subspace $V_{a}^{*}$ spanned by $|n\rangle$ with $2s+1\leq n\leq\ell_{2}-1$.

The restricted Lax matrix $\mathbf{L}_{sj}^{*}(u)$ coincides with the transpose (in both auxiliary and physical space) $\mathbf{L}_{-s-1,j}(u)^{\mathrm{T}}$ of the Lax operator $\mathbf{L}_{-s-1,j}(u)$, whose auxiliary space has dimension $\ell_{2}-2s-1$. Therefore $\mathbf{T}_{s}^{*}(u,\phi)=\mathbf{T}_{-s-1}(u,\phi)$, and we get

$$\tilde{\mathbf{T}}_{s}(u,\phi)=\mathbf{T}_{s}(u,\phi)+e^{\mathrm{i}(2s+1)\phi}\,\mathbf{T}_{-s-1}(u,\phi)\,. \tag{D.3}$$

Now consider the transfer matrix $\tilde{\mathbf{T}}_{s}^{*}$ obtained like $\tilde{\mathbf{T}}_{s}$ truncating the trace to the basis of $\tilde{V}_{a}$ translated (raised) by $2s+1$, i.e. to $|n+2s+1\rangle$ with $0\leq n\leq\ell_{2}$. Again, since $2s+1<\ell_{2}$ and $\mathbf{S}^{+}|\ell_{2}\rangle=0$, the trace can be split into two transfer matrices. The first one coincides with $\mathbf{T}_{s}^{*}$,

Table 2: Numerical results for the Bethe roots of the Bethe vector $|\{u_m\}_{m=1}^3\rangle$ deforming $|\{\alpha_1^{\text{FM}}\}\rangle$ as a small twist $\phi$ is turned on.

| $\phi$ | $u_1$ | $u_2$ | $u_3$ |
|---|---|---|---|
| 0 | $-0.49887047 - 0.52359877\,\mathrm{i}$ | $-0.49887047 + 0.52359877\,\mathrm{i}$ | $-0.49887047 + 1.5707963\,\mathrm{i}$ |
| $10^{-5}\pi$ | $-0.49886307 - 0.52359861\,\mathrm{i}$ | $-0.49886307 + 0.52359861\,\mathrm{i}$ | $-0.49888095 + 1.5707963\,\mathrm{i}$ |
| $10^{-4}\pi$ | $-0.49879646 - 0.52359713\,\mathrm{i}$ | $-0.49879646 + 0.52359713\,\mathrm{i}$ | $-0.49897526 + 1.5707963\,\mathrm{i}$ |
| $10^{-3}\pi$ | $-0.49813029 - 0.52358228\,\mathrm{i}$ | $-0.49813029 + 0.52358228\,\mathrm{i}$ | $-0.49991897 + 1.5707963\,\mathrm{i}$ |
| $10^{-2}\pi$ | $-0.49147893 - 0.52344350\,\mathrm{i}$ | $-0.49147893 + 0.52344350\,\mathrm{i}$ | $-0.50942065 + 1.5707963\,\mathrm{i}$ |

while the second one is obtained as $\mathbf{T}_s$ in the basis shifted by $\ell_2$, i.e. $|n+\ell_2\rangle$ with $0 \le n \le 2s+1$, and therefore picks up a factor $\varepsilon^N$ compared to $\mathbf{T}_s$ times the twist contribution. Proceeding as before, one can verify that the second transfer matrix in the decomposition of $\tilde{\mathbf{T}}_s^*$ and $\tilde{\mathbf{T}}_{-s-1}$ derive from similar Lax matrices and are both equal to $e^{\mathrm{i}(\ell_2-2s-1)\phi}\,\varepsilon^N\,\mathbf{T}_s$. Thus one has

$$\tilde{\mathbf{T}}_s^*(u,\phi) = \tilde{\mathbf{T}}_{-s-1}(u,\phi) = \mathbf{T}_{-s-1}(u,\phi) + \varepsilon^N e^{\mathrm{i}(\ell_2-2s-1)\phi}\,\mathbf{T}_s(u,\phi)\,. \tag{D.4}$$

Subtracting $e^{\mathrm{i}(\ell_2-2s-1)\phi}$ times (D.3) from (D.4) we obtain (6.13).

# E  Deforming FM strings

## E.1  Tuning a small twist

We study the Bethe roots for states that include FM strings at $\phi = 0$ numerically in the presence of a small twist. The results illustrate that FM strings are formed by combining the last two string types from the string hypothesis.

Consider the case $N = 6$ and $\Delta = \frac{1}{2}$, like the example in Section 2.3. At zero twist $\phi = 0$ the primitive state $|\uparrow\uparrow\uparrow\uparrow\uparrow\rangle$ has, by the results of Section 7.5, corresponding state beyond the equator with Bethe roots (2.21). The descendant tower further includes the states

$$|\{\alpha_1^{\text{FM}}\}\rangle = |\{u_m\}_{m=1}^3\rangle\,, \qquad |\{\alpha_2^{\text{FM}}\}\rangle = |\{v_m\}_{m=1}^3\rangle\,, \tag{E.1}$$

with energy eigenvalues $E = 0$.

Now we turn on a very small twist and study what happens to the corresponding states by computing the eigenstates with $M = 3$ and eigenvalues for the transfer matrices $\mathbf{T}_s$ that are very close to those of (E.1). The resulting Bethe roots are collected in Tables 2–3, where we have expressed the analytic initial values (2.21) numerically to facilitate the comparison. We observe that the Bethe roots are string deviations of two $(2,+)$ strings, namely $u_1, u_2$ and $v_1$, $v_2$, with string deviations that decrease as $\phi$ approaches zero. In the limit $\phi \to 0$ an FM string forms when two strings, here of type $(2,+)$ and $(1,-)$ respectively, have coinciding real part of the string centres. This motivates our conjecture in Section 10.2 about the relation between the last two string types of the string hypothesis and the string centres of FM strings, even for systems with finite system sizes.

## E.2  Tuning the anisotropy

Next we study the behaviour of Bethe roots for states including FM strings at $\phi = 0$ when $\eta$ is slightly deformed. We give two examples, both with $\Delta \approx 1/2$.

First we revisit the example with $N = 6$ from Section 2.3 and Appendix E.1. When $\Delta = 1/2$ the states (E.1) with Bethe roots (2.21), i.e. numerical values given by the first rows of Tables 2–3, are degenerate for the fundamental transfer matrix, with eigenvalue (recall that

Table 3: Numerical results for the Bethe roots of the Bethe vector $|\{v_m\}_{m=1}^3\rangle$ deforming $|\{\alpha_2^{\text{FM}}\}\rangle$ when introducing a small twist $\phi$.

| $\phi$ | $v_1$ | $v_2$ | $v_3$ |
|---|---|---|---|
| 0 | $0.49887047 - 0.52359877\,\mathrm{i}$ | $0.49887047 + 0.52359877\,\mathrm{i}$ | $0.49887047 + 1.5707963\,\mathrm{i}$ |
| $10^{-5}\pi$ | $0.49887788 - 0.52359894\,\mathrm{i}$ | $0.49887788 + 0.52359894\,\mathrm{i}$ | $0.49886000 + 1.5707963\,\mathrm{i}$ |
| $10^{-4}\pi$ | $0.49894452 - 0.52360045\,\mathrm{i}$ | $0.49894452 + 0.52360045\,\mathrm{i}$ | $0.49876570 + 1.5707963\,\mathrm{i}$ |
| $10^{-3}\pi$ | $0.49961090 - 0.52361550\,\mathrm{i}$ | $0.49961090 + 0.52361550\,\mathrm{i}$ | $0.49782342 + 1.5707963\,\mathrm{i}$ |
| $10^{-2}\pi$ | $0.50628580 - 0.52377622\,\mathrm{i}$ | $0.50628580 + 0.52377622\,\mathrm{i}$ | $0.48846413 + 1.5707963\,\mathrm{i}$ |

$t = e^u$)

$$
\begin{aligned}
T_{1/2}(t) &= \frac{1}{32}t^6 - \frac{3}{32}t^4 - \frac{15}{64}t^2 + \frac{5}{8} - \frac{15}{64}t^{-2} - \frac{3}{32}t^{-4} + \frac{1}{32}t^{-6} \\
&= 0.03125\,t^6 - 0.093750000\,t^4 - 0.23437500\,t^2 + 0.62500000 \\
&\quad - 0.23437500\,t^{-2} - 0.093750000\,t^{-4} + 0.03125\,t^{-6} ,
\end{aligned}
\tag{E.2}
$$

where we give the numerical values for later convenience. In particular their energy is $E = 0$. Now we vary the anisotropy a little, taking $\eta = \mathrm{i}(\frac{\pi}{3} \pm \epsilon)$ for small $\epsilon > 0$, to move away from root of unity. The numerically obtained values of the Bethe roots of the two states at $M = 3$ with energy $E \approx 0$ are given in Table 4, including the values that one would obtain as limits at $\epsilon = 0$. (These values obey Eq. (1.11) from [31].) Note that the limiting values at root of unity are rather different from the Bethe roots (2.21) obtained from the zeroes of the Q operator. Here we use the notation $\{u'_m\}_m, \{u''_m\}_m$ rather than $\{u_m\}_m, \{v_m\}_m$ because we cannot distinguish between the two FM strings arising as the zeroes of the Q function at $\epsilon = 0$, i.e. we cannot say which of $\{u'_m\}_m, \{u''_m\}_m$ jump to which of $\{u_m\}_m, \{v_m\}_m$ in the limit.

The corresponding states

$$
\lim_{\epsilon \to 0} |\{u'_m\}_{m=1}^3\rangle, \qquad \lim_{\epsilon \to 0} |\{u''_m\}_{m=1}^3\rangle,
\tag{E.3}
$$

can be constructed via the FM string creation operators of [32] (see Section (8.5)) to produce eigenvectors of $\mathbf{H}$ with $E = 0$. However, these vectors are not eigenvectors of the two-parameter transfer matrix or the Q operator at $\Delta = 1/2$. We find that nontrivial linear combinations of them (namely: a multiple of their sum and difference) equal $|\{u_m\}_{m=1}^3\rangle$ and $|\{v_m\}_{m=1}^3\rangle$, which are eigenvectors of Q operator, respectively.

Despite the jump in the Bethe roots $\{u_m\}_m$ and $\{v_m\}_m$ as $\epsilon \to 0$ we stress that physical

Table 4: The numerical Bethe roots of the two states at $M = 3$ with energy close to zero as $\eta$ changes away from $\mathrm{i}\pi/3$, including the values $0, \mp\mathrm{i}\pi/3$ and $\mathrm{i}\pi/2, \mp\mathrm{i}\pi/6$ (indicated with a question mark) that one would extrapolate as $\phi \to 0$.

| $\eta - \mathrm{i}\pi/3$ | $u'_1$ | $u'_2$ | $u'_3$ | $u''_1$ | $u''_2$ | $u''_3$ |
|---|---|---|---|---|---|---|
| $-\mathrm{i}\,10^{-2}$ | 0 | $-1.03763873\,\mathrm{i}$ | $1.03763873\,\mathrm{i}$ | $1.5707963\,\mathrm{i}$ | $-0.51859878\,\mathrm{i}$ | $0.51859878\,\mathrm{i}$ |
| $-\mathrm{i}\,10^{-4}$ | 0 | $-1.04710210\,\mathrm{i}$ | $1.04710210\,\mathrm{i}$ | $1.5707963\,\mathrm{i}$ | $-0.52354878\,\mathrm{i}$ | $0.52354878\,\mathrm{i}$ |
| 0 | 0? | $-1.04719755\,\mathrm{i}$? | $1.04719755\,\mathrm{i}$? | $1.5707963\,\mathrm{i}$? | $-0.52359878\,\mathrm{i}$? | $0.52359878\,\mathrm{i}$? |
| $\mathrm{i}\,10^{-4}$ | 0 | $-1.04729300\,\mathrm{i}$ | $1.04729300\,\mathrm{i}$ | $1.5707963\,\mathrm{i}$ | $-0.52364878\,\mathrm{i}$ | $0.52364878\,\mathrm{i}$ |
| $\mathrm{i}\,10^{-2}$ | 0 | $-1.05672931\,\mathrm{i}$ | $1.05672931\,\mathrm{i}$ | $1.5707963\,\mathrm{i}$ | $-0.52859878\,\mathrm{i}$ | $0.52859878\,\mathrm{i}$ |

quantities are continuous. For instance, fundamental transfer matrix has eigenvalue

$$
\begin{aligned}
T_{1/2}(t) = {} & 0.03125\,t^6 - 0.093758120\,t^4 - 0.23428568\,t^2 + 0.62482949 \\
& - 0.23428568\,t^{-2} - 0.093758120\,t^{-4} + 0.03125\,t^{-6}
\end{aligned}
\quad \epsilon = -10^{-4},
$$

$$
\begin{aligned}
T_{1/2}(t) = {} & 0.03125\,t^6 - 0.093758120\,t^4 - 0.23428568\,t^2 + 0.62482949 \\
& - 0.23428568\,t^{-2} - 0.093758120\,t^{-4} + 0.03125\,t^{-6}
\end{aligned}
\quad \epsilon = +10^{-4},
$$

(E.4)

for $|\{u'_m\}_{m=1}^3\rangle$ and

$$
\begin{aligned}
T_{1/2}(t) = {} & 0.03125\,t^6 - 0.093750738\,t^4 - 0.23430930\,t^2 + 0.62486049 \\
& - 0.23430930\,t^{-2} - 0.093750738\,t^{-4} + 0.03125\,t^{-6}
\end{aligned}
\quad \epsilon = -10^{-4},
$$

$$
\begin{aligned}
T_{1/2}(t) = {} & 0.03125\,t^6 - 0.093766977\,t^4 - 0.23437426\,t^2 + 0.62505535 \\
& - 0.23437426\,t^{-2} - 0.093766977\,t^{-4} + 0.03125\,t^{-6}
\end{aligned}
\quad \epsilon = +10^{-4},
$$

(E.5)

for $|\{u''_m\}_{m=1}^3\rangle$. These polynomials are very close to (E.2).

The second example is $N = 10$. There is an eigenstate $|\{u_m\}_{m=1}^4\rangle$ with energy eigenvalue $E = 1/2$ when $\Delta = \frac{1}{2}$. Its Bethe roots are

$$
u_1 = \frac{i\pi}{2},
$$
$$
u_2 = -0.56101744 - \frac{i\pi}{6}, \quad u_3 = -0.56101744 + \frac{i\pi}{6}, \quad u_4 = -0.56101744 + \frac{i\pi}{2}.
$$

(E.6)

Here $u_2, u_3, u_4$ form an FM string. The numerically obtained eigenvalue for $\mathbf{T}_{1/2}$ is

$$
\begin{aligned}
\mathbf{T}_{1/2}(u,0)|\{u_m\}_{m=1}^4\rangle = \big( & 0.0009765625\,t^{10} - 0.022460937\,t^8 + 0.076171875\,t^6 \\
& + 0.084960938\,t^4 - 0.72949219\,t^2 + 1.1806641 \\
& - 0.72949219\,t^{-2} + 0.084960938\,t^{-4} + 0.076171875\,t^{-6} \\
& - 0.022460937\,t^{-8} + 0.0009765625\,t^{-10} \big)|\{u_m\}_{m=1}^4\rangle,
\end{aligned}
$$

(E.7)

where $t = e^u$.

The behaviour of the Bethe roots under a small change of $\eta$ is given in Table 5. We see that the Bethe roots change drastically, jumping to form a 2-string and two 1-strings with odd parity. The same is true for the eigenvalues of the Q operator. On the other hand, one can check that the eigenvalues of $\mathbf{T}_s$ change smoothly. For instance, the eigenvalue of $\mathbf{T}_{1/2}$ for the corresponding state at $\eta = i(\pi/3 + 10^{-5})$ is very close to (E.7):

$$
\begin{aligned}
\mathbf{T}_{1/2}(u,0)|\{u_m\}_{m=1}^4\rangle = \big( & 0.00097654559\,t^{10} - 0.022460999\,t^8 + 0.076172376\,t^6 \\
& + 0.084963582\,t^4 - 0.72950736\,t^2 + 1.18068832 \\
& - 0.72950736\,t^{-2} + 0.084963582\,t^{-4} + 0.076172376\,t^{-6} \\
& - 0.022460999\,t^{-8} + 0.00097654559\,t^{-10} \big)|\{u_m\}_{m=1}^4\rangle.
\end{aligned}
$$

(E.8)

# F  Examples of last two string types of TBA at non-principal root of unity

We present several examples on the last two string types of the TBA at root of unity using Takahashi's notation [78, 79] at non-principal root of unity to illustrate our conjecture (10.8) for the string centre of FM strings.

Table 5: The numerical Bethe roots of $|\{u_m\}_{m=1}^4\rangle$ as $\eta$ changes away from $i\pi/3$.

| $\eta - i\pi/3$ | $u_1$ | $u_2$ | $u_3$ | $u_4$ |
|---|---|---|---|---|
| 0 | $1.5708\,i$ | $-0.56102 + 1.5708\,i$ | $-0.56102 - 0.52360\,i$ | $-0.56102 + 0.52360\,i$ |
| $i10^{-6}$ | $-4.9936 \times 10^{-4} + 1.5708\,i$ | $4.9936 \times 10^{-4} + 1.5708\,i$ | $2.6516 \times 10^{-5} - 0.52360\,i$ | $2.6516 \times 10^{-5} + 0.52360\,i$ |
| $i10^{-5}$ | $-1.5382 \times 10^{-3} + 1.5708\,i$ | $1.5382 \times 10^{-3} + 1.5708\,i$ | $2.9364 \times 10^{-6} - 0.52360\,i$ | $2.9364 \times 10^{-6} + 0.52360\,i$ |
| $i10^{-4}$ | $-4.8592 \times 10^{-3} + 1.5708\,i$ | $4.8592 \times 10^{-3} + 1.5708\,i$ | $2.5441 \times 10^{-7} - 0.52360\,i$ | $2.5441 \times 10^{-7} + 0.52360\,i$ |
| $i10^{-3}$ | $-0.015342 + 1.5708\,i$ | $0.015342 + 1.5708\,i$ | $3.2550 \times 10^{-8} - 0.52360\,i$ | $3.2550 \times 10^{-8} + 0.52360\,i$ |

We start with $\eta = \frac{2i\pi}{3}$, which is a non-principal root of unity. The allowed string types are $(1,-)$, $\underline{(2,+)}$, $\underline{(1,+)}$. Here we have underlined the last two string types, which are of the form

$$\alpha_1 = \alpha - \frac{i\pi}{3}, \quad \alpha_2 = \alpha + \frac{i\pi}{3}, \qquad \alpha \in \mathbb{R},$$
$$\alpha_3 = \alpha', \qquad\qquad\qquad \alpha' \in \mathbb{R}. \tag{F.1}$$

According to the conjecture (10.8) the FM string with $\eta = \frac{2i\pi}{3}$ should be expressed as

$$\alpha_1' = \alpha^{\mathrm{FM}} + \frac{i\pi}{3}, \quad \alpha_2' = \alpha^{\mathrm{FM}}, \quad \alpha_3' = \alpha^{\mathrm{FM}} - \frac{i\pi}{3}, \qquad \alpha^{\mathrm{FM}} \in \mathbb{R}. \tag{F.2}$$

Clearly, (F.1) and (F.2) describe the same FM string when the real parts of the string centres coincide. This results are confirmed by all the examples in Section 8 with finite-size calculations.

For another example with non-principal root of unity we consider $\eta = \frac{2i\pi}{5}$. Using the method in Chapter 9.2 of [79], we obtain the allowed string types which are $(1,+)$, $(1,-)$, $\underline{(3,+)}$, $\underline{(2,+)}$, where we again underline the last two string types, given by

$$\alpha_1 = \alpha - \frac{2i\pi}{5}, \quad \alpha_2 = \alpha, \quad \alpha_3 = \alpha + \frac{2i\pi}{5}, \qquad \alpha \in \mathbb{R},$$
$$\alpha_4 = \alpha' - \frac{i\pi}{5}, \quad \alpha_5 = \alpha' + \frac{i\pi}{5}, \qquad\qquad \alpha' \in \mathbb{R}. \tag{F.3}$$

From the conjecture (10.8) the FM string should be expressed a

$$\alpha_1' = \alpha^{\mathrm{FM}} + \frac{2i\pi}{5}, \quad \alpha_2' = \alpha^{\mathrm{FM}} + \frac{i\pi}{5}, \quad \alpha_3' = \alpha^{\mathrm{FM}},$$
$$\alpha_4' = \alpha^{\mathrm{FM}} - \frac{i\pi}{5}, \quad \alpha_5' = \alpha^{\mathrm{FM}} - \frac{2i\pi}{5}, \qquad \alpha^{\mathrm{FM}} \in \mathbb{R}. \tag{F.4}$$

Again, when the real parts of the string centres in (F.3) and (F.4) coincide they describe the same FM string.

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
