# Peer review of "On the Q operator and the spectrum of the XXZ model at root of unity"

_SciPost Physics, doi:SciPost Phys. 11, 067 (2021)_

## Round 2 · Referee Report · Anonymous (Referee 1) · 2021-5-1

Strengths

1 - interesting algebraic method to build the Q operator at root of unity
2 - most of the paper is well explained and well written

Weaknesses

1 - The relation with previous works is not sufficiently explained, see the report

Report

The authors provide an algebraic construction of the Q function for any state in XXZ at root of unity and twist. The difficulty at root of unity is that the degeneracies in the spectrum correspond to "Fabricius MacCoy" strings whose expressions cannot be determined directly from the Bethe equations. A number of related results are obtained as a by-product. The paper is well written and interesting. However I have the following comments:

  • the information that is missing in Bethe's equations to build the Q is the \alphas, the FM string centres. It is said at different places that the method presented in the paper enables one to compute these \alpha. But in [29] is also given a set of equations for the \alphas, by studying their deformation when the anisotropy is changed. Why is that never mentioned? It is said in the abstract that "there are still open issues regarding the spectrum at root of unity", and mention as an example in the introduction the fact that the Q cannot be constructed from the BE. But why the equations of [29] do not answer that? The authors even study this deformation of strings in section 8.4 and appendix E, and say (in contradiction with [29]) that they are discontinuous. That is also argued in section 10.3 to make a connection with the fractal nature of the Drude weight. Yet the authors of [29] claim to have checked numerically their equations for the \alpha in small sizes. An important mention of this previous result on the \alphas and a discussion about this disagreement are missing.

  • I am not sure to understand the take-home message of section 10. It is said that the states with FM strings cannot be distinguished with Ts, but with the truncated transfer matrix they can be distinguished by the quasilocal Z charge. But they can be simply distinguished by the local charge that is the magnetization, right? It is said then that for this reason the Z charges have to be included in the GGE. But that was known without resorting to FM strings? To be sure to understand what is meant: are the authors claiming "TBA is incomplete at root of unity because one would have to include these FM strings too"? Above eq (10.6) is said that sometimes two Bethe strings form a FM string if their centre coincide, and so in this sense FM strings appear in the root densities that characterize the GGE. From what is written, I am not quite convinced these FM strings are Bethe strings. First, Bethe strings are very often only approximate, contrary to FM strings whose exactness is a defining feature, so the fact that their centre coincide is not enough to say that it is a FM string. Second, standard Bethe vectors are always highest weight, and so should be all states with two Bethe strings without FM strings. Shouldn't the fact that FM strings are not highest weight disqualify them from being viewed as two Bethe strings? Also, since there is a FM string, there should be another solution to the BE without FM string with the same eigenvalue, thus without these two Bethe strings, and thus describable without string root density at all. So if we count the FM string as a Bethe string this eigenvalue will be counted twice, whereas the degeneracy is usually not counted (for example there is no "root density" to take into account the infinite roots in XXX). At the end of the day I cannot understand if the authors really claim that these FM strings should be taken into account by a root density in the TBA formalism or not, whether it is on top of the two strings solution or instead of. Finally they say below (10.6) that they "checked" in small sizes that FM strings "can be viewed" as the last two string types, which I don't quite understand the meaning. Does it mean the authors checked that these FM strings have a property that usual FM strings don't have but that Bethe strings have? Or that they counted their numbers? That they varied phi or eta? For these reasons, Section 10 which contains "hints" of applications of these FM strings seems to me less well argued and less clear than the rest of the paper.

  • the discussion of section 2.2 is premised on the assumption that roots at infinity do not scatter among themselves, as said above (2.12). It would be interesting to know if it is well supported analytically or numerically.

  • the method described in Appendix C is often referred to as "McCoy's method" and seems to have been first used in G. Albertini, S. Dasmahapatra and B. McCoy, Int. J. Mod. Phys. A 7, Suppl. 1A (1992) (Spectrum and completeness of the integrable three state Potts model: A Finite size study)

Requested changes

1 - discuss the equations of [29] for the center of FM strings in relation with the results of the draft
2 - clarify the aspects mentioned in the report about section 10

---

## Round 2 · Referee Report · Anonymous (Referee 2) · 2021-6-6

Strengths

1- a novel two-parameter transfer matrix is introduced and the corresponding factorization and TQ relation are derived 2- a proof of the interpolation formula that had been conjectured and used in previous work is provided 3- a better understanding of the Fabricius McCoy strings is achieved and their connection with the last two strings in the Takahashi construction is elucidated

Weaknesses

3- A connection with the general theory of quantum groups representation is not completely clear.

Report

This work analyses the spectrum and the hierarchy of transfer matrices for the XXZ (or six vertex) model at root of unity. It presents the construction of a novel two-parameter transfer matrix for the XXZ at root of unity, which generalises the already known transfer matrices for different auxiliary spin representation, including the one with a complex spin. For this object, a factorization property is derived and the corresponding TQ relation is obtained. This approach has several advantages, providing a neater derivation of the already known TQ relation and of the fusion relation between higher transfer matrices. Moreover, it provides a rigorous derivation of the interpolation formula for the spectrum of of the complex spin transfer matrix.
Additionally a clearer understanding of the Fabricius-McCoy strings is obtained. Although they do not contribute to the eigenvalues of the standard transfer matrices, they play a role for the complex spin one and are thus relevant for dynamical phenomena of spin transport.

I think that this is a nice work which elucidates important aspects of one of the most studied integrable model in the literature. Considering that it is a very technical and mathematical work, it is understandable and well-written.

I would have appreciated a discussion about how the construction of the two parameter transfer matrix and its corresponding factorization is generally associated with the quantum group structure underlying the spin chain. For instance, are primary and descendants states clearly related to highest weights of the quantum group representations? This kind of more abstract discussion would be for instance necessary to generalise the current construction to other models, a task which the author have included in their perspectives. I can understand that such a generalisation would deserve a paper on its own, but a few words about it could be useful.

Requested changes

1- I find that the introductory discussion about trasfer matrices in Sec 3.1 is a bit too vague. Of course, a reader experienced in the field can follow, but as it is the paper is not very self contained on this part. Since transfer matrices are the main object of this work, I think that a few explicit formulas and examples about how they are constructed would be helpful.

2- After Eq. 2.5 it is said that H does not commute with the q-deformed SU(2). Does that require a fine tuning of the twist? Can the authors clarify this comment a bit further?

3- At the end of page 12 it is said that transfer matrices corresponding to different auxiliary spaces commute one with the other. While this is known to be true, I think that it would require a generalization of Eq. 3.3.

4- In the discussion about the string-charge duality has been included, it is unclear whether a generalization of Eq. 7 in Ref. 36 has been obtained beyond the principal roots of unity. Does the mapping between the FM root density and the last two Takahashi strings help for this?

---

## Round 3 · Referee Report · Anonymous (Referee 1) · 2021-7-28

Report

The authors have satisfactorily answered the requests and the paper is ready for publication.

---

## Round 3 · Referee Report · Anonymous (Referee 2) · 2021-9-7

Report

I am satisfied with the answers that the authors have provided to my previous report and I can recommend the current version of the manuscript for publication.

---

## Round 3 · Author Response

Dear referees and editor,

We thank the referees for their comments and suggestions. We hope that we have satisfactorily addressed all questions in our resubmission, supplemented by the comments that follow.

Report 2

``I would have appreciated a discussion about how the construction of the two parameter transfer matrix and its corresponding factorization is generally associated with the quantum group structure underlying the spin chain. For instance, are primary and descendants states clearly related to highest weights of the quantum group representations? [...] I can understand that such a generalisation would deserve a paper on its own, but a few words about it could be useful.''
-- We do not use such a connection between primary and descendant states on the one hand, and highest-weight properties for a quantum group on the other. We have made this more explicit by adding a remark at the end of Sect 7.1; see also Sect 8.5 for a related discussion, and the remark at the end of Sect 10.2.

Requested changes

``1- I find that the introductory discussion about trasfer matrices in Sec 3.1 is a bit too vague. Of course, a reader experienced in the field can follow, but as it is the paper is not very self contained on this part. Since transfer matrices are the main object of this work, I think that a few explicit formulas and examples about how they are constructed would be helpful.''
-- We have reformulated the start of Sect 3.1, and added Eq (3.13) to address this.

``2- After Eq. 2.5 it is said that H does not commute with the q-deformed SU(2). Does that require a fine tuning of the twist? Can the authors clarify this comment a bit further?''
-- We have added a footnote on Page 6 to comment on this.

``3- At the end of page 12 it is said that transfer matrices corresponding to different auxiliary spaces commute one with the other. While this is known to be true, I think that it would require a generalization of Eq. 3.3.''
-- We have added a comment in parentheses to clarify this on Page 13.

``4- In the discussion about the string-charge duality has been included, it is unclear whether a generalization of Eq. 7 in Ref. 36 has been obtained beyond the principal roots of unity. Does the mapping between the FM root density and the last two Takahashi strings help for this?''
-- String-charge duality has been investigated beyond principal root of unity in Ref. [81] for instance. In fact, it has been used extensively to investigate the out-of-equilibrium properties of quantum integrable models in recent years, cf. Ref. [12], [41] and [81]. We have added a footnote to point this out on Page 52. See also the new examples illustrating our conjecture beyond principal root of unity in Appendix F

Report 1

``- the information that is missing in Bethe's equations to build the Q is the \alphas, the FM string centres. It is said at different places that the method presented in the paper enables one to compute these \alpha. But in [29] is also given a set of equations for the \alphas, by studying their deformation when the anisotropy is changed. Why is that never mentioned? It is said in the abstract that "there are still open issues regarding the spectrum at root of unity", and mention as an example in the introduction the fact that the Q cannot be constructed from the BE. But why the equations of [29] do not answer that? The authors even study this deformation of strings in section 8.4 and appendix E, and say (in contradiction with [29]) that they are discontinuous. That is also argued in section 10.3 to make a connection with the fractal nature of the Drude weight. Yet the authors of [29] claim to have checked numerically their equations for the \alpha in small sizes. An important mention of this previous result on the \alphas and a discussion about this disagreement are missing.''
-- We thank the referee for pointing this out. Note that Ref [29] is now [31]. We have mentioned and commented on this in various places: Sect 2.3 (p9, preceding the example), the new Section 8.5 dedicated to discussing the differences, as well as Appendix E.2.

``- I am not sure to understand the take-home message of section 10. It is said that the states with FM strings cannot be distinguished with Ts, but with the truncated transfer matrix they can be distinguished by the quasilocal Z charge. But they can be simply distinguished by the local charge that is the magnetization, right?''
-- For states that are degenerate with eigenvalues of $\mathbf{T}_s (u)$, they might share the same magnetisation, see e.g. Fig. 4 on Page 37. For those state, the transfer matrices $\mathbf{T}_s (u)$ cannot distinguish them and they differ from each other by different FM strings in their Q functions, which can be distinguished by the truncated transfer matrix $\tilde{\mathbf{T}}_s (u)$ or the Z charges.

``It is said then that for this reason the Z charges have to be included in the GGE. But that was known without resorting to FM strings?''
-- Indeed, the Z charges were originally needed as an odd-parity charge that could account for non-zero spin-Drude weight. In this work we offer another, finite-size motivation for the consideration of Z charges.

``To be sure to understand what is meant: are the authors claiming "TBA is incomplete at root of unity because one would have to include these FM strings too"?''
-- We do not claim that the TBA is incomplete, but rather suggest that it is complete, and that FM strings have been automatically accounted for via string-charge duality. See the discussions on Page 52.

``Above eq (10.6) is said that sometimes two Bethe strings form a FM string if their centre coincide, and so in this sense FM strings appear in the root densities that characterize the GGE. From what is written, I am not quite convinced these FM strings are Bethe strings. First, Bethe strings are very often only approximate, contrary to FM strings whose exactness is a defining feature, so the fact that their centre coincide is not enough to say that it is a FM string.''
-- FM strings are exact even for any finite-size system. However, as illustrated in Appendix E.1, when introducing a small twist the FM strings split into two Bethe strings (allowed by string hypothesis in thermodynamic limit) that each have string deviations due to the finite-size effect (i.e. that are not exact: note that the imaginary values differ even for u_1, u_2 in Table 2 on Page 65).

``Second, standard Bethe vectors are always highest weight, and so should be all states with two Bethe strings without FM strings. Shouldn't the fact that FM strings are not highest weight disqualify them from being viewed as two Bethe strings? Also, since there is a FM string, there should be another solution to the BE without FM string with the same eigenvalue, thus without these two Bethe strings, and thus describable without string root density at all.''
-- At finite size there are only FM strings and we do not rely on any string hypothesis; the relation with the last two string of Takahashi only appears when comparing to the known results of TBA in the thermodynamic limit.

``So if we count the FM string as a Bethe string this eigenvalue will be counted twice, whereas the degeneracy is usually not counted (for example there is no "root density" to take into account the infinite roots in XXX).''
-- In fact, for XXX, although sl_2 descendants (infinite rapidities) usually play a negligible role in thermodynamics, ideally one would like to be able to take into account their densities too: this would e.g. be relevant for a Néel quench. See for instance Appendix F of Brockman et al, "Quench action approach for releasing the Néel state into the spin-1/2 XXZ chain" [arXiv:1408.5075].

``At the end of the day I cannot understand if the authors really claim that these FM strings should be taken into account by a root density in the TBA formalism or not, whether it is on top of the two strings solution or instead of.''
-- The FM string density will be determined from the other densities through some functional equation. For a trivial example: if root density of the last string type of Takahashi equal the hole density of the one-to-last string type (or reversely), then the FM string density will vanish. Also, if the last two string types of Takahashi have the same root and hole densities separately, then this will equal the FM string density. These can be inferred from the last line of Eq. (9.66) in Takahashi's book that the total densities of the last two strings coincide (There is a typo in the last line, which should read $\rho_{m_l -1 } + \rho_{m_l -1 }^h = \rho_{m_l } + \rho_{m_l }^h = s_l \ast \rho_{m_l -2 }^h$).

``Finally they say below (10.6) that they "checked" in small sizes that FM strings "can be viewed" as the last two string types, which I don't quite understand the meaning. Does it mean the authors checked that these FM strings have a property that usual FM strings don't have but that Bethe strings have? Or that they counted their numbers? That they varied phi or eta? For these reasons, Section 10 which contains "hints" of applications of these FM strings seems to me less well argued and less clear than the rest of the paper.''
-- We have modified the discussion in Sect 10.2 and added Appendix F on Page 67 to clarify this.

``- the discussion of section 2.2 is premised on the assumption that roots at infinity do not scatter among themselves, as said above (2.12). It would be interesting to know if it is well supported analytically or numerically.''
-- Although naively the S-matrix from Eq. (2.11) is not unity for two infinite roots, its contribution is very simple and can instead be reinterpreted as a twist, see Eqs. (2.14)--(2.15). In terms of algebraic Bethe ansatz, $m$ infinite roots (say roots at $-\infty$) corresponds to operator $( \mathbf{S}^- )^m$, cf. Eq. (A.7) on Page 57, which should be taken into account all together.

``- the method described in Appendix C is often referred to as "McCoy's method" and seems to have been first used in
G. Albertini, S. Dasmahapatra and B. McCoy, Int. J. Mod. Phys. A 7, Suppl. 1A (1992) (Spectrum and completeness of the integrable three state Potts model: A Finite size study)''
-- We included the term and reference on Page 60.

Requested changes

``1 - discuss the equations of [29] for the center of FM strings in relation with the results of the draft''''
-- See above.

``2 - clarify the aspects mentioned in the report about section 10''
-- See above.

---

## Round 3 · List of Changes

p3: added "References" to table of contents
p4: added references [17], [19]
p4: added references [37], [39]
p4 and onwards: we have changed the notation for $\mathfrak{sl}_2$ loop algebra, which is denoted as $\widehat{\mathfrak{sl}_2}$ in the previous version (which is usually used for the affine Lie algebra, the central extension of loop algebras).
p6: added footnote to address second request of Report 2
p7: added sentence to about our meaning of Bethe root to address first request of Report 1
p7: moved Footnote 3 here (previously Footnote 7 on p35); updated references to that footnote elsewhere (e.g. p36)
p9: reformulated the paragraph before the example to address first request of Report 1
p9: fixed reference to subsection with conjecture
p10: clarified meaning of [3]_q ! following Eq (2.22)
p13: reformulated start of Sect 3.1 to address first request of Report 2
p13: added half sentence to address third request of Report 2
p30: added remark to address comment of Report 2
p45: improved formatting of Table 1
p45: added Sect 8.5
p51--53: reformulated some sentences to address second request of Report 1
p52: added footnote to address fourth request of Report 2
p53: added remark to address second comment in Report 1
p55: added sentence to thank the referees
p60: added sentence and reference to address fourth comment in Report 1
p65: improved formatting of tables and corrected typo
p65: commented on connection with Ref [31] to address first request of Report 1
p67–68: added Appendix F to address first request of Report 1
p68 onwards: updated the references, in particular added all arXiv preprints links to the references.

---

## Editorial Decision

published